# P-selectin axis plays a key role in microglia immunophenotype and glioblastoma progression

Eilam Yeini [1], Paula Ofek[1], Sabina Pozzi [1], Nitzan Albeck [1,2], Dikla Ben-Shushan [1], Galia Tiram [1], Sapir Golan [1], Ron Kleiner [1], Ron Sheinin[3], Sahar Israeli Dangoor [1], Shlomit Reich-Zeliger[4], Rachel Grossman[5], Zvi Ram[5], Henry Brem [6], Thomas M. Hyde [7,8,9], Prerna Magod[10], Dinorah Friedmann-Morvinski [2,10], Asaf Madi [3] & Ronit Satchi-Fainaro [1,2✉]

Glioblastoma (GB) is a highly invasive type of brain cancer exhibiting poor prognosis. As such, its microenvironment plays a crucial role in its progression. Among the brain stromal cells, the microglia were shown to facilitate GB invasion and immunosuppression. However, the reciprocal mechanisms by which GB cells alter microglia/macrophages behavior are not fully understood. We propose that these mechanisms involve adhesion molecules such as the Selectins family. These proteins are involved in immune modulation and cancer immunity. We show that P-selectin mediates microglia-enhanced GB proliferation and invasion by altering microglia/macrophages activation state. We demonstrate these findings by pharmacological and molecular inhibition of P-selectin which leads to reduced tumor growth and increased survival in GB mouse models. Our work sheds light on tumor-associated microglia/macrophage function and the mechanisms by which GB cells suppress the immune system and invade the brain, paving the way to exploit P-selectin as a target for GB therapy.

[1] Department of Physiology and Pharmacology, Sackler Faculty of Medicine, Tel Aviv University, Tel Aviv, Israel. [2] Sagol School of Neurosciences, Tel Aviv University, Tel Aviv, Israel. [3] Department of Pathology, Sackler Faculty of Medicine, Tel Aviv University, Tel Aviv, Israel. [4] Department of Immunology, Weizmann Institute of Science, Rehovot, Israel. [5] Department of Neurosurgery, Tel Aviv Sourasky Medical Center, Tel Aviv, Israel. [6] Department of Neurosurgery, Johns Hopkins University School of Medicine, Baltimore, MD, USA. [7] Lieber Institute for Brain Development, Johns Hopkins Medical Campus, Baltimore, MD, USA. [8] Department of Psychiatry & Behavioral Science, Johns Hopkins University School of Medicine, Baltimore, MD, USA. [9] Department of Neurology, Johns Hopkins University School of Medicine, Baltimore, MD, USA. [10] Department of Biochemistry and Molecular Biology, George S. Wise Faculty of Life Sciences, Sherman Building, Tel Aviv University, Tel Aviv, Israel. ✉email: ronitsf@tauex.tau.ac.il

Glioblastoma (GB), the most common and lethal type of primary brain tumor[1], is a highly heterogeneous tumor characterized by enhanced angiogenesis[2]. Even with the latest standard of care, which includes surgery followed by chemotherapy and radiotherapy, the median survival is only 20 months[3]. Complete surgical removal of the tumor is challenging due to the invasive nature of the disease[4]. Therefore, uncovering new therapeutic targets involved in GB establishment and progression could be of immense use. It is well known that the tumor microenvironment and the immune system play an important role in tumor progression[5]. Although T cells are not abundant in the brain microenvironment, the myeloid lineage comprises 30% of the cells found in GB tumors, with recent accumulating evidence for their involvement in GB tumorigenesis[6].

Microglia are macrophage-like cells that serve as the brain immune system. Both microglia cells and bone marrow-derived macrophages (BMDM), which are recruited to the brain in pathological conditions, have been shown to have a role in GB progression[7]. Glioma-associated microglia/macrophages (GAMs) can be found in different activation states, presenting a complex and heterogeneous population. GAMs may possess a pro-inflammatory/anti-tumorigenic phenotype in which they are phagocytic, cytotoxic, and bear antigen-presentation capabilities[8]. This phenotype allows microglia/macrophages to attack transformed-cancerous cells and harness cytotoxic T cells against the tumor. In contrast, GAMs may also be often found in anti-inflammatory/pro-tumorigenic phenotype which is related to tissue repair. This phenotype is characterized by tissue remodeling and angiogenesis properties and the secretion of anti-inflammatory cytokines such as IL-10 and TGF-$\beta$[9] which support tumor development.

Although mixed populations are found in GB tumors, GAMs contribute to immune-escape and promote tumor progression, and have been shown to actively enhance glioma growth and invasion, and secrete angiogenic factors[10]. Despite the evidence pointing at the role of GAMs in gliomas, current immunotherapies have not yet been demonstrated to improve survival for GB patients in a clinically-significant manner. This emphasizes the need for alternative or combination approaches[11]. Although the exact role and activation state of GAMs are still unclear, reverting their phenotype to a more pro-inflammatory state has been shown to be a promising therapeutic approach for glioma and might also improve the outcome of immunotherapies in this disease. Several targets associated with this phenotypic switch, such as CSF-1R and STAT3, are currently under pre-clinical and clinical investigation for the treatment of GB[12]. Hence, uncovering new immune-modulators that could regulate GAMs phenotype in GB is of great importance and could lead to the development of new therapeutic strategies to treat this devastating disease.

We hypothesized that P-selectin (SELP), an adhesive molecule involved in leukocyte rolling, may represent one such immune-modulator. SELP is known to be involved in several immunological processes such as platelet activation and leukocyte recruitment and function[13,14]. In addition, there is accumulating evidence to suggest that SELP and P-selectin glycoprotein ligand-1 (PSGL-1) are involved in the spreading and metastasis of melanoma and colon cancer via platelet activation[15]. Recent studies have shown that PSGL-1 serves as an immune checkpoint expressed on T cells[16]. In addition, we and others have previously shown that SELP is expressed by different cancer cells, including our recent report of overexpression of SELP in GB cells in vitro and in vivo[17]. Despite these observations, SELP has so far only been exploited for selective delivery of therapeutics, and its role in GB tumorigenesis remained unexplored[18].

Here, we investigate the functional role of SELP in GB progression. We show that when GB cells and microglia/macrophages are exposed to each other, GB cells overexpress and over-secrete SELP and microglia/macrophages overexpress SELP and PSGL-1. To elucidate the role of the SELP-PSGL-1 axis in GB progression, we blocked SELP function by either a commercially-available SELP inhibitor (SELPi), a neutralizing anti-SELP antibody and by generating several SELP-knockdown GB cells by retroviral infection of five murine and a pool of three human shRNA sequences. Single Cell RNA-seq analysis of GB-shSELP-derived microglia/macrophages reveals an increase in pro-inflammatory and T cell recruitment signatures compared to GB-negative control shRNA (shNC)-derived microglia/macrophages. Indeed, our results show that blocking SELP function leads to delayed tumor growth, prolonged survival and improved immune infiltration in vivo. These findings suggest that SELP expression on GB cells modulates microglia/macrophages activation and T cell recruitment. Here we demonstrate the key involvement of this axis in GB-GAMs interactions, promoting proliferation, invasion and immune modulation presenting a potential therapeutic approach for GB.

## Results

**GB-microglia interactions promote GB cell migration and proliferation**. Our initial objective was to investigate the role of microglia in GB progression. Using patient GB samples, human xenogeneic and murine syngeneic GB mouse models, we detected the presence of activated microglia in the tumor by Iba1 immunostaining (Fig. 1A). We then proceeded to evaluate the effect of microglia on GB cell proliferation and migration in vitro. To that end, we have used commercially-available human microglia and freshly-isolated murine microglia from naïve mice by CD11b positive selection. These microglia cells were characterized by the specific microglia markers TMEM119 and P2Y12 (Supplementary Fig. 1C, D). Co-culture proliferation assay and TransWell migration assay revealed increased proliferation and migration of patient-derived GB cells (PD-GB4) in the presence of human microglia (Fig. 1B, C). In addition, we found that PD-GB4 cells facilitate the proliferation and migration of human microglia as well (Supplementary Fig. 1A, B), indicating the reciprocal activation of microglia following the interaction with GB cells in our models. This suggests that GB cells may induce microgliosis in the brain as a result of the neuroinflammation induced by their crosstalk. We observed a direct correlation between the increased proliferation rate of murine GL261 GB cells and the concentration of murine microglia in the culture, and increased migration of GL261 cells towards murine microglia (Supplementary Fig. 2A, B). These findings were reproduced using human U251 GB cell line showing enhanced proliferation and migration of U251 cells following the interactions with human microglia (Supplementary Fig. 2D, E).

**GB-microglia interactions induce high expression of SELP that facilitates GB progression**. In order to detect secreted factors that play a key role in GB-microglia interactions and regulation of tumor progression, we performed protein-arrays of human PD-GB4 GB cells and primary human microglia. Our findings revealed several factors that were over-secreted following co-culture of GB cells with microglia (Fig. 1D). We then screened for over-secreted factors in a co-culture of primary murine microglia and murine GL261 GB cells, to verify whether the detected factors are relevant for both human and murine GB models (Supplementary Fig. 2C). Interestingly, in both assays, we found a significant increase in the secretion of SELP in the co-culture medium compared to the mono-cultures, which led us to further investigate the role of SELP in GB progression. The corresponding location on the cytokine array membranes of all the

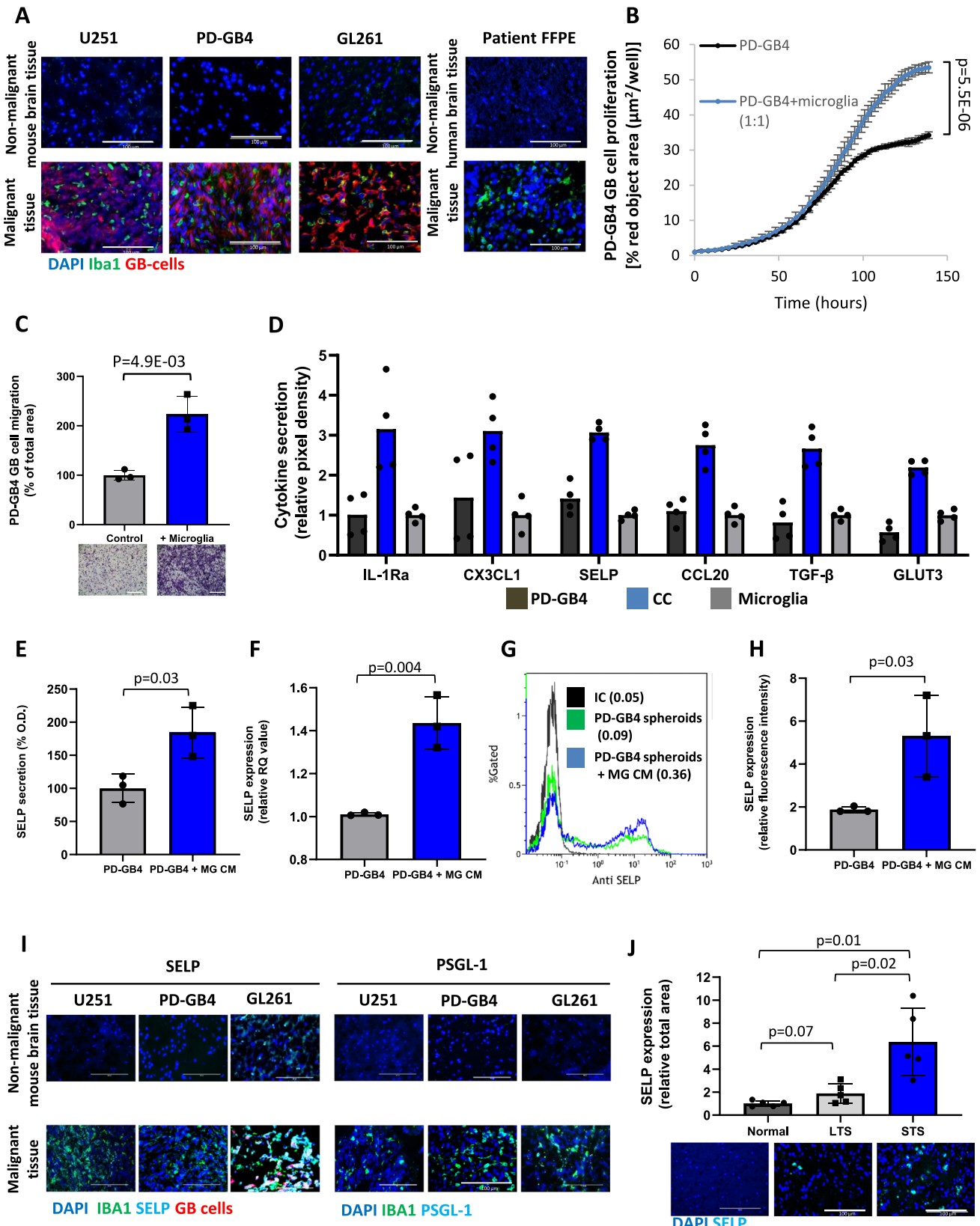

cytokines identified, as well as their gene IDs, are presented in Supplementary Fig. 4.

Next, we performed ELISA and RT-PCR to validate the secretion and expression of SELP by PD-GB4 cells. We found enhanced secretion and high expression of SELP following

incubation of GB cells with human microglia conditioned medium (CM) (Fig. 1F, G). As the expression of membrane-bound SELP was low in GB cells grown in traditional 2D cultures (data not shown), we chose to use an in vitro 3D model involving a modified Hanging-Drop method that we previously developed

**Fig. 1 Microglia facilitates the proliferation and migration rate of GB cells and enhances the expression of SELP by GB cells. A** Iba1 immunostaining showing activated microglia in GB tumors compared to normal/adjacent tissue in a GB patient FFPE sample and three GB mouse models; iRFP-labeled human U251, patient-derived (PD-GB4) GB xenografts and mCherry-labeled murine GL261, $N = 3$ mice or 3 patient samples. Scale bars represent 100 μm. The proliferation (**B**) and migration (**C**) rates of iRFP-labeled PD-GB4 GB cells were enhanced in the presence of human microglia. Data represent mean ± s.d. of triplicate wells. The graphs are representative of three independent repeats. Statistical significance was determined using an unpaired, two-sided Student's *t* test. **D** Cytokine profile showing over-secretion of SELP and other factors in a co-culture of human PD-GB4 GB cells and primary human microglia compared to monocultures. Data represent mean. Duplicates were measured for each cytokine. The graph shows the average of two independent studies including internal repeats. ELISA (**E**) and Real-time PCR (**F**) results showing over-secretion and mRNA expression of SELP by PD-GB4 cells when treated with human microglia CM (MG CM) compared to naïve microglia medium. Data represent mean ± s.d. Each dot represents a triplicate. The graphs show the average of three independent studies. Statistical significance was determined using an unpaired, two-sided Student's *t* test. Representative image (**G**) and quantification (**H**) of flow cytometry analysis showing over expression of SELP when PD-GB4 spheroids were treated with microglia CM compared to naïve microglia medium. Data represent mean ± s.d. The graph shows the average of three independent studies. Statistical significance was determined using an unpaired, two-sided Student's *t* test. **I** SELP and PSGL-1 are highly expressed in tumor areas enriched with activated microglia in GB mouse models. $N = 3$ mice per staining. Scale bars represent 100 μm. **J** SELP is highly expressed in short-term survivors (STS) GB patient FFPE samples compared to long-term survivors (LTS) or normal human brain tissue. Data represent mean ± s.d. Each dot represents the average of three images per sample. $N = 5$ human samples per group. Scale bars represent 100 μm. Statistical significance was determined using one-way ANOVA test with multiple comparisons adjustment. Source data are provided as a Source Data file.

to create GB spheroids followed by Matrigel seeding[17]. SELP was found to be expressed by PD-GB4 tumor spheroids, and its level of expression was increased when the spheroids were treated with microglia CM as demonstrated by flow cytometry (Fig. 1H, I). These findings were also observed using human U251 GB cells (Supplementary Fig. 2F–I). Furthermore, the results with human U251, patient-derived xenografts (PDX) and GL261 mouse models revealed positive staining of SELP and PSGL-1 in tumor areas enriched with activated microglia (Fig. 1I). This was performed using two additional PDX mouse models (Supplementary Fig. 3A, B). We note that SELP staining correlated mainly with the labeled GB cells while PSGL-1 staining seemed to correlate with Iba1 staining. Furthermore, we co-stained for Ki-67 together with Iba1 to validate in vivo the effects of microglia on GB cell proliferation observed in vitro, and found high expression of Ki-67 in areas enriched with Iba1 (Supplementary Fig. 3C). As SELP is known to be expressed by activated tumor endothelial cells, we co-stained for SELP and CD31 in two human and one murine GB mouse samples (Supplementary 3D). As expected, we also found positive staining of SELP on the tumor blood vessels. Since we used anti-mouse SELP antibody, human xenografts showed staining which correlates only with CD31 staining. This is in contrast to the murine model which showed positive staining that correlated with both blood vessels and tumor cells. These findings together with the staining presented in Fig. 1I indicate that both the tumor cells and their associated endothelial cells express SELP in vivo. In order to further validate the expression and clinical relevance of SELP, we obtained FFPE samples from short-term GB survivors (STS, <2 months) and long-term GB survivors (LTS, >5 years), as well as normal, healthy human brain tissues. Immunostaining of these samples, demonstrated higher expression of SELP in STS tumor samples compared to LTS tumors or normal brain samples (Fig. 1J). Separate channels of the immune-fluorescence staining, H&E staining of STS and LTS samples and flow-Cytometry gating strategy are presented in Supplementary Fig. 3. These results indicate that SELP is involved in the interaction between GB cells and microglia.

The observation that SELP is expressed by GB cells in vitro and in vivo, and overexpressed in the presence of microglia, prompted us to further investigate the functional role of SELP in GB progression, and specifically in GB cell-microglia (tumor-host) interactions. To that end, we established SELP knockdown GB cells using retroviral infection of SELP-shRNA (Fig. 2A–C). We mono-cultured or co-cultured GB cells with microglia and followed GB cell proliferation and migration in 2D in vitro models. SELP knockdown GB cells (shSELP) exhibited reduced

proliferation and migration compared to negative control shRNA infected (shNC) and control WT cells in the presence of microglia (Fig. 2D, E). Interestingly, in the absence of microglia, these differences in proliferation were significantly reduced (Fig. 2D), indicating the relevance of SELP in GB-microglia interactions. The results show that shSELP GB cell proliferation in a co-culture is similar (no significant difference) to that of the WT cell proliferation in mono-culture, and that shSELP cell proliferation in the co-culture is similar to shSELP proliferation in mono-culture. This suggests that SELP knockdown eliminates the differences observed between mono-culture and co-cultures (Fig. 1B), hence inhibiting the microglia-enhanced GB cell proliferation. Since we observed high expression of SELP in our 3D model (Fig. 1G, H), we co-cultured GB cells and microglia in 3D spheroids. While the invasion of GB cells into the Matrigel was increased in the presence of microglia, SELP knockdown significantly inhibited PD-GB4 3D spheroid invasion (Fig. 2F). To confirm that this effect was due to SELP knockdown, we treated GB-microglia spheroids with a SELP specific small-molecule inhibitor (SELPi). SELPi treatment of GB cells in the presence of microglia resulted in a similar reduction in cell invasion (Fig. 2F), indicating that SELP is an important mediator of GB cell invasion. Inhibition of spheroid invasion following SELP knockdown was also observed using human U251 GB cells, and the effect of SELPi on GB spheroids was observed using human U251, PD-GB1, PD-GB3, and murine GL261 GB models (Supplementary Fig. 5B–F). In order to evaluate whether the observed effects of SELPi are not due to cytotoxicity, we have stained the treated cells for apoptosis-necrosis (Supplementary Fig. 6). The results showed slight increase in the percentage of apoptotic cells only when GB cells were treated with high concentrations of 5 and 10 μM SELPi, while no toxicity was observed using 0.5 μM which is the working concentration used for all in vitro experiments. Thus, showing that SELPi delayed the proliferation and invasion of GB cells in the presence of microglia and did not induce direct cell-killing. Since GBs are highly heterogeneous tumors, consisting of several molecular subtypes, we used three lenti-induced murine GB cells which represent the mesenchymal, proneural and classical GB subtypes (Supplementary Fig. 7). We found that the mesenchymal, proneural and classical GB spheroids express high levels of SELP compared to 2D cultures and even higher levels when co-cultured with murine microglia (Supplementary Fig. 7B, D, F). Moreover, treatment with SELPi significantly reduced the invasion of GB spheroids co-cultured with murine microglia using all three GB subtypes (Supplementary Fig. 7C, E, G). Our results showed that SELP

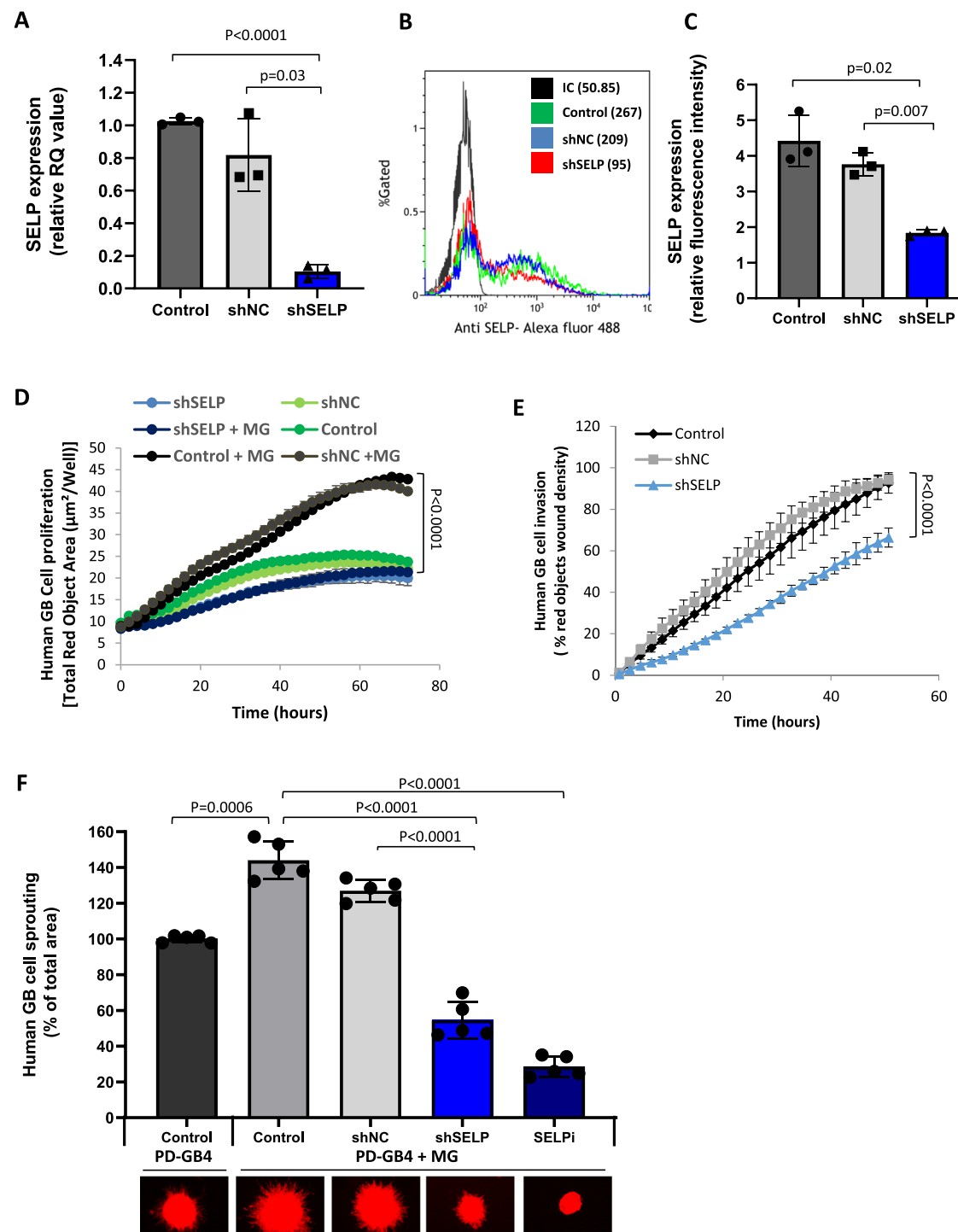

mediates GB cell proliferation and invasion in the presence of microglia in 2D and 3D in vitro models using several human and murine GB models.

**GB cells promote the anti-inflammatory/pro-tumorigenic function of microglia/macrophages via SELP-PSGL-1 axis.** When we investigated the effect of SELP on microglia, we found that not only GB cells, but also microglia, express SELP-mRNA and secrete SELP. Both expression and secretion of SELP were elevated when human microglia were treated with GB CM (Fig. 3A, B). Moreover, human microglia upregulated the expression of PSGL-1

when exposed to GB CM (Fig. 3C, D). Indeed, further analysis of single-cell RNA-seq data, obtained by Darmanis et al., of microglia isolated from fresh patient-derived GB samples[19] (Supplementary Fig. 8A) revealed high expression of PSGL-1 in a population defined as resident microglia (Fig. 3E).

To evaluate the effect of SELP on microglia phenotype, we followed the expression of arginase 1 (ARG1) which is related to immunosuppressive phenotype of tumor-associated macrophages (TAMs)[20]. Furthermore, we evaluated the expression of inducible nitric oxide synthase (iNOS) which is involved in nitric oxide production and is associated with pro-inflammatory phenotype[21], in addition to the assessment of the immunosuppressive cytokines

**Fig. 2 SELP mediates GB cell invasion, migration, and proliferation. A** Real-time PCR showing reduced SELP mRNA levels in shSELP PD-GB4 cells, compared to control WT (Control), and negative control shRNA (shNC). Data represent mean ± s.d. Each dot represents a triplicate. The graph showing the average of three independent experiments. Statistical significance was determined using one-way ANOVA test with multiple comparisons adjustment. Representative image (**B**) and quantification (**C**) of flow cytometry analysis of SELP expression showing reduced expression in shSELP PD-GB4 compared to control WT and shNC PD-GB4 cells. Data represent mean ± s.d. The graph shows the average of three independent experiments. Statistical significance was determined using one-way ANOVA test with multiple comparisons adjustment. **D** Proliferation of iRFP-labeled PD-GB4 cells alone or co-cultured with unlabeled human microglia showing lower proliferation rate of shSELP PD-GB4 compared to control WT and shNC PD-GB4 cells. Cell proliferation was followed and analyzed using the IncuCyte imaging system for 72 h. Data represent mean ± s.d. of triplicate wells. The graph a is representative of three independent repeats. Statistical significance was determined using two-ways ANOVA test with multiple comparisons adjustment. **E** Wound healing assay using iRFP-labeled PD-GB4 cells co-cultured with microglia, showing lower migration rate of shSELP PD-GB4 compared to control WT and shNC PD-GB4 cells. Wound closure was followed and analyzed by the IncuCyte imaging system for 60 h. Data represent mean ± s.d. of 4 wells per group. The graph is a representative of three independent repeats. Statistical significance was determined using two-ways ANOVA test with multiple comparisons adjustment. **F** 3D spheroid invasion in Matrigel showing enhanced invasion of iRFP-labeled PD-GB4 GB cells when co-culture with unlabeled human microglia, and reduced invasion following SELP knockdown or treatment with 0.5 μM SELPi compared to untreated control WT or shNC PD-GB4 cells. Sprouting was followed for 72 h. Data represent mean ± s.d. of 5 spheroids per group. The graph a is representative of three independent experiments. Scale bars represent 100 μm. Statistical significance was determined using one-way ANOVA test with multiple comparisons adjustment. Source data are provided as a Source Data file.

IL-10 and TGF-β (Fig. 3F–I). We found that treatment with recombinant SELP (rSELP) resulted in increased expression of ARG-1, IL-10 and TGF-β and reduced expression of iNOS. We were able to rescue the effects of rSELP by adding SELPi or anti-PSGL-1 neutralizing antibody on top of rSELP treatment. Interestingly, only partial effect, if any, was observed when neutralizing antibodies against the additional SELP ligands CD44 and CD24 were added[22,23] (Fig. 3F–I). Phagocytosis and NO release are functional characteristics of microglia/macrophages associated with pro-inflammatory activation state[8]. A phagocytosis assay with fluorescently-labeled latex beads, demonstrated a reduction in phagocytic activity of human microglia when treated with rSELP (Fig. 3J). Adding SELPi or anti-PSGL-1 neutralizing antibody restored the phagocytic function of human microglia while neutralizing antibodies for CD44 or CD24 did not affect this ability. Finally, we were also able to show that treatment with rSELP reduced the amount of NO released by human microglia following LPS induction. NO levels were restored when we added SELPi or anti-PSGL-1 neutralizing antibody but not when adding anti-CD44 or CD24 neutralizing antibodies (Fig. 3K). These results suggest that SELP mediates the anti-inflammatory phenotype of microglia by binding PSGL-1 and not other SELP ligands. The secretion of IL-10 and TGF-β was also evaluated in the protein level using cytokine array, showing higher secretion by human microglia when treated with rSELP (Supplementary Fig. 8B, C). The cell surface proteins CD163 and CD206 (MRC1), are associated with anti-inflammatory function of microglia/macrophages[24]. Thus, we evaluated their expression by human microglia using flow cytometry. In the presence of PD-GB4 CM, human microglia expressed higher levels of CD163 than untreated cells. This CD163 expression was reduced when microglia cells were treated with PD-GB4 CM supplemented with SELP neutralizing antibody (Supplementary Fig. 8E, F). Treating human microglia with PD-GB4 CM and rSELP induced the expression of CD206 by human microglia while adding SELPi to PD-GB4 CM reduced CD206 expression (Supplementary Fig. 8G). Accordingly, microglia isolated from control WT GB-microglia spheroids expressed higher levels of CD163 than microglia isolated from spheroids prepared with shSELP PD-GB4 cells (Supplementary Fig. 8H–J). To reveal the downstream effects of SELP-PSGL-1 axis in microglia, we have explored the activation of the NF-kB pathway. We found an increase in the phosphorylation of the NF-kB protein p65 when human microglia were treated with rSELP compared to untreated cells (Supplementary Fig. 8K). Validating the effects of SELP on microglia immunophenotype, we used freshly-isolated primary murine microglia in addition to human

microglia. We showed that SELP-PSGL-1 axis mediates the expression of IL-10 and TGF-β by murine microglia and that treatment with rSELP increased the secretion of IL-10 and reduced the phagocytic ability of murine microglia (Supplementary Fig. 9). In order to elucidate whether these findings are restricted to GB-microglia interactions, or apply also to GB interactions with macrophages, we used freshly-isolated bone marrow derived macrophages (BMDM; Supplementary Fig. 10A). First, we found that BMDM also express PSGL-1. This expression was elevated when the cells were treated with rSELP and GB CM, and reduced when SELPi was added to GB CM (Supplementary Fig. 10B). Then, we observed increased CD206 expression and decreased CD38 expression when BMDM were treated with rSELP. CD38 is known to induce pro-inflammatory cytokines secretion by TAMs[25]. Following the addition of SELPi to rSELP containing medium, the expression of CD206 was reduced while the expression of CD38 was elevated (Supplementary Fig. 10C, D). In addition, we found that SELP-PSGL-1 interactions mediate BMDM phagocytosis activity and nitric oxide release. These results show that SELP-PSGL-1 axis is involved in GB-macrophages interactions as well. The association between PSGL-1 and microglia/macrophages was also explored using the GBM datasets in TCGA. The results of the analysis showed a positive correlation between the expression of microglia gene signature—Tmem119, Cx3cr1, P2ry12, Arg1, Olfml3, Gpr34, Ccl4 and Ccl3, and PSGL-1 expression in GB patients (Supplementary Fig. 11A). Interestingly, we also found a positive correlation between PSGL-1 and CD4 (Supplementary Fig. 11B); and, following analysis of single-cell RNA-seq data obtained from the Single Cell Portal[26], we found that PSGL-1 was expressed not only by microglia/macrophages, but also by infiltrating T cells in GB samples (Supplementary Fig. 11C). This suggests that the SELP-PSGL-1 axis might also play a role in the interactions with infiltrating T cells.

Taken together, these results demonstrate the immunosuppressive effect of SELP on microglia/macrophages which is mediated by PSGL-1 in contrast to other SELP ligands.

**SELP inhibition delays tumor growth and affects the microenvironmental landscape in GB mouse models.** In order to evaluate the effect of SELP inhibition in vivo, we performed several experiments using murine and human GB mouse models. First, we evaluated the effect of SELP knockdown in human GB mouse models, we injected shSELP, shNC or control WT PD-GB4 or U251 cells, intracranially into SCID mice (Fig. 4). Mice bearing shSELP tumors exhibited delayed tumor growth and prolonged survival (Fig. 4A, B, D, E). Ki-67 immunostaining

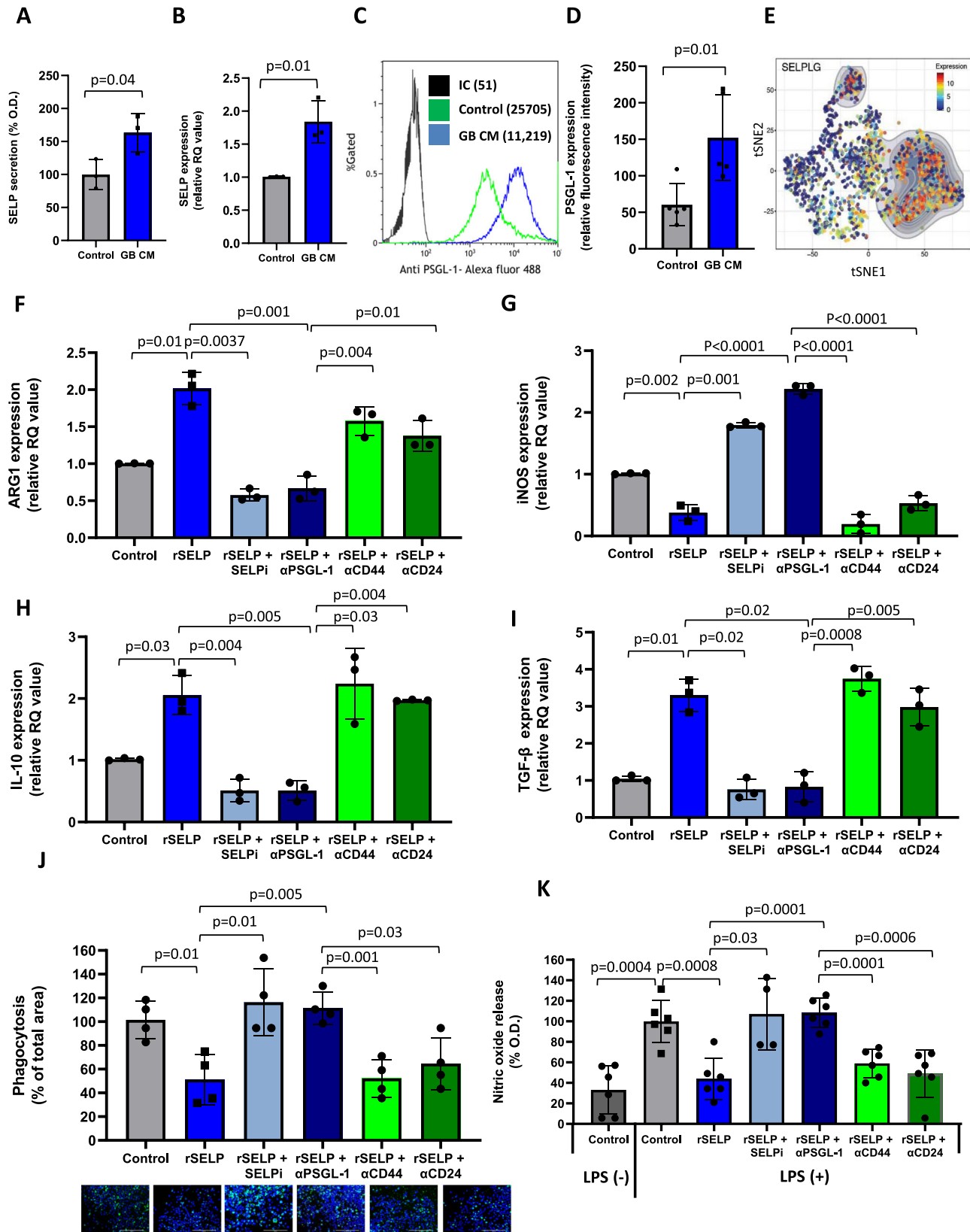

showed reduced proliferation of GB cells in shSELP tumors and a low density of activated microglia and blood vessels in shSELP tumors was also observed by Iba1 and CD31 immunostaining, respectively (Fig. 4C, F).

Investigating the effect of SELP knockdown on the adaptive immune response, we generated shSELP murine GL261 cells. Out

of five different shRNA sequences, the shSELP plasmid which showed the highest silencing effect (approximately 90%) was chosen (Supplementary Fig. 12A, B). Control WT, shSELP, and shNC GL261 cells, were injected intracranially into immuno-competent mice. Tumors were visible by MRI imaging only since day 9 post tumor cell inoculation and mice were imaged 2–3

**Fig. 3 SELP inhibits the microglia pro-inflammatory phenotype and promotes their immunosuppressive activity. A** ELISA assay showing higher secretion of SELP by human microglia treated with PD-GB CM compared to naïve DMEM. Data represent mean ± s.d. Each dot represents a triplicate. The graph shows the average of three independent repeats. Statistical significance was determined using an unpaired, two-sided Student's $t$ test. **B** Real-time PCR showing higher expression SELP mRNA by human microglia treated with PD-GB4 CM compared to naïve DMEM. Data represent mean ± s.d. Each dot represents a triplicate. The graph shows the average of three independent repeats. Statistical significance was determined using an unpaired, two-sided Student's $t$ test. Representative image (**C**) and quantification (**D**) of flow cytometry analysis of PSGL-1 expression by human microglia, showing higher expression when treated with PD-GB4 GB CM compared to naïve DMEM. Data represent mean ± s.d. The graph shows the average of five independent experiments. Statistical significance was determined using an unpaired, two-sided Student's $t$ test. **E** tSNE plot of single-cell RNA-seq analysis of microglia and macrophages isolated from patients' GB tumors, showing the expression of PSGL-1 by microglia compared to macrophages. Data obtained from Darmanis et al.[56]. Treatment with rSELP resulted in elevated mRNA expression of ARG1 (**F**), reduced expression of iNOS (**G**) and elevated expression of IL-10 (**H**) and TGF-β (**I**) by human microglia. SELPi or anti-PSGL-1 neutralizing antibody rescued rSELP effects while anti-CD44 or CD24 neutralizing antibody did not affect gene expression. Data represent mean ± s.d. Each dot represents a triplicate. The graphs show the average of three independent studies. Statistical significance was determined using one-way ANOVA test with multiple comparisons adjustment. Phagocytic activity (**J**) and NO release (**K**) by human microglia were reduced when treated with rSELP, and were restored when inhibiting SELP or PSGL-1 but not when neutralizing CD44 or CD24. Data represent mean ± s.d. of four wells per group. The graphs are representative of three independent repeats. Scale bars represent 200 μm. Statistical significance was determined using one-way ANOVA test with multiple comparisons adjustment. Source data are provided as a Source Data file.

times per week. Reduction in tumor growth and prolonged survival were observed in the shSELP group (Fig. 5A, B). Immunostaining showed reduced proliferation (Ki-67), increased apoptosis (caspase-3), and a lower density of blood vessels (CD31) in shSELP tumors (Fig. 5C). In addition, flow cytometry and immunostaining analysis of the tumors revealed a higher percentage of CD8+ and lower percentage of CD4+/FOXP3+ T cells in shSELP tumors (Supplementary Fig. 12C, D). We note that the growth curve shows only time points in which all the mice were still alive. However, shSELP exhibited delayed tumor growth beyond these time points (Supplementary Fig. 12E).

To further characterize the influence of SELP-PSGL-1 interactions on GB cells and the brain microenvironment, we performed single-cell RNA-seq of shNC and shSELP GL261 tumors focusing on three main subpopulations: tumor cells, T cells and microglia/macrophages (Fig. 6). We identified multiple clusters and were able to annotate most of them to known cell types with sufficient representation of our populations of interest (Fig. 6A). We analyzed the most upregulated genes in each cluster and their distribution between the groups (Supplementary Fig. 13). Deeper examination of the cancer-cell population revealed several tumor cell clusters that were differentially distributed between shNC and shSELP tumors (Fig. 6B, C). Within these clusters and following differential expression analysis we found genes related to cancer invasion such as Col3a1, Map4k4, Chd7 and Tubb2b[27–29], genes associated with proliferation and tumor progression such as Pdgfa and the oncogenes Fos, Jun and Myc[30–33], and angiogenesis genes as VEGFa, Tcf4, Timp1 and Timp3[34–37] (Fig. 6D). Thus, we next applied unbiased, published signatures, representing these three processes on our data[38–40]. We found all three signatures related to cancer invasion, proliferation, and angiogenesis to be enriched in the shNC group (Fig. 6D). To further confirm the annotation of the tumors clusters in addition to signature projection and markers, we used inferred CNV analysis as previously described[41] (Supplementary Fig. 14C). Within the brain microenvironment representing cell populations, the microglia/macrophages subclusters showed the most uneven internal distribution between shNC and shSELP cells (Supplementary Fig. 15). Thus, to get a better resolution of these changes, we further increased the number of cells analyzed by generating additional single-cell RNA-seq analysis using only CD11b positive cells (Fig. 6E–I). Our focused single-cell RNA-seq data of freshly-isolated CD11b+ sorted microglia/macrophages from shSELP and shNC GL261 tumors, identified twelve sub-populations, termed clusters 0–11, that all expressed microglia/macrophages-related genes (Fig. 6E–G). The three clusters (0, 1, and 4), which represent a large portion of the total

population, appeared to have a different distribution of cells when the cells were isolated from SELP knockdown GB or control tumors. For example, clusters 0 and 1 had more cells in shSELP tumors and cluster 4 had more cells in control samples (Fig. 6F, G). Interestingly, GB-derived microglia/macrophages expressed a gene signature that strongly resembled the pro-inflammatory microglia signature (including genes such as Spp1, Il1b, and Cxcl9), described by Krasemann et al. in microglia derived from models of amyotrophic lateral sclerosis (ALS), multiple sclerosis (MS), and Alzheimer's disease (AD)[42]. More importantly, perturbation of the SELP-PSGL-1 interactions, caused the microglia/macrophages cells to express an even higher score of this neurodegenerative signature (Fig. 6H). Another observation was that GB-derived microglia/macrophages expressed a variety of genes associated with antigen presentation, such as H2-Eb1, H2-Ab1, H2-Aa, and H2-DMb1, immune cell recruiting factors such as Ccl16, Ccl5, Cxcl9, and Cxcl10, as well as other immune ligands such as, Tnf, Icoslg, Hebp1, Trf, Adam17, Pros1, Grn, IL-1a, CD86, Clar, Edn1 and Sema4d were also enriched after manipulation of the SELP-PSGL-1 axis (Fig. 6I). Signature enrichment algorithms demonstrated significant difference for both the neurodegenerative microglia signature score ($p$ value $= 2.2 \times 10^{-16}$, Wilcoxon rank sum test and $p$ value $= 4.4 \times 10^{-9}$ CERNO test, see Methods) and the antigen presentation and chemokine signature scores ($p$ value $= 8.969 \times 10^{-16}$, Wilcoxon rank sum test and $p$ value $= 6.4 \times 10^{-8}$ CERNO test, see Methods). Thus, as for T cells, it appears that common mechanisms are shared between microglia in autoimmunity-related inflammation and cancer immunity and this inflammatory environment can be further enhanced by perturbation of the SELP-PSGL-1 axis. Specific genes and signatures identifying the microglia/macrophages populations are presented in Supplementary Fig. 16A. In addition, we found reduced expression of PSGL-1 by microglia isolated from shSELP tumors compared to shNC (Supplementary Fig. 16B) showing the reciprocal effect of SELP knockdown in GB cells and its influence on the surrounding microglia. Of note, examination of the T cell representing clusters, showed similar cluster distribution between the shNC and shSELP groups (Supplementary Fig. 17A–D). However, we did find several differentially expressed genes which represents T cell activation that were up-regulated in the shSELP group such as granzyme B, perforin, PD-1 and LAG-3 (Supplementary Fig. 17E).

The changes in antigen presentation and T cell recruitment signatures as well as the higher percentage of CD8+ T cell infiltration in the shSELP tumors and the minor differences found in the single-cell analysis, raise the question of the dependency of the observed in vivo effect on peripheral immunity. However, our

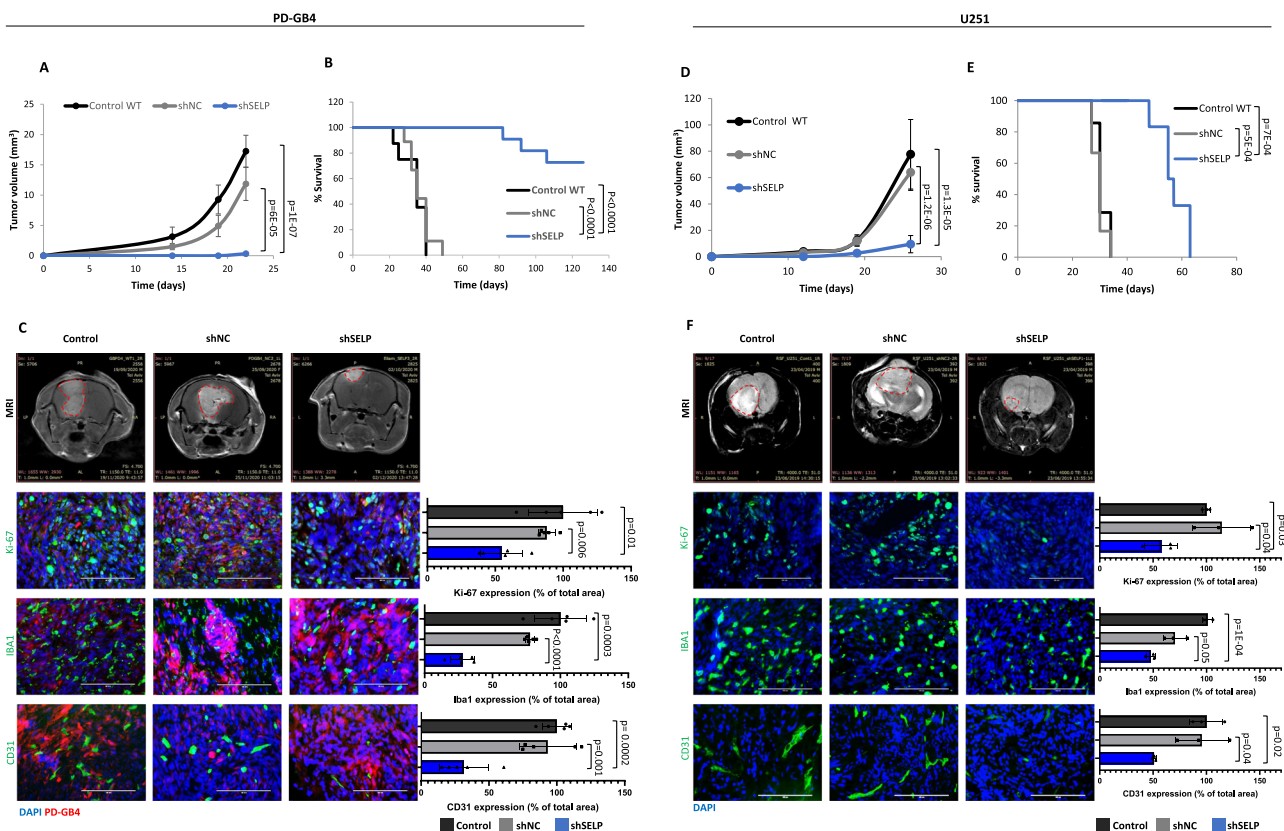

**Fig. 4 SELP-knockdown inhibits tumor growth and prolongs survival in human GB mouse models. A** SELP knockdown reduced tumor growth rate of PD-GB4 tumors in mice compared to control WT (control) or shNC GB tumors. Tumor volume was calculated using Radiant software. Data represent mean ± s.e.m. $N = 8$ control, 9 shNC, and 14 shSELP. One-way ANOVA, Dunn's method, $p < 0.001$. **B** Kaplan–Meier curve showing prolonged survival of shSELP PD-GB4 tumor-bearing mice compared to control WT and shNC. $N = 8$ control, 9 shNC, and 11 shSELP. P values were determined using two sided log rank test. **C** Representative T1 weighted MRI images of PD-GB4 tumors following Gd-DTPA administration, detected at day 22 post tumor inoculation. Representative images and quantification of immunostaining for proliferating cells (Ki-67), activated microglia (Iba1), and blood vessels (CD31) in the tumors showing reduced proliferation and blood vessel density, and activated microglia in shSELP tumors. Data represent mean ± s.d. $N = 5$ images per mouse. The graphs show data from a representative mouse per group out of two mice per group. Statistical significance was determined using one-way ANOVA test with multiple comparisons adjustment. Scale bars represent 100 μm. **D** Tumor growth of control WT (control), shNC, and shSELP U251 tumors in SCID mice demonstrating delayed tumor growth in shSELP U251 tumors. Tumors were detected by MRI imaging (MR solutions, T2 weighted). Tumor volume was calculated using Radiant software. Data represent mean ± s.e.m. $N = 9$ control, 8 shNC, and 9 shSELP. One-way ANOVA, Dunn's method, $p < 0.001$. **E** SELP knockdown prolonged the survival of U251 tumor-bearing mice compared to control WT and shNC. $N = 7$ control, 6 shNC, and 6 shSELP. P values were determined using two-sided log rank test. **F** Representative T2 weighted MRI images of U251 tumors detected at day 19 post tumor inoculation. Representative images and quantification of immunostaining for proliferating cells (Ki-67), activated microglia (Iba1), and blood vessels (CD31) in the tumors showing reduced proliferation and blood vessel density, and activated microglia in shSELP tumors. Data represent mean ± s.d. Each dot represents the average of three images per mouse. $N = 3$ mice per group. Statistical significance was determined using one-way ANOVA test with multiple comparisons adjustment. Scale bars represent 100 μm. Source data are provided as a Source Data file.

results using human GB mouse models show that the inhibitory effect of SELP knockdown is independent of T cells as these studies were performed using immunocompromised mice lacking the adaptive immune system. We note that caspase staining increased in shSELP tumors only in immunocompetent mice (Fig. 5C). This phenomenon could be correlated with enhanced infiltration of CD8+ T cells (Supplementary Fig. 12C, D) suggesting that these cells play a cytotoxic role in shSELP GB. Thus, the T cells may have partial contribution to the observed effects.

Finally, we evaluated the therapeutic potential of SELP inhibition for GB treatment. To this end, we treated GL261 and the lenti-induced, mesenchymal iAGR53 murine GB-bearing mice IV with SELPi. The results showed a significant reduction in tumor volume evaluated by MRI imaging in both models and prolonged survival in the iAGR53 model (Fig. 7A, C, D). In addition, immunostaining analysis of GL261 tumors displayed a

clear reduction in proliferation and blood vessel density (Fig. 7B). We also found an increase of CD8+ T cells, and decrease in CD4+/FOXP3 + T cells in the tumors treated with SELPi (Supplementary Fig. 18). These results demonstrate the effect of SELP inhibition, not only on GB cells, but also on the brain microenvironment.

In our human GB mouse models, SELPi treatment of iRFP-labeled U251 cells, reduced the fluorescence signal detected by the CRI-Maestro™ imaging system (Fig. 7E). Immunostaining of these tumors revealed reduced invasion and proliferation of GB cells in the treated group, as shown by H&E and Ki-67 staining, respectively (Fig. 7F). In addition, we observed a reduction of blood vessel density (CD31) and microglia activation (Iba1) in the treated group (Fig. 7F).

In order to increase the signal obtained following IV treatment of SELPi, we used brain cannulas to inject SELPi intraventricularly into animals in the GB PDX mouse model (Fig. 7G). When

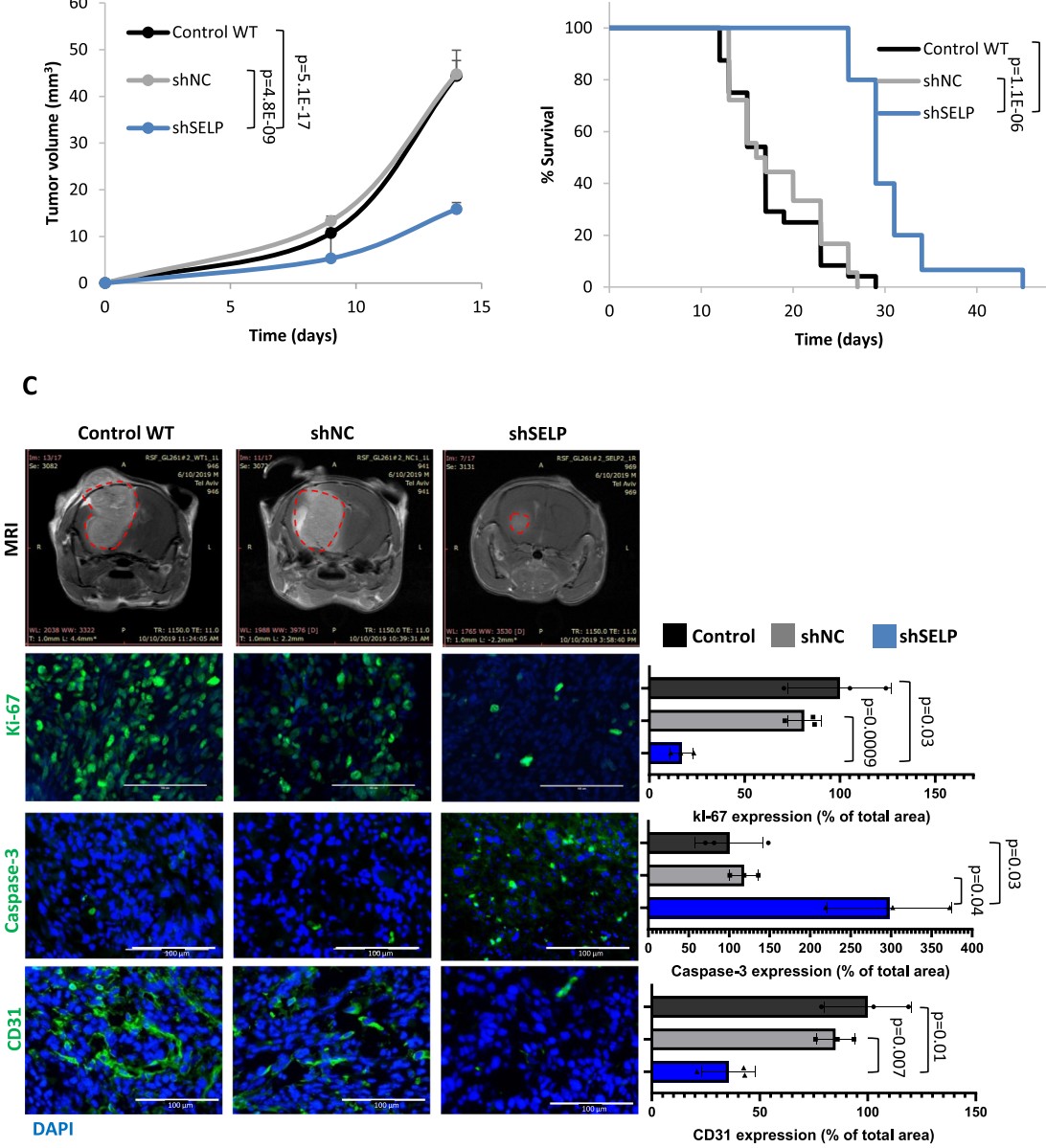

**Fig. 5 SELP-knockdown inhibits tumor growth and prolongs survival in a murine GB mouse model. A** Tumor growth of control WT (control), shNC, and shSELP GL261 tumors in C57BL/6 mice demonstrating delayed tumor growth in the shSELP group. Tumors were detected by MRI imaging (MR solutions, T1 weighted). Tumor volume was calculated using Radiant software. Data represent mean ± s.e.m. $N = 29$, two independent experiments. One-way ANOVA, Dunn's method, $P < 0.001$. **B** Kaplan–Meier curve showing prolonged survival of shSELP GL261 tumor bearing mice compared to control WT and shNC. $N = 24$ Control, 18 shNC, 15 shSELP, two independent experiments. $p$ values were determined using two-sided log rank test. **C** Representative T1 weighted MRI images of GL261 tumors following Gd-DTPA administration, detected at day 19 post tumor inoculation. Representative images and quantification of immunostaining for proliferative cells (Ki-67), apoptotic cells (caspase-3) and blood vessels (CD31). The results demonstrate reduced proliferation, blood vessel density and increased apoptosis in shSELP tumors. Data represent mean ± s.d. Each dot represents the average of three images per mouse. $N = 3$ mice per group. Statistical significance was determined using one-way ANOVA test with multiple comparisons adjustment. Scale bars represent 100 μm. Source data are provided as a Source Data file.

we removed the cannulas 21 days post injection, MRI imaging revealed a marked 67% reduction in tumor volume. We observed reduced invasion and proliferation of GB cells and reduction in Iba1 and CD31 density in SELPi treated PD-GB4 tumors (Fig. 7H). As for validation that the observed effect on in vivo tumorigenesis is specifically due to SELP inhibition, we assessed for potential adverse effects following systemic administration of SELPi (Supplementary Fig. 19). Following single IV dose of 16 mg/kg injection of SELPi into healthy mice, blood chemistry results showed no significant changes between the saline, vehicle

and SELPi groups (Supplementary Fig. 19A). Complete blood count revealed an increase in blood lymphocytes, which correlates with the results showing here and the previously published results by others on the effect of SELP on T cell trafficking[43] (Supplementary Fig. 19B). Detailed characterization of the small molecule SELPi is shown in Supplementary Fig. 19C as provided by the manufacturer website.

In conclusion, our findings highlight the relevance of SELP in GB cell–microglia/macrophages interactions and demonstrate its effect on GB progression in several in vitro and in vivo human

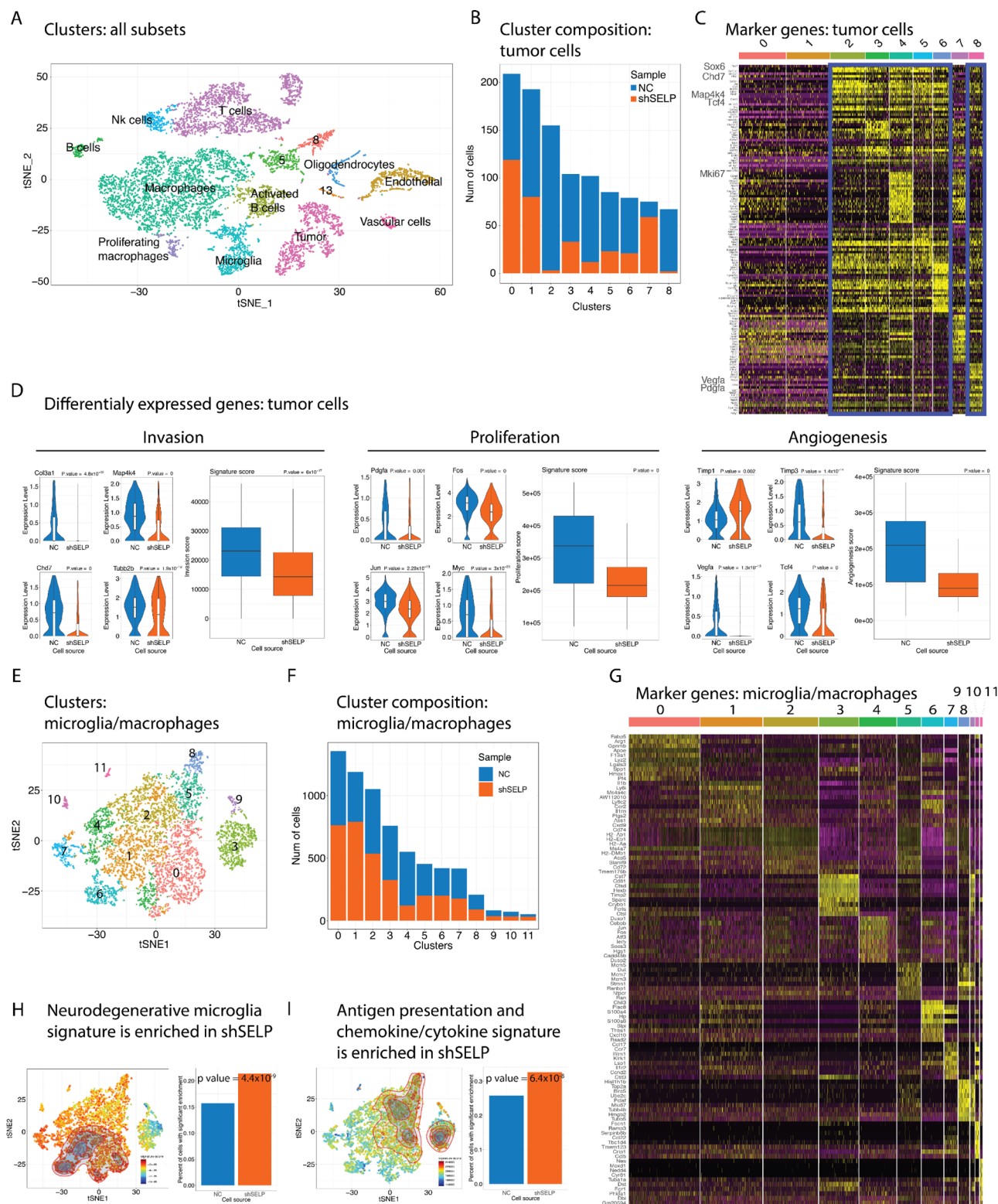

and murine models as well as clinical samples. Our hypothesis is based on the presented data demonstrating the mechanisms by which SELP mediates GAMs activation state and GB tumorigenesis (Fig. 8).

## Discussion

One major characteristic of GB is its diffusive nature, which stems from the ability of GB cells to invade the brain parenchyma and spread through the brain tissue. This constitutes one of the biggest challenges in the treatment of GB patients[44]. Another characteristic of GB is the immunosuppressive microenvironment associated mainly with the high prevalence of immunosuppressive GAMs in the tumor[45]. The invasiveness of GB cells and the immunosuppressive microenvironment may well be linked since GAMs have been shown to facilitate GB invasion, growth, and angiogenesis[10,12].

Here, we established the role of SELP in GB cell invasion and in the determination of microglia/macrophages activation state.

**Fig. 6 Perturbation of the SELP-PSGL-1 interactions causes tumor cells to exhibit reduced tumorigenesis and the microglia/macrophages cells to express higher score of neurodegenerative and antigen presentation signatures. A** Unsupervised clustering of the single-cell RNA-seq profiles of different cell types populations of interest identified in GL261 tumors. **B** Clusters distribution between the shNC (NC) and shSELP groups within the tumor cell population. **C** Heat map showing the most up-regulated differentially expressed genes in each cluster within the tumor cells population. **D** Representative genes from the tumor cells clusters and comparison of gene signature scores showing down-regulation of the invasion, proliferation, and angiogenesis signatures in shSELP tumors compared to shNC. Center of the box plots shows median values, boxes extent from 25% to the 75% percentile, whiskers show minimum and maximum values. Statistical significance was determined using an unpaired, two-sided Student's $t$ test. **E** Unsupervised clustering of the single-cell RNA profiles of microglia/macrophage cells. **F** Bar graphs showing the number of cells present in each cluster from microglia/macrophages isolated from shNC *versus* shSELP GL261 tumors. **G** Heat map showing the ten most up-regulated differentially expressed genes in each cluster within microglia/macrophages population. **H** Projection of a neurodegenerative-associated microglia signature[42] and bar plots showing up-regulation in shSELP tumors compared to shNC ($p$ value $= 2.2 \times 10^{-16}$, two-sided Wilcoxon rank sum test and $p$ value $= 4.4 \times 10^{-9}$ CERNO enrichment test, see "Methods"). **I** Projection of an antigen presentation and chemokines/cytokines microglia signature and bar plots showing up-regulation in shSELP tumors compared to shNC ($p$ value $= 8.969 \times 10^{-16}$, two-sided Wilcoxon rank sum test and $p$ value $= 6.4 \times 10^{-8}$ CERNO enrichment test, see "Methods"). The contour marks the region of highly scored cells by taking into account only cells with a signature score above the 10th percentile.

Our results indicate that SELP is expressed by GB cells and microglia/macrophages and that GB cell expression of SELP mRNA and the secretion of soluble SELP (sSELP) were increased in the presence of microglia CM. Interestingly, the expression of membrane bound SELP was higher when GB cells were grown in 3D spheroids seeded in Matrigel than in 2D cultures in a petri dish made of rigid plastic. This expression was further elevated when GB spheroids were treated with microglia CM or when microglia were incorporated into the spheroids. This phenomenon did not occur when cells were grown on 2D. Using our multicellular 3D in vitro models, we have shown that both molecular and pharmacological inhibition of SELP resulted in obstruction of GB cell invasion into Matrigel. This effect was demonstrated using human GB cell lines as well as several PD-GB cells and four murine GB cell lines which represent the diverse GB molecular subtypes (mesenchymal, proneural and classical), out of which 3 were lenti-induced genetically-engineered cells. These findings suggest that SELP plays an important role in GB invasion and can be captured in vitro using setting that better recapitulates the brain microenvironment. Further findings indicated that SELP mediates GB invasion and proliferation also in vivo. It is important to note, that our 3D spheroids model using a modified Hanging Drop method, does not include stem cell medium and does not enrich for cancer stem cells as other tumor spheres assays, but rather creates differentiated tumor spheroids which maintain the cell population heterogeneity. Thus, the differences observed between 2D and 3D models are not due to differences between differentiated GB cells and glioma stem cells, but due to the 3D structure, ECM-mimicking environment and the presence of the tumor microenvironment. Evaluating the effect of SELP on microglia function, we found that the reciprocal effect of GB and microglia cells also leads to an increase in the expression of SELP mRNA in microglia and the secretion of sSELP from them, as well as the expression of membrane-bound PSGL-1 in response to GB CM. These results suggest a positive feedback loop which leads to overexpression of SELP and PSGL-1 by GB and microglia, following exposure to sSELP and possibly other secreted factors. Single cell RNA-seq showed a reduction in PSGL-1 expression by microglia isolated from shSELP tumors. Moreover, our results using BMDM showed elevated PSGL-1 expression in response to rSELP or GB CM and reduction in PSGL-1 expression when SELPi was added to GB CM. These findings support the positive feedback loop hypothesis of SELP and PSGL-1 expression. However, further experiments are required in order to fully reveal this mechanism. One question that still remains open is whether SELP directly mediates GB invasion, or whether, by blocking its expression, we prevent microglia/macrophages from facilitating GB invasion by different mechanisms. Evidence from the literature suggests that

SELP expression is downstream to the NF-κB pathway, which has been shown to be involved in GB progression[46–49]. Furthermore, NF-kB pathway was shown to mediate tumor-promoting phenotype of TAMs[50]. Indeed, we found an increase in the phosphorylation of the NF-kB subunit protein p65 in human microglia when exposed to rSELP. These findings may suggest that NF-kB pathway in downstream to PSGL-1 signaling, which then promotes GAMs to support tumor progression and induce the expression of SELP in the suggested positive feedback loop mechanism. SELP binding to PSGL-1 has been shown to facilitate neuroblastoma growth through the activation of Src and ERK-1[51]. These findings point towards possible mechanisms for SELP-mediated GB invasion. Studying the correlation between the SELP-PSGL-1 axis and cancer-promoting transcription factors may reveal important pathways in GB progression that could be affected by the contribution of microglia to this axis.

Importantly, we observed that GB cells not only expressed SELP on the cell membrane but also secreted the soluble protein (sSELP). Moreover, SELP had an immunosuppressive effect on microglia/macrophages in the presence and in the absence of GB cells when grown in culture. This indicates that SELP-PSGL-1 axis may be activated in microglia/macrophages by membrane-bound SELP and sSELP, and that both forms may have a role in microglia/macrophages phenotype not only in the context of GB. Indeed, it was shown that sSELP can be found in both the monomeric and the dimeric forms in human plasma[52] and that both forms can bind PSGL-1 expressed on neutrophils and monocytes[53]. Although the exact role of sSELP has yet to be elucidated, increased plasma levels of sSELP are known to be associated with certain types of cancer[54]. However, neither its role nor its effect was shown in the brain, in general, or in GB, in particular, until now. This raises the question as to whether the soluble and membrane forms of SELP function in the same way in GB progression. One speculation could be that one form, SELP, contributes to GB invasion due to its adhesion properties, while the other, sSELP, contributes to microglia/macrophages suppression due to its higher exposure to the neighboring stromal cells.

The anti-inflammatory/pro-tumorigenic activation of microglia/macrophages is characterized by reduced levels of iNOS expression and NO release, impaired phagocytic abilities which are important for their anti-tumor response[55], and elevated levels of ARG1, CD163, CD206 and a number of cytokines including IL-10 and TGF-β[10,56]. In the context of GB, GAMs have been shown to secrete TGF-β, which facilitates growth, invasion, and angiogenesis[57]. Our results indicate that treating microglia/macrophages with rSELP reduces their phagocytic activity, decreases the expression of iNOS and the release of NO, and increases the expression of anti-inflammatory markers and cytokines (such as IL-10 and TGF-β) by microglia/macrophages. In parallel,

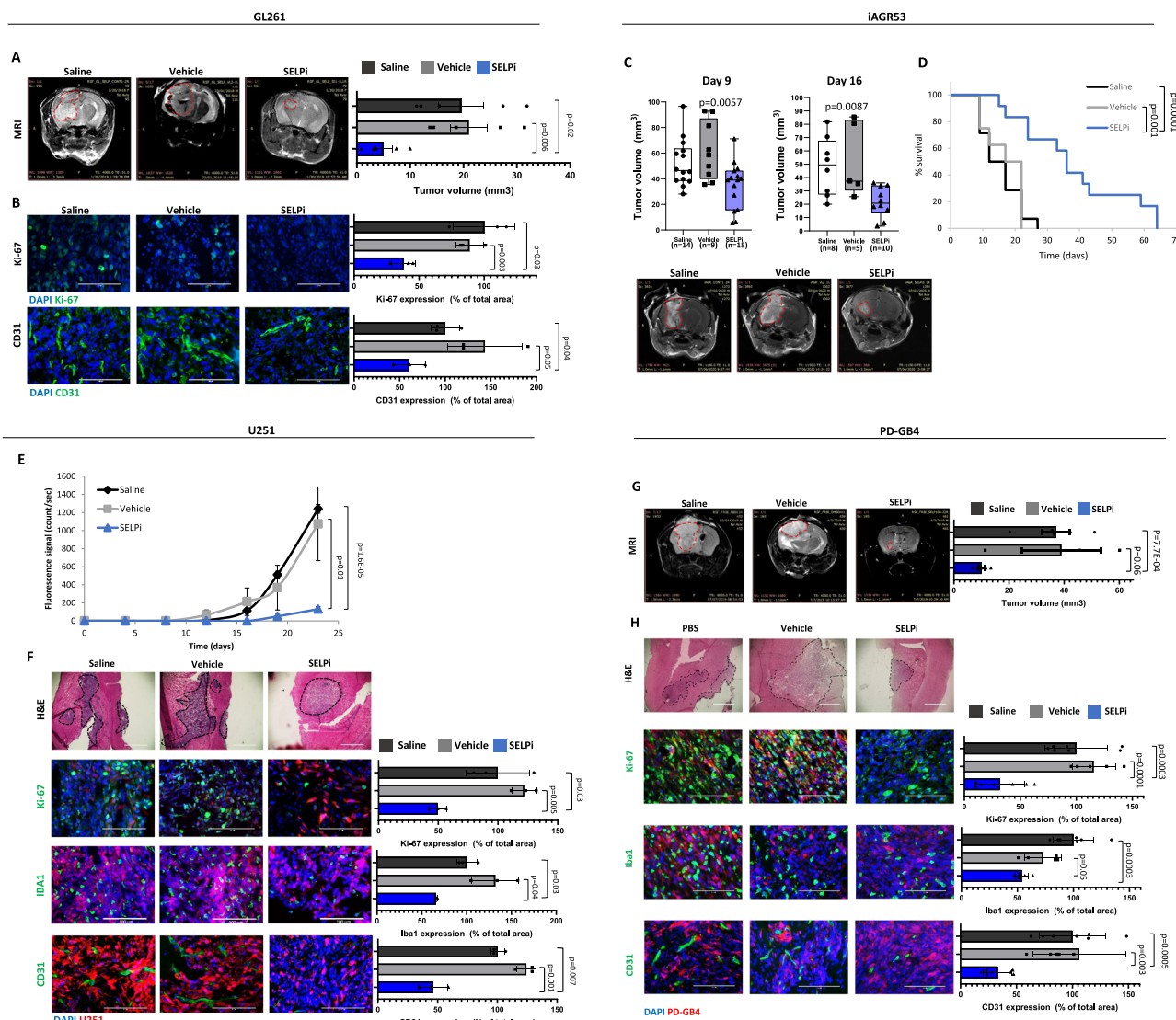

**Fig. 7 SELPi treatment delays the growth of murine and human GB tumors in mice. A** Treatment with SELPi reduced tumor growth of GL261 GB tumors in mice. Representative MRI scanning of day 17 post tumor inoculation. Tumor volume was calculated using Radiant software. $N = 5$ mice per group. Data represent mean ± s.e.m. One-way ANOVA, Holm-Sidak's method, $p < 0.009$. **B** SELPi treatment reduced the proliferation (Ki-67) and microvessel density (CD31) in GL261 GB tumors. Data represent mean ± s.d. Each dot represents the average of three images per mouse. $N = 3$ mice per group. **C** Systemic SELPi treatment reduced tumor growth of iAGR53 tumors in mice. Representative images of day 16 scan. $P$ value was calculated by One-way ANOVA test with multiple comparisons adjustment. Tumor volume was calculated using Radiant software. "N" refers to number of mice per group. Center of the box plots shows median values, boxes extent from 25% to the 75% percentile, whiskers show minimum and maximum values. **D** SELPi treatment prolonged the survival of iAGR53 tumor-bearing mice. $N = 15$ saline, 8 vehicle, and 12 SELPi (mice per group). $P$ values were determined using two-sided log rank test. **E** SELPi treatment delayed tumor growth of U251 tumors by SELPi treatment. Data represent mean ± s.e.m. $N = 3$ saline, 4 vehicle, and 4 SELPi (mice per group). One-way ANOVA, Dunn's method, $p < 0.006$. **F** H&E staining of U251 tumors. Immunostaining demonstrating reduction in proliferation (Ki-67), activated microglia (Iba1), and blood vessels (CD31) in SELPi treated tumors. Data represent mean ± s.d. Each dot represents the average of three images per mouse. $N = 3$ mice per group. **G** Tumor volume of PD-GB4 tumors in mice demonstrating delayed growth of PD-GB4 tumors by local SELPi treatment. Representative MRI scanning of day 21 post tumor inoculation. Data represent mean ± s.e.m. Tumor volume was calculated using Radiant software. $N = 5$ PBS, 3 DMSO, and 5 SELPi. One-way ANOVA, Dunn's method, $p < 0.018$. **H** H&E staining of PD-GB4 tumors. Immunostaining demonstrating reduction in proliferation (Ki-67), activated microglia (Iba1), and blood vessels (CD31) in SELPi treated tumors. Data represent mean ± s.d. $N = 3$–4 images per mouse, 2 mice per group. Statistical significance of all the immunostaining was determined using one-way ANOVA test with multiple comparisons adjustment. All Scale bars represent 100 μm. Source data are provided as a Source Data file.

inhibiting SELP or PSGL-1 using SELPi or anti-PSGL-1 neutralizing antibody, rescued rSELP effects. Since SELP can bind not only to PSGL-1 but to other ligands as well, we used anti-CD44 and anti-CD24 neutralizing antibodies. Inhibiting CD44 or CD24 had only partial effect on microglia/macrophages phenotype. This shows that SELP binding to PSGL-1 is required for the immunosuppressive effects of SELP on microglia/macrophages, and

that by blocking SELP or PSGL-1, we are able to enhance both the brain and the systemic immune response against GB tumors.

Although the primary microglia isolated by their CD11b expression showed positive expression of the specific microglia markers TMEM119 and P2Y12, we could not fully discriminate between microglia and macrophages in our models. Thus, we repeated the experiments using freshly-isolated BMDM which

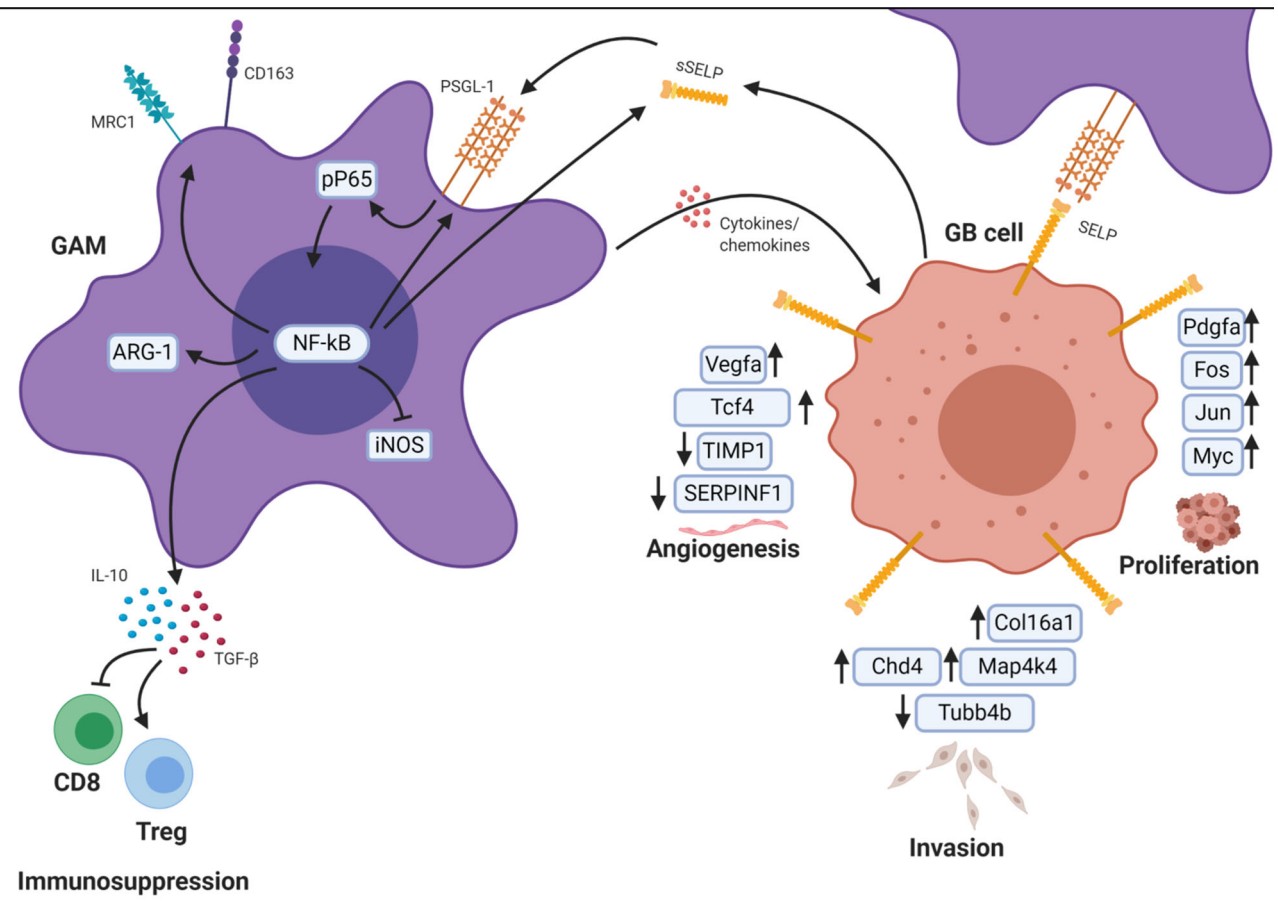

**Fig. 8 Working model.** Illustration showing a proposed mechanism by which SELP-PSGL-1 axis mediates GAMs phenotype and GB progression. This image was created with BioRender.com.

showed similar results. Our analysis of published single-cell RNA-seq data obtained from microglia and macrophages isolated from GB patients revealed that the population which correlates with microglia signature does express higher levels of PSGL-1 compared to the macrophages one. However, these differences in PSGL-1 expression between the two populations seem to be smaller in our single-cell data obtained from murine GL261 GB tumors, than those seen in the human data. This may be due to differences between murine and human microglia/macrophages or in data acquisition and analysis. This may suggest that in human GB tumors, SELP is involved in the regulation of resident microglia to a higher extent than in infiltrating macrophages, although further research will be required in order to fully reveal the different effects of SELP on microglia compared to macrophages. Nonetheless, our results demonstrated the role of SELP in GB interactions with both resident microglia and BMDM.

When looking ahead to examine the feasibility of SELP to become a druggable target for GB, we implanted human or murine shSELP GB cells orthotopically in mice and indeed, it dramatically prolonged survival compared to control WT GB cells. The shSELP U251 and PD-GB4 tumors showed a reduction in microvessel density and Iba1 positive microglia at the tumor site. Furthermore, a low density of blood vessels was also observed in a murine mouse model. In addition, we found an increase in cytotoxic CD8+ T cells and a decrease in Treg cells in shSELP GL261 tumors. The effect on the blood vessels can be explained by the fact that GAMs are known to promote angiogenesis in GB[58] so that changing their phenotype to a more pro-inflammatory state could lead to a reduction in new blood vessel

recruitment and formation. The alteration of GAMs phenotype could also be responsible for the effect on immune infiltration since, as shown here, SELP induces the expression of IL-10 and TGF-β both of which are known to inhibit CD8+ T cells and promote Treg cells[59,60]. Another explanation could relate to the interactions between GB and T cells since PSGL-1 is now considered to be a T cell inhibitory molecule[16]. Indeed, we observed a positive correlation between PSGL-1 and CD4 expression in the TCGA data of GB patients, and a high level of T cell expression of PSGL-1 in published GB RNA-seq data. To further investigate these questions, we have performed single cell RNA-seq analysis using shNC and shSELP GL261 tumors. Since we mixed equal amounts of CD11b-positive cells, TILs and the remaining cell-suspension which contained the cancer cells as well as other brain microenvironment components, this data does not represent the distribution of the different cell types in GL261 tumors. However, it allowed us to observe the differentially-expressed genes and the distribution of sub-clusters in each cell type. We first observed that tumor cell-representing clusters were found to be enriched for cancer cell invasion, cancer proliferation and angiogenesis signatures in the shNC group. These direct and indirect changes in the cancer cell populations may be induced by blocking the binding of the membrane-bound SELP expressed on the cancer cells to PSGL-1 expressed on the microglia/macrophages. Another explanation may be that the polarization state change of the GAMs phenotype resulted in reduced secretion of tumor-promoting factors. Looking at the T cell population and clusters, we did not see large differences in their genes expression. However, we did observe a mild increase in T cell activation factors

such as granzyme B and perforin in T cells isolated from shSELP tumors. These data together with the FACS and staining analysis of the tumors indicate that T cells may have a partial contribution to the observed effects of SELP inhibition on tumor progression. Looking at the microglia/macrophages population, we observed larger and more meaningful changes in the clusters distribution between the shNC and shSELP groups. Thus, we performed an additional single-cell RNA-seq analysis using only CD11b-positive cells, isolated from shNC and shSELP GL261 tumors. These single cell RNA-seq results demonstrated that SELP knockdown in the cancer cells resulted in alteration of the microglia/macrophage. Interestingly, we found our dataset to be enriched for a microglial signature that was previously defined as belonging to dysfunctional microglia, found in neurodegenerative disorders such as ALS and AD[42]. This signature was associated with pro-inflammatory microglia that contribute to neurodegeneration. As neurodegenerative diseases and GB are known to have an inverse relationship[61], this microglial sub-population may have anti-tumor potential. Indeed, It was reported that PSGL-1 levels in microglia were reduced in EAE model of multiple sclerosis, and reduced plasma levels of SELP and L-Selectin were detected in AD patients[62,63]. However, the role and the mechanisms in which SELP is involved in neurodegenerative and autoimmune disorders remain unknown. Our findings in cancer may contribute to elucidate the role of microglia in neurodegenerative diseases. In addition, we identified a signature of antigen presentation and T cell recruitment chemokines that was enriched in shSELP tumors. These results suggest that by interfering with SELP-mediated GB-microglia interactions, we may be able to improve the local and systemic immune response against the tumor. Although increased T cell infiltration and activation, and caspase staining were also observed, our results with human GB xenografts in immunodeficient mice indicated that the effect of SELP inhibition on tumor growth is not T cell-dependent. Thus, a combination of SELP inhibition with current checkpoint inhibitors may further improve the therapeutic outcomes. To examine the therapeutic potential of SELP inhibition, we treated four different GB mouse models using SELPi. In all models used, inhibiting SELP resulted in smaller tumors as detected by MRI or by the CRI Maestro™ imaging system, and prolonged survival was observed using iAGR53 GB model. Immunostaining of SELPi-treated U251, PD-GB4 and GL261 tumors revealed similar effects on GB cell proliferation and the brain microenvironment as was observed using shSELP cells. Our results demonstrate the diverse possible benefits of using SELPi for GB therapy.

Other members of the selectin family also bind PSGL-1. Although SELP binds to PSGL-1 at higher affinity than E-Selectin and L-Selectin[64], their expression may serve as a compensation mechanism for PSGL-1 activation. In addition, it is plausible that there is some redundancy between the different Selectins. However, the shRNA used in this research was designed specifically to silence SELP, and SELPi was previously shown by the manufacturer to be selective for SELP with no effect observed neither on E-Selectin nor on L-Selectin (https://www.tocris.com/products/kf-38789_2748). Thus, we show that specific inhibition of SELP using both molecular and pharmacological agents delayed tumor progression. Nonetheless, further research is required in order to discriminate the possibility of compensation by other Selectins while looking into the added value of inhibiting all three, P, E and L selectins, in order to halt GB's aggressive progression.

As we show and discuss the different roles of SELP in mediating immune function in pathological and normal conditions, inhibition of SELP may affect the balance of immune maintenance. Adverse effects may present autoimmunity disorders, as PSGL-1 was shown to mediate T cell activation, or impaired

immune-trafficking. As SELP mediates platelet activation as well, its inhibition may impair platelet function in homeostasis and pathological conditions. However, several studies have identified SELP as therapeutic target for several pathological conditions such as cardiovascular diseases and sickle-cell anemia. Furthermore, several clinical trials have shown safety, and now evaluating the efficacy of anti-SELP monoclonal antibodies and recombinant PSGL-1 in phase II trials for the above mentioned conditions (NCT03814746, NCT03474965 NCT01245634, NCT00876902). This demonstrates the therapeutic potential and safety of anti-SELP treatments. In our studies in mice, SELPi did not present any changes in behavior, blood chemistry or body weight in healthy mice at the maximal tolerated dose (MTD), and maximal dose was chosen according to solubility limitations. Since inhibition of SELP did not cause direct tumor-killing, combining anti-SELP treatment with the standard chemotherapy (TMZ) may be beneficial. Radiation therapy was shown to increase SELP expression on endothelial cells on the lumen of angiogenic vessels[65], which may affect the phenotype of recruited BMDM. Thus, SELP inhibition may also improve the efficacy of radiotherapy. Moreover, as we show that SELP mediates the secretion of T cells suppressing cytokines, and that its inhibition facilitates the infiltration of CD8 positive T cells into the tumor, anti-SELP treatment may improve the susceptibility of GB tumors to existing immunotherapies, hence sensitizing this non-responsive tumor to become immune checkpoint therapy (ICT)-responsive.

Taken together, our results show that SELP plays an important role in GB progression and associated immunosuppressive effects in the brain microenvironment. In addition, we used advanced in vitro and ex vivo techniques, as well as pre-clinical human and murine GB mouse models to demonstrate that blocking SELP has a powerful anti-tumorigenic effect on all these parameters. Using three lenti-induced murine GB cells representing the mesenchymal, proneural and classical GB subtypes, as well as human cell lines and patient-derived GB cells, we showed the relevance of SELP-mediated GB-GAMs interactions to the clinical settings. These findings may improve our understanding of GB tumorigenesis and have the potential to lead to the development of advanced and effective strategies for GB therapy.

## Methods

**Materials**. DMEM, fetal bovine serum (FBS), L-glutamine, penicillin, streptomycin, mycoplasma detection kit, EZ-RNA II total RNA isolation kit and fibronectin (1 mg/ml, dilution: 1:100) were purchased from Biological Industries Ltd. (Kibbutz Beit HaEmek, Israel). Percoll medium (Cat. No. p4937) and all other chemical reagents, including salts and solvents, were purchased from Sigma-Aldrich (Rehovot, Israel). Milli-Q water was prepared using a Millipore water purification system. Amicon Ultra Centrifugal Filters; molecular weight cut-off (MWCO) 5 or 3 kDa and Poly-L-Lysine (PLL) (Cat. No. A-005-C; 0.1 mg/ml) were purchased from Merck Millipore (Burlington, Massachusetts, USA). The qScript™ cDNA Synthesis Kit was purchased from Quantabio (Beverly, MA, USA). Fast SYBR™ green Master Mix was purchased from Applied Biosystems (California, USA). Collagenase IV, Dispase II (neutral protease) and DNase I were purchased from Worthington Biochemical Corporation (NJ, USA). RBC lysis solution (Cat. No. 420301) was purchased from BioLegend (San Diego, California, USA). MACS MS magnetic columns for cell separation (Cat. No. 130-042-201), CD11b MicroBeads for cell isolation (Cat. No. 130-093-634) and CD45 (TIL) MicroBeads for cell isolation (Cat. No. 130-110-618) were purchased from Miltenyi Biotec (Bergisch Gladbach, Germany). SELP inhibitor (SELPi) KF38789 (Cat. No. 2748) was purchased from Tocris BioScience (Bristol, United Kingdom). Recombinant human SELP (Cat. No. ADP3; Lot. No. ARL6019071), Recombinant murine SELP (rSELP) (Cat. No. 10094-PS; Lot. No. DKLJ0118111), Human SELP ELISA kit (Cat. No. DPSE00), Total NO/Nitrite/Nitrate Immunoassay (Cat. No. KGE001), Mouse XL Cytokine Array Kit (Cat. No. ARY028), Human Cytokine Array kit (Cat. No. ARY005B), anti-human PSGL-1 neutralizing antibody (Cat. No. MAB3345; Lot. No. CLYK0120111; Clone 688102), and anti-human SELP neutralizing antibody (Cat. No. AF137; Lot. No. FBX0518051) were purchased from R&D Systems (Minneapolis, Minnesota, USA). Human L-507 cytokine array kit (Cat. No. AAH-BLM-1A-4; Lot. No. 102920 009) was purchased from RayBiotech (Norcross, Georgia, United States). Anti-human/mouse CD44 neutralizing antibody (Cat. No. NBP2-2530; Lot. No. VC289186) was purchased from Novus (Colorado, USA).

Anti-murine PSGL-1 neutralizing antibody (Cat. No. BE0188; Lot No. 676818M2) was purchased from Bio X Cell (Massachusetts, USA). MEBCYTO Apoptosis Kit, was purchased from MBL International (UK), Recombinant murine GM-CSF (Cat. No. 315-03-50ug; Lot. No. 091855) was purchased from PeproTech (Rehovot, Israel). Latex beads for phagocytosis assays (Cat. No. L4655) were purchased from Sigma-Aldrich (Rehovot, Israel). ProLong® Gold mounting with DAPI (Cat. No. p36935) and Hoechst 33342 (Cat. No. H3570) were purchased from Invitrogen (Carlsbad, California, USA). Mayer's Hematoxylin solution (Cat. No. 05-06002) and Eosin Y solution (Cat. No. 05-10002) were purchased from Bio-Optica (Milano, Italy).

**Plasmids**. mCherry was subcloned by our group into the pQCXIP vector (Clontech, USA) as previously described[66]. iRFP was used as previously described[67]. Human SELP shRNA (Cat. No. sc-29421-SH) and human negative control (NC) shRNA (Cat. No. sc-108060) containing plasmids were purchased from Santa Cruz Biotechnology, Inc. (Dallas, Texas, USA). Murine SELP shRNA and murine NC shRNA plasmids (Simple hairpin shRNAs in the pLKO.1 lentiviral vector designed by The RNAi Consortium) were purchased from GE Healthcare Dharmacon, Inc. (Lafayette, Colorado, USA). Anti-human (SWA11) and anti-murine (Ml.69) CD24 neutralizing antibodies were kindly provided by Nadir Arber and Shiran (Tel Aviv Sourasky Medical Center).

**Primary immunostaining antibodies**. Rabbit anti-mouse/human Iba1 (Cat. No. NBP2-19019; Lot. No. 41556; Dilution: 1:200), rat anti-human/mouse PSGL-1 (Cat. No. NB100-78039; Lot. No. C; Dilution 1:50), rabbit anti-human/mouse Ki-67 (Cat. No. NB500-170; Lot. No. G15; Dilution 1:50), and rabbit anti-mouse FOXP3 (Cat. No. NB600; Lot. No. D-1; Dilution 1:30) were purchased from Novus (Colorado, USA). Mouse anti-human SELP (Cat. No. BBA1; Lot. No. APB081704; Clone BBIG-E; Dilution 1:30) was purchased from R&D Systems (Minneapolis, Minnesota, USA). Mouse anti-mouse SELP (Cat. No. 148302; Lot No. B186735; Clone RMP-1; Dilution 1:50) was purchased from BioLegend (San Diego, California, USA). Rat anti-mouse CD31 (Cat. No. 550272; Lot. No. 6273859; Dilution 1:25) was purchased from BD Biosciences (Franklin Lakes, NJ, USA). Rabbit anti-human/mouse Caspase-3 (Cat. No. CST-9664L; Lot. No. 21; Dilution 1:30) was purchased from Cell Signaling Technology (Danvers, Massachusetts, USA). Rat anti-mouse CD4 (Cat. No. 14-9766-82; Lot. No. 4307664; Clone 4SMAS; Dilution 1:100) and Rat anti-CD8 (Cat. No. 14-0808-82; Lot. No. 2003225; Clone: 4SMIS; Dilution 1:50) were purchased from eBioscience (San Diego, California, USA).

**Secondary immunostaining antibodies**. Goat anti-mouse Alexa Fluor® 647 (Cat. No. ab15115; Lot. No. GR309891-3; Dilution 1:300), goat anti-rabbit Alexa Fluor® 488 (Cat. No. ab150077; Lot No. GR315933-2; Dilution 1:300), and goat anti-rabbit Alexa Fluor® 647 (Cat. No. Ab150079; Lot. No. Gr3176223-2; Dilution 1:300) were purchased from Abcam (Cambridge, United Kingdom). Goat anti-rat Alexa Fluor® 488 (Cat 112-545-068; Lot. No. 143654; Dilution 1:300) and goat anti-rat Alexa Fluor® 647 (Cat. No. 112-605-003; Lot. No. 137652; Dilution 1:300) were purchased from Jackson ImmunoResearch Laboratories, Inc. (West Grove, Pennsylvania, USA).

**Flow cytometry antibodies**. Mouse anti-human SELP (Cat. No. BBA1; Lot. No. APB081704; Clone BBIG-E; Dilution 1:20), Mouse IgG1 isotype control (Cat. No. mab002; Dilution 1:20) were purchased from R&D Systems (Minneapolis, Minnesota, USA). Rat anti-human/mouse PSGL-1 (Cat. No. NB100-78039; Lot. No. C; Dilution 1:24) was purchased from Novus (Colorado, USA). Rabbit anti-human/mouse CD163 (Cat. No. AB182422; Lot. No. GR3232711-5; Dilution 1:20), anti-mouse CD11b (Cat. No. ab8878; Lot. No. GR131048-4; Dilution 1:25), and goat anti-rabbit Alexa Fluor® 488 (Cat. No. ab150077; Lot No. GR315933-2; Dilution 1:50) were purchased from Abcam (Cambridge, United Kingdom). Anti-mouse CD3-FITC (Cat. No. 130-119-798; Lot. No. 5190919162; Clone REA641; Dilution 1:10), anti-mouse CD8-APC (Cat. No. 130-111-712; Lot. No. 5190919051; Clone: REA793; Dilution 1:10), anti-mouse CD4 VioBlue® (Cat. No. 130-118-696; Lot. No. 5190919087; Clone REA605; Dilution 1:10), REA Control-APC (Cat. No. 130-113-446; Lot. No. 5190711317; Clone REA293; Dilution 1:10), REA Control-VioBlue® (Cat. No. 130-113-545; Lot. No. 5190711335; Clone REA293; Dilution 1:10), anti-mouse CD38-APC-Vio770 (Cat. No. 130-125-227; Lot. No. 5200405654; Clone REA616; Dilution 1:50), anti-mouse F4/80-FITC (Cat. No.130-117-509; Lot. No. 5200306686; Clone REA126; Dilution 1:50), anti-CD11b-PE-Vio770 (Cat. No. 130-113-808; Lot No. 5190919070; Clone REA592; Dilution 1:50) and REA Control-FITC (Cat. No. 130-113-449; Lot. No. 5190711318; Clone REA293; Dilution 1:10) were purchased from Miltenyi Biotec (Bergisch Gladbach, Germany). Anti-mouse FOXP3 Alexa Fluor® 647 (Cat. No. 126408; Lot. No. B264076; Clone MF-14; Dilution 1:25), Alexa Fluor® 647 IgG2b, k Isotype ctrl (Cat. No. 400626; Lot. No. B243822; Clone RTK4530; Dilution 1:25), anti-mouse P2Y12-PE (Cat. No. 848003; Lot No. B264216; Clone S16007D; Dilution 1:50), anti-human P2Y12-Briliant Violet 421 (Cat. No. 392105; Lot No. B286137; Clone S16001E; Dilution 1:50), anti-human TMEM119 (Cat No. 853301; Lot No. B272769; Clone A16075D; Dilution 1:50), anti-mouse CD206-PE (Cat. No. 141706; Lot No. B280038; Clone C068C2; Dilution 1:50) and LEAF Purified Rat IgG2b, k Isotype Ctrl (Cat. No. 400621; Lot. No. B209798; Clone RTK4530) were purchased from BioLegend (San Diego, California, USA). Mouse-IgG k BP-CFL 488 (Cat. No. sc-516176; Lot. No. HO118; Dilution 1:20) was purchased from Santa Cruz Biotechnology, Inc. (Dallas, Texas, USA). Goat anti-rat Alexa Fluor® 488 (Cat 112-545-068; Lot. No. 143654; Dilution 1:50) was purchased from Jackson ImmunoResearch Laboratories, Inc. (West Grove, Pennsylvania, USA). anti-human CD206 (Cat No. 60143-1-ig; Dilution 1:100) and anti-human TMEM119 (Cat. No. 27585-1-ap; Dilution 1:100) were purchased from ProteinTech (Rosemont, California, USA).

**Western blot antibodies**. Anti-human total p65 (Cat. No. D14E12; Lot. No. 13; Dilution 1:1000), anti-human phospho p65 (Cat. No. S536; Lot. No. 16; Dilution 1:1000), anti-human vinculine (Cat. No. E1E9V; Lot. No. 6; Dilution 1:1000) and anti-rabbit HRP (Cat. No. 7074P2; Lot. No. 28; Dilution 1:2000) were purchased from Cells Signaling (Massachusetts, USA)

**Cell culture**. U251 human GB cell line was obtained from the European Collection of Authenticated Cell Cultures (ECACC) (Porton Down, Salisbury, UK). GL261 murine GB cell line was obtained from the National Cancer Institute (Frederick, MD, USA). Human embryonic kidney 293T cells (HEK 293T) were obtained from the American Type Culture Collection (ATCC, Manassas, VA, USA). U251 cell line was authenticated by the European Collection of Authenticated Cell Cultures (ECACC) using morphology, karyotyping, PCR-based techniques, and Cytocrome oxidase I assay, following manufacturer validated procedures. The mesenchymal iAGR53, the proneural PNp53 and the classical EGFRviii-shP16 murine GB cell lines were prepared as previously described[68,69]. Cells were cultured in Dulbecco's modified Eagle's medium (DMEM) supplemented with 10% FBS, 100 U/ml Penicillin, 100 μg/ml Streptomycin, 12.5 U/ml Nystatin, 2 mM L-glutamine (Biological Industries, Israel). Primary human microglia were obtained from Celprogen (Torrance, California, USA) and cultured in microglia medium (Celprogen) supplemented with 10% FBS. Cells were routinely tested for mycoplasma contamination with a mycoplasma detection kit (Biological Industries, Israel). All cell lines used were tested negative for mycoplasma contamination. All cells were grown at 37 °C in 5% $CO_2$.

**Human primary GB cells**. Patient-derived GB cells (PD-GB) were isolated from clinical samples obtained from surgical procedures. Tumor tissues were kept in cold PBS and processed within 40 min. In order to isolate tumor cells, tumor samples were dissected to 0.5 mm pieces, and were then plated in 6 cm plates, and cultured in 1 ml DMEM supplemented with 10% FBS, 100 U/ml Penicillin, 100 mg/ml Streptomycin, and 2 mM L-glutamine. Viable cancer cells remained attached to culture plates during medium changes and kept growing in culture, while stromal cells and cell debris were washed away. Cells were routinely tested for mycoplasma contamination with a mycoplasma detection kit (Biological Industries, Israel). All cells were grown at 37 °C in 5% $CO_2$.

**Primary murine brain cell isolation**. Brains resected from healthy, 5–8 week old C57BL/6 mice, were chopped and incubated with 1 mg/ml Collagenase IV, 2 mg/ml Dispase II (neutral protease), and 0.02 U/ml DNase I for 50 min at 37 °C. Red blood cells (RBC) were lysed with RBC lysis solution followed by a Percoll gradient for myelin separation. The resultant cell-suspension was then incubated with CD11b microbeads for microglia or with CD45 microbeads for total leukocytes, and the desired population was isolated on MACS MS magnetic columns. Murine microglia were then seeded on poly-L-Lysine (PLL)-coated plates in microglia medium (ScienCell, California, USA).

**Primary murine bone marrow-derived macrophages (BMDM) isolation**. Bone marrow cells were freshly isolated from the tibias and the femurs of 8–12 weeks old C57BL/6 mice (Envigo CRS, Israel). Bone marrow cells were extracted using 25-gauge syringe and passed through a 70 μm nylon strainer (Corning, Israel). For macrophages differentiation cells were incubated with 50 ng/ml recombinant murine GM-CSF for 1 week. The medium was replaced 4 days post cells isolation. Cells were tested for CD11b and F4/80 expression using flow cytometry.

**Conditioned medium (CM) preparation**. To generate CM, GB cells or microglia were seeded in their complete medium. Forty-eight hours post seeding, medium was replaced with starvation medium (0–2% serum) for an additional 24–48 h after which CM was collected.

**Tumor spheroids**. Tumor spheroids were prepared from mCherry/iRFP/GFP-labeled patient-derived, human U251 or murine GB cells (GL261, iAGR53, PNp53, EGFRviii-shP16), either alone or co-cultured with unlabeled primary human/murine microglia (4:1 ratio). GB spheroids were prepared using our modified hanging-drop method[17], in which drops of cell suspension are held hanging from the bottom of an inverted tissue-culture plate until cells agglomerate spontaneously at the lower part of the drop due to gravity. Cells were deposited in 25 μl droplets on the inner side of a 20 mm dish and incubated for 48 h at 37 °C when the plate is facing upside down to allow for spheroid formation. For assessment of cell invasion, GB spheroids (4,000 cells/sphere) were embedded in Matrigel (Corning), seeded in a 96-well plate, and incubated with complete DMEM or complete

DMEM supplemented with SELPi (0.5 μM). 3D spheroid invasion was visualized after 24–72 h with an EVOS FL Auto cell imaging system (ThermoFisher Scientific, Massachusetts, USA). For FACS analysis, GB spheroids (8,000 cells/sphere) were embedded in Matrigel, seeded in a 12-well plate, and treated with complete DMEM, 0% serum microglia medium or microglia CM for 24–48 h. Cells were then recovered using Cell Recovery Solution (Corning) and further analyzed by FACS.

**Flow cytometry**. For flow cytometry assays, cells were harvested using a cell scraper, and were then washed with PBS followed by additional washes with PBS supplemented with 1% BSA and 5 mM EDTA (FACS buffer). Tumor spheroids were recovered from Matrigel using Cell Recovery Solution (Corning) and washed with FACS buffer. To validate the purity of the isolated microglia population, CD11b isolated cells or human microglia were incubated with TMEM119 and P2Y12-labeled antibodies for 1 h. For human TMEM119, primary antibody was unlabeled and cells were washed and incubated with secondary, Alexa 488 conjugated antibody. To assess P-Selectin and PSGL-1 expression, GB cells and microglia were seeded alone, co-cultured in a 2D petri dish or grown as spheroids seeded in Matrigel. Cells were treated with control medium or CM for 48 h. They were then incubated with mouse anti-human P-Selectin antibody for 1 h on ice, and then were washed, and incubated with Alexa-488 labeled anti-mouse IgG binding protein for 1 h on ice. Alternatively, cells were incubated with rat anti-human PSGL antibody, and then were washed, and incubated with goat anti-rat, Alexa-488 labeled antibody for 1 h on ice. For evaluation of CD163, CD206, CD38, and PSGL1 expression by microglia and BMDM, cells were seeded in 2D petri dishes treated with DMEM 0% serum, PD-GB4 CM, PD-GB4 CM + SELP neutralizing antibody (2 μg/ml) or SELPi or rSELP + SELPi for 48 h. Alternatively, human microglia were grown in spheroids containing either iRFP labeled control WT PD-GB4 cells or shSELP PD-GB4 cells. Cells were then incubated with labeled anti- CD163, CD206, CD38 or with unlabeled PSGL-1 antibody for 1 h on ice, before being washed and incubated with Alexa-488 labeled secondary antibody for PSGL-1 expression for 1 h on ice. Microglia from spheroids were identified by negative iRFP staining. For in vitro cytotoxicity of SELPi, U251 and PD-GB4 GB cells were seeded in 6 well plates ($1 \times 10^5$ cells per well) for 72 h, treated with DMSO ($2.5^{-4}$) or with 0.5, 1, 2, 5, and 10 μM of SELPi. Following 72 h, cells were trypsinized and staining for annexin-FITC and PI was performed as the manufacturers recommended. For immune-infiltration assessment, CD45 cells freshly isolated from GB tumors were divided into two panels: (1) T cell panel- cells were incubated with FITC-labeled anti-CD3, APC-labeled anti-CD8 and Vio-Blue anti-CD4 antibodies; (2) Treg panel- cells were incubated with Vio-Blue anti-CD4 and Alexa-647 anti-FOXP3. Cells were incubated for 1 h on ice. Single stained cells for each antibody and a pool of the corresponding isotype control were used as negative staining controls. Fluorescence intensity was assessed using either an Attune flow cytometer (Life Technologies) or a Gallios$^{TM}$ flow cytometer (Beckman Coulter, USA) and the results were analyzed by Kaluza 2.1 software (Beckman Coulter, USA).

**Gene knockdown and fluorescence-labeling by retroviral infection**. Human embryonic kidney 293T (HEK 293T) cells were co-transfected with the specific expression/knockdown plasmid and the compatible packaging plasmids (pMD.G. VSVG and pGag-pol.gpt). Supernatant containing retroviral particles was collected 48 h post transfection. GB cells or microglia were incubated with the virus for 48 h and positive cells were selected by puromycin resistance for mCherry, shSELP, and shNC infections, and hygromycin for iRFP infection. All retroviral infections showed 70–90% positive infected cells.

**Co-culture proliferation assay**. Primary murine/human microglia were seeded in a 24 well plate (Corning) at different concentrations, and the same amount of mCherry-labeled murine GL261 (1:4, 1.5:1, 3:1, 6:1 ratios), iRFP-labeled patient-derived (PD-GB4) or human U251 GB cells were added 24–72 h post seeding (1:1). The plates were then incubated for 96 h (37 °C; 5% CO₂) in microglia medium and proliferation of fluorescently-labeled cells was measured by the IncuCyte Zoom Live cell analysis system (Essen Bioscience). GB cells in microglia medium were used as a control.

**Wound healing assay**. iRFP-labeled human GB cells, untreated or shRNA-infected, were seeded in a 96-well plate, either alone or with primary human microglia (5:1 ratio) at 100% confluence, and were treated with naïve microglia medium (not exposed to cells) or microglia CM. Twenty-four hours post-seeding, a wound maker (Essen Biosciences) was used to create a uniformed wound to the monolayer and then the healing process was monitored and analyzed over the next 48–96 h with the IncuCyte Zoom live-cell analysis system.

**Transwell migration assay**. Murine GL261, human U251 or PD-GB cells ($1 \times 10^5$ cells) were seeded in 8 μm inserts (Costar Inc., USA) coated with fibronectin in 0% serum DMEM. After 2 h, once the cells had attached, inserts were transferred to a fresh 24-well plates with either microglia medium alone or with the addition of $1 \times 10^5$ human or murine microglia seeded in microglia medium. For evaluation of human microglia migration, primary human microglia ($1 \times 10^5$ cells) were seeded in 8 μm inserts (Costar Inc., USA) coated with fibronectin in 0% serum DMEM. Following cell attachment, inserts were transferred to fresh 24 well plates with either 0% serum DMEM or untreated GB cells seeded in 0% serum DMEM. The cells were allowed to migrate for 5–20 h, before fixation and staining (Hema 3 Stain System; Fisher Diagnostics, USA). The stained migrated cells were imaged using an EVOS FL Auto cell imaging system (ThermoFisher Scientific). The numbers of migrated cells per membrane were evaluated in the captured images by ImageJ 1.52v software.

**Cytokine array**. Cytokine secretion following GB-microglia interactions in murine GB model was assessed by culturing murine GL261 GB cells ($2 \times 10^4$ cells/ml) and primary murine microglia ($7 \times 10^4$ cells/ml), either alone or together, in microglia medium for 72 h. Medium was then collected, concentrated x20 (Amicon Ultra Centrifugal filters), and analyzed for cytokine secretion by a murine Cytokine Array (R&D). Cytokine secretion following GB-microglia interactions in human GB model was assessed by culturing human PD-GB4 GB cells ($1 \times 10^5$ cells/ml) and primary human microglia ($6 \times 10^4$ cells/ml), either alone or together, in microglia medium for 72 h. Medium was then collected, concentrated x20 (Amicon Ultra Centrifugal filters), and analyzed for cytokine secretion by a human Cytokine Array (RayBiotech). Cytokines secreted by microglia were detected by incubating primary human/murine microglia ($1 \times 10^5$ cells/ml) in PLL-coated plates for 24 h before treatment with 0% serum DMEM or 0% serum DMEM supplemented with rSELP (3.5 μg/ml) for 24 h. The medium was then collected and concentrated x20 (Amicon Ultra Centrifugal filters). Secreted cytokines were detected by Protein Profiler-Cytokine Array. Membranes were visualized by a myECL imager or the iBright imaging system (ThermoFisher Scientific). Cytokine levels were quantified using ImageJ 1.52v software.

**ELISA**. PD-GB cells, U251 cells and primary human microglia ($1 \times 10^5$ cells/ml) were seeded separately in PLL-coated plates and treated with each other's CM or naïve medium for 24 h. The medium was replaced with fresh medium for an additional 24 h, and was then collected and concentrated (Amicon Ultra Centrifugal filters). SELP levels in the medium were detected by a SELP ELISA kit. Optical density was measured by a SpectraMax M5e multi-detection microplate reader system (Molecular Devices, California, USA).

**Nitric oxide secretion evaluation**. Primary human microglia ($1 \times 10^5$ cells/ml) were seeded in PLL-coated plates. Twenty-four hours post-seeding, cells were treated with: 0% serum DMEM, 0% serum DMEM supplemented rSELP (1.5 μg/ml), 0% serum DMEM exposed to rSELP and SELPi (0.5 μM), 0% serum DMEM exposed to rSELP and anti-PSGL-1 neutralizing antibody (2.5 μg/ml), 0% serum DMEM exposed to rSELP and anti-CD44 neutralizing antibody (2.5 μg/ml), or 0% serum DMEM exposed to rSELP and anti-CD24 neutralizing antibody (2.5 μg/ml) for 24 h. Alternatively, macrophages were seeded in 96 well plates ($5 \times 10^5$ cells/ml) treated with macrophages medium, macrophages medium supplemented with rSELP (3.5 μg/ml), macrophages medium supplemented with rSELP and SELPi (0.5 μM), macrophages medium supplemented with rSELP and anti-PSGL-1 neutralizing antibody (2.5 μg/ml), macrophages medium supplemented with rSELP and anti-CD44 neutralizing antibody (2.5 μg/ml) or macrophages medium supplemented with rSELP and anti-CD24 neutralizing antibody (2.5 μg/ml) for 24 h. Medium was then collected for evaluation of nitrite levels by the Total Nitric Oxide and Nitrate/Nitrite Parameter Assay Kit. Optical density was measured by a SpectraMax M5e multi-detection microplate reader system (Molecular Devices).

**Phagocytosis assay**. Primary human/murine microglia were seeded in 96 well plates ($1 \times 10^5$ cells/ml) in microglia medium, microglia medium supplemented with rSELP (1.5 μg/ml for human microglia and 3.5 μg/ml for murine microglia), microglia medium supplemented with rSELP and SELPi (0.5 μM), microglia medium supplemented with rSELP and anti-PSGL-1 neutralizing antibody (2.5 μg/ml), microglia medium supplemented with rSELP and anti-CD44 neutralizing antibody (2.5 μg/ml) or microglia medium supplemented with rSELP and anti-CD24 neutralizing antibody (2.5 μg/ml) for 48 h. The medium was then replaced by fresh microglia medium containing green fluorescence latex beads for 2–6h and washed with PBS supplemented with Hoechst staining solution for 10 min. For evaluation of murine macrophages phagocytosis abilities, macrophages were seeded in 96 well plates ($5 \times 10^5$ cells/ml), treated with macrophages medium, macrophages medium supplemented with rSELP (3.5 μg/ml), macrophages medium supplemented with rSELP and SELPi (0.5 μM), macrophages medium supplemented with rSELP and anti-PSGL-1 neutralizing antibody (2.5 μg/ml), macrophages medium supplemented with rSELP and anti-CD44 neutralizing antibody (2.5 μg/ml) or macrophages medium supplemented with rSELP and anti-CD24 neutralizing antibody (2.5 μg/ml) for 48 h. Cells were incubated with latex beads for 3.5 h and washed with PBS supplemented with Hoechst staining solution for 10 min. Phagocytosis was visualized using the EVOS FL Auto cell imaging system (ThermoFisher Scientific) and analyzed by ImageJ 1.52v software.

**Gene expression analysis**. SELP mRNA expression was evaluated using qPCR. PD- GB4 cells, U251 cells and primary human microglia ($1 \times 10^5$ cells/ml) were seeded separately and treated with each other's CM or naïve medium for 48 h.

For qPCR evaluation of SELP silencing, GB cells were seeded in complete DMEM ($2 \times 10^5$ cells) for 72 h.

To evaluate microglial expression of IL-10 TGF-β, ARG1 and iNOS by primary human microglia ($1 \times 10^6$ cells), cells were seeded in PLL-coated plates. Twenty-four hours post-seeding cells were treated with: 0% serum DMEM alone, or exposed to rSELP (1.5 μg/ml), or exposed to rSELP and SELPi (0.5 μM), or exposed to rSELP and anti-PSGL-1 neutralizing antibody (2.5 μg/ml), or exposed to rSELP and anti-CD44 neutralizing antibody (2.5 μg/ml), or exposed to rSELP and anti-CD24 neutralizing antibody (2.5 μg/ml) for 48 h.

To evaluate microglial expression of IL-10 and TGF-β by primary murine microglia, cells were freshly isolated and seeded ($1 \times 10^6$ cells) in PLL-coated plates for 4–6 days. Then, cells were treated with: 2% serum-supplemented microglia medium alone, or exposed to rSELP (3.5 μg/ml), or exposed to rSELP and SELPi (0.5 μM), or exposed to rSELP and anti-PSGL-1 neutralizing antibody (2.5 μg/ml), or exposed to rSELP and anti-CD44 neutralizing antibody (2.5 μg/ml), or exposed to rSELP and anti-CD24 neutralizing antibody (2.5 μg/ml) for 48 h.

**RNA isolation**. EZ-RNA II total RNA isolation kit (Biological Industries Ltd., Israel) was used to isolate total RNA, according to the manufacturer's protocol. Briefly, samples were lysed with 0.5 ml Denaturing Solution/10 cm² culture plate. Water saturated phenol was then added, and the samples were centrifuged. Iso-propanol was added to precipitate the RNA and the centrifuged RNA pellet was washed with 75% ethanol, centrifuged, and re-suspended with ultra-pure double distilled water. RNA concentration was evaluated using a NanoDrop® ND-1000 Spectrophotometer according to the manufacturer's V3.5 User's Manual (Nano-Drop Technologies, Wilmington, DE).

**cDNA synthesis**. qScript™ cDNA synthesis kit for RT-PCR was used to synthesize cDNA, according to the manufacturer's protocol. Briefly, 1 μg of total RNA sample was mixed with qScript Reverse Transcriptase, dNTPs, and nuclease free water. The reaction tube was then incubated at 42 °C for 30 min and heated at 85 °C for 5 min to stop cDNA synthesis.

**Real-time PCR**. The expression level of target genes was assessed by SYBR green real-time PCR (StepOne plus, Life Technologies) and normalized to GAPDH housekeeping gene. Primers used were:

Human SELP: forward- 5′- TCCTTGAGAGCGTTTCAGTATG -′3, reverse -5′- CTGTCCACTGTCCCAAGTTAT C -′3

Murine SELP: forward- 5′- GAC TTTGAGCTACTGGGATCTG -3′, reverse -5′- CAG GAA GTG ATG TTA TGC CTT TG -3′

Human IL-10: forward- 5′- CGCATGTGAACTCCCTGG -3′, reverse- 5′ -TAG ATGCCTTTCTCTTGGAGC -3′

Human TGF-β: forward- 5′- GCCTTTCCTGCTTCTCATGG -3′, reverse- 5′ -GTACATTGACTTCCGCAAGGA -3′

Human/ murine GAPDH: forward- 5′- ATTCCACCCATGGCAAATTC –3′, reverse -5′- GGATCTCGCTCCTGGAAGATG –3′

Human iNOS: forward- 5′- CGCCTTTGCTCATGACATTG -′3, reverse -5′- TCAAACGTCTCACAGGCTG C -′3

Human ARG1: forward- 5′- AGGTCTGTGGGAAAAGCAAG -′3, reverse -5′- GCCAGAGATTCCAATTG -′3

Murine IL-10: forward- 5′- TGAATTCCCTGGGTGAGAAGC -3′, reverse- 5′- CACCTTGGTCTTGGAGCTTATT -3′

Murine TGF-β: forward- 5′- ACTGGAGTTGTACGGCAGTG -3′, reverse- 5′- GGGGCTGATCCCGTTGATT -3′

**Western blot**. To assess the activation of NF-κB pathway in the presence of rSELP, human microglia were exposed for 2 h to serum-free medium followed by 3 h incubation with serum-free medium supplemented with rSELP (1 μg/ml). Cells were subsequently lysate in RIPA buffer supplemented with fresh proteases and phosphatases inhibitors (Invitrogen, USA). Cells lysate were loaded into a 10% acrylamide/bis-acrylamide gel and the proteins were transferred on nitrocellulose filter and blocked with 5% BSA in Tris-HCl buffer with 1% Tween. Phosphorylated p65 (p-p65) and total p65 antibodies were incubated with the nitrocellulose filter for overnight at 4 °C. Vinculin antibody was incubated for 1h at RT and used as housekeeping to normalize the expression of p-p65 and p65. Rabbit secondary antibody HRP conjugated was incubated for 1 h at RT. SuperSignal™ West Pico Plus chemiluminescent substrate (Thermo Scientific) was added to the membrane, and images were developed using iBright 1500 instrument (Life Technologies, USA). Pixel density of the corresponding protein bands was quantified using ImageJ 1.52v software.

**Animals and ethics statement**. Animals were housed in the Tel Aviv University animal facility. All experiments received ethical approval by the animal care and use committee (IACUC) of Tel Aviv University (approval no. 01-19-015, 01-19-097) and conducted in accordance with NIH guidelines.

**Animal models**. In order to assess microglia density and SELP expression in GB tumors by histology, 6 week-old male C57BL/6 mice (Envigo CRS, Israel) were anesthetized by ketamine (150 mg/kg) and xylazine (12 mg/kg) injected intraperitoneally (IP), and mCherry-labeled murine GL261 GB cells ($5 \times 10^4$ cells) were stereotactically implanted into their striatum ($N = 10$). Patient-derived and iRFP-labeled U251 human GB cells ($5 \times 10^4$ cells) were stereotactically implanted into the striatum of 6 week-old male SCID mice (Envigo CRS, Israel) ($N = 10$). Tumor development was followed by MRI (T1 weighted with contrast agent, MR Solutions) and CRI Maestro™ imaging system.

To investigate the effect of SELP shRNA on tumor growth in the murine GB model, control wild-type (WT), shSELP, or shNC GL261 GB cells ($5 \times 10^4$ cells) were stereotactically implanted into the striatum of 6 week-old male C57BL/6 mice ($N = 15$) (Envigo CRS, Israel). Tumor volume and development were followed by MRI (T1 weighted with contrast agent, MR solutions). Mice were euthanized when they lost 10% body weight in a week, had lost 20% of their initial weight, or when neurological symptoms appeared. Six mice per group were euthanized at day 17 post injection and the brains were resected for further analysis by immunostaining, flow cytometry, and evaluation of gene expression.

Single-Cell RNA-seq of CD11b-positive cells and additional flow cytometry analysis was performed following stereotactical implantation of $5 \times 10^4$ control—WT, shSELP or shNC GL261 GB cells into the striatum of 6 week-old male C57BL/6 mice ($N = 6$) (Envigo CRS, Israel). Tumor volume and development were followed by MRI (T1 weighted with contrast agent, MR solutions). Mice were euthanized at day 17 post injection and the brains were resected. A pool of 3 brains per group of mice was used to generate cell suspensions as described above, and CD11b microbeads were used to isolate microglia/macrophages for single-cell RNA-seq analysis. The remaining cell-suspension was treated with CD45 microbeads and flow cytometry analysis was performed as described above. Alternatively, CD11b and the remaining CD45-isolated cells, as well as the rest of the cell-suspension, were mixed at equal amounts and analyzed for whole-tumor single-cell RNA-seq.

To assess the effect of SELP shRNA on tumor growth in the human GB models, control WT, shSELP or shNC human iRFP-labeled U251 or GB-PD4 GB cells ($5 \times 10^4$ cells) were stereotactically implanted into the striatum of 6 week-old male SCID mice ($N = 9$) (Envigo CRS, Israel). Tumor volume and development were followed by MRI (T2 weighted, MR solutions). Mice were euthanized when they lost 10% body weight in a week, had lost 20% of their initial weight, or when neurological symptoms appeared. Three mice per group were euthanized at day 27 for U251 or day 35 for PD-GB4 post injection and brains were resected for further analysis by immunostaining.

To evaluate the effect of SELPi treatment in the human U251 GB model, iRFP-labeled U251 cells ($5 \times 10^4$ cells) were stereotactically implanted into the striatum of 7 week-old male SCID mice (Envigo CRS, Israel) ($N = 10$). SELPi (0.8 mg/ml) was dissolved in DMSO (0.01%), polyethyleneglycol (93.2 mg/ml), Tween-80 (14.8 mg/ml), and DDW. Mice were treated intravenously (IV) twice a week with PBS, 16 mg/kg SELPi or vehicle (0.01% DMSO, 93.2 mg/ml polyethyleneglycol and 14.8 mg/ml Tween-80 in DDW). Tumor development was followed by monitoring the fluorescence intensity using the CRI Maestro™ imaging system. Mice were euthanized at day 26 post injection and brains were resected for further analysis by immunostaining.

To examine the effect of SELPi treatment in the murine GL261 GB model, GL261 cells ($5 \times 10^4$ cells) were stereotactically implanted into the striatum of 7 week-old male C57BL/6 mice (Envigo CRS, Israel) ($N = 5$) and treated with SELPi three times a week as described above. Tumor volume was evaluated by MRI (T2 weighted, MR solutions). Mice were euthanized when they lost 10% body weight in a week, had lost 20% of their initial weight, or when neurological symptoms appeared. Mice were euthanized at day 17 post injection and brains were resected for further analysis by immunostaining.

To evaluate the effect of SELPi treatment in the murine mesenchymal iAGR53 GB model, GFP-labeled iAGR53 cells ($1 \times 10^5$ cells) were stereotactically implanted into the striatum of 7 week-old male C57BL/6 mice (Envigo CRS, Israel) ($N = 15$) and treated with 16 mg/kg SELPi, IV, QOD. Tumor volume was evaluated by MRI (T1 weighted with contrast agent, MR solutions). Mice were euthanized when they lost 10% body weight in a week, had lost 20% of their initial weight, or when neurological symptoms appeared. Twelve days post cell inoculation, 3 mice per group were euthanized and brain were resected and further processed for histology analysis. To examine the effect of SELPi treatment in the PD-GB model, $5 \times 10^4$ iRFP-labeled patient-derived cells were stereotactically implanted into the striatum of 7 week-old male SCID mice (Envigo CRS, Israel) ($N = 5$). Brain cannulas (NBT) were implanted intraventricularly in the co-lateral hemisphere and the mice were treated stereotactically via the cannulas (1 μl at 0.2 μl/min flow rate) with PBS, 2 mg/kg SELPi dissolved in DMSO or vehicle (DMSO) QOD. Three weeks post cell-inoculation, the mice were euthanized, the cannulas were removed, and tumor volume was evaluated by MRI (T2 weighted, MR Solutions). Brains were resected for immunostaining. To test the effect of SELPi treatment in vivo on mice blood chemistry and blood count, C57BL/6 mice were injected with 16 mg/kg SELPi IV. Twenty-four hours following injecting, mice were euthanized and blood was collected. Blood samples were analyzed by AML Ltd (Herzliya, Isreal).

For all models, $50 \times 10^3$ GB cells were stereotactically injected in 5 μl at 1 μl/min flow rate. Injection coordination relative to the mouse Bregma point were: 2 mm lateral (left), 0.5 mm anterior, 3.5 mm ventral.

In some rare cases, extracranial tumor growth was observed (Supplementary Fig. 20). This occurred mainly in fast-proliferating tumors at late stages of tumor development (Supplementary Fig. 20A), and only in small numbers of mice in each experiment (Fig. 20B). The extracranial part was not included in tumor volume quantification.

**SELPi treatments**. KF 38789 (SELPi) is a commercially-available P-selectin inhibitor manufactured by Tocris BioScience. SELPi was shown to be a selective P-Selectin inhibitor ($IC_{50} = 1.97$ μM) with no inhibiting effects on L-Selectin or E-Selectin. It was shown to block SELP-binding in vitro and in vivo[70]. This compound is commonly used for research of immune-related processes such as tumor metastases formation[71,72]. For the experiment described above, SELPi (0.8 mg/ml) was dissolved in DMSO (0.01%), polyethyleneglycol (93.2 mg/ml), Tween-80 (14.8 mg/ml) and DDW. Mice were injected with 16 mg/kg IV, twice a week, three times a week or QOD as described above. As control (vehicle), mice were injected with DMSO (0.01%), polyethyleneglycol (93.2 mg/ml), Tween-80 (14.8 mg/ml), in DDW at equivalent volume.

**Droplet-based single-cell RNA-Seq**. GL261 GB tumor cells were initially exposed to shSELP or control shRNA plasmid as described above, and injected into mice. Microglia/macrophages (CD11b+), T cells (CD11b-CD45+) and tumor cells (the remaining cell suspension) isolated from the tumors were mixed in equal amounts (see animal model part). We also isolated Microglia/Macrophage cells (CD11b+) only from the two groups of GL261 GB tumor-bearing mice. Cells were encapsulated into droplets, and libraries were prepared using Chromium Single Cell 30 Reagent Kits v3 according to manufacturer's protocol (10× Genomics). The generated single cell RNA-seq libraries were sequenced using a 75 cycle NextSeq 500 high output V2 kit.

**Droplet-based single-cell RNA-Seq data processing**. Gene counts were obtained by aligning reads to the mm10 genome using CellRanger software (v1.3 10× Genomics). To remove doublets and poor-quality cells, we excluded cells that contained more than 10% mitochondrially-derived transcripts, or where less than 500 genes were detected. Among the retained cells, we considered only genes present in >3 cells, which yielded for the global experiment 4654 and 5708 cells from mice exposed to shSELP or control tumors respectively. We then randomly selected 4580 from each group. For the microglia/macrophages focused experiment 4409 and 3376 cells from mice exposed to shSELP or control tumors respectively. We then randomly selected 3300 from each group. Transcript counts for each library were scaled using the ScaleData function in Seurat 3.1.4, which scales and centers features in the dataset. The results were then normalized using the Log-Normalize function in Seurat, by which the feature counts for each cell are divided by the total counts for that cell and multiplied by the scale factor. This is then natural-log transformed using log1p. For principal component analysis (PCA) and clustering, we used a log-transformed expression matrix. The top 12 and 8 PCs, from the global and microglia/macrophages focused experiment respectively, were selected for subsequent tSNE analysis, determined by a drop in the proportion of variance explained by subsequent PCs. We confirmed that the resulting analyses were not particularly sensitive to this choice.

**Single-cell RNA-seq clustering**. The FindClusters function in Seurat was used to identify clusters of cells by a shared nearest neighbor (SNN) modularity optimization-based clustering algorithm. First, we calculated the k-nearest neighbors and constructed the SNN graph. We then optimized the modularity function to determine clusters as previously described[73]. Shifts in the distribution of cells from mice exposed to shSELP or control tumors for each of the clusters were calculated using Fisher's exact test. To further confirm the annotation of the tumors clusters we used inferred CNV analysis implementation of the R package InferCNV (https://github.com/broadinstitute/inferCNV). We first run the analysis on the cell clusters we identified as tumor cells (with a separation to shSELP and shNC). Next, we used the rest of the cell clusters as a reference. Thus, we were also able to confirm that we did not miss any other tumor cluster. The analysis was run with the following arguments: "denoise", default hidden markov model (HMM) settings, "cutoff" of 0.1 as suited for 10× data – set, and a mouse gene map originated in the Mouse Genome Informatics data base.

**Genes differentially expressed between clusters**. The FindAllMarkers function in Seurat was used to find marker genes that were differentially expressed between clusters. This function identifies differentially expressed genes between two groups of cells using a Wilcoxon Rank Sum test with limit testing chosen to detect genes that display an average of at least 0.25-fold difference (log-scale) between the two groups of cells and genes that are detected in a minimum fraction of 0.25 cells in either of the two populations. This step was intended to speed up the function by not testing genes that are very infrequently expressed.

**Visualization of single cell data**. To generate tSNE plots[74,75] of single cell profiles, the scores along the 8 significant PCs described above were used as input for the R implementation of tSNE, by the RunTSNE function in Seurat. Heatmaps were generated using DoHeatmap function in Seurat for the top 10 DE genes in each cluster.

**Single-cell gene signature scoring**. Single-cell gene signature scoring was done as described previously[76]. Briefly, as an initial step, to remove bias towards highly expressed genes the data was scaled using z-score across each gene. For a given gene signature (list of genes), a cell-specific signature score was computed. Scores were computed by first sorting the normalized scaled gene expression values for each cell followed by summing up the indices (ranks) of the signature genes. The same method was also applied to gene-signatures which includes both up-regulated and down-regulated genes. In this case, two ranking scores were obtained separately, and the down-regulated associated signature score was subtracted from the up-regulated generated signature score. A contour plot which takes into account only those cells that have a signature score above the indicated threshold was added on top of the tSNE space, in order to further emphasize the region of highly scored cells. In the second method, total expression of the up-regulated genes was summed and subtracted from the total expression of the down-regulated. Finally, we scaled and averaged the two scoring values.

In addition, we used another independent statistical approach for signature enrichment in single cell data based on the recently published CERNO algorithm[77,78]. Briefly, we used the CERNO method to recalculate the enrichment for a specific gene signature per cell. First, genes were ranked by their normalized expression level, followed by calculation of the statistic $F = -2 \sum_{i=1}^{N} \ln\left(\frac{r_i}{N_{tot}}\right)$, where $N_{tot}$ the total number of the gene in the experiment, and the size of the gene signature is $N$, and for every $g_i \in GS : r_i = rank(g_i)$ as shown before, $F \sim X_{2N}^2$, therefore, we used the chi-square CDF function to calculate a $p$ value per cell, and then, we used FDR correction on the result of each condition. Finally, we used the Fisher exact test to see if one condition is more enriched with cells that are significantly enriched in the CERNO test than the other condition. The CERNO enrichment test shows that the differences reported are significant ($p$ value $= 4.4 \times 10^{-9}$ for the neurodegenerative signature and $p$ value $= 6.48 \times 10^{-8}$ for the T cell communication signature).

**MRI imaging**. Tumor bearing mice were imaged at the Sackler Cellular & Molecular Imaging Center (SCMIC), Tel Aviv University. Mice were anesthetized by ketamine (150 mg/kg) and xylazine (12 mg/kg) injected IP. T1 weighted with contrast agent (Magnetol, Gd-DTPA, Soreq M.R.C. Israel Radiopharmaceuticals) and T2 weighted images were taken by 4.7T MRI—MRS 4000™ (MR solutions). Tumor volume was calculated using Radiant Dicom Viewer 2020.1.1.

**Intravital imaging**. Tumor growth of iRFP- or mCherry- labeled human U251 GB tumors was followed using the CRI Maestro™ non-invasive intravital fluorescence imaging system. Mice were anesthetized with an IP injection of ketamine (150 mg/kg) and xylazine (12 mg/kg) and then placed inside the imaging system. Multi-spectral image-cubes of 550–800 nm were used with spectral range in 10 nm steps using excitation (575–605 nm) and emission (645 nm longpass) filter set. Auto-fluorescence and background were subtracted by a linear unmixing algorithm.

**Ethics**. GB patient tissues: experiments involving human GB tissues were performed following an informed consent, with the approval of the Institutional Review Board (IRB) and in compliance with all legal and ethical considerations for human subject research (approval no. 0735-13-TLV). Healthy human brain tissues: Post-mortem human brain tissue was obtained by autopsy from the Offices of the Chief Medical Examiner of the District of Columbia, and of the Commonwealth of Virginia, Northern District, all with informed consent from the legal next of kin (protocol 90-M-0142 approved by the NIMH/NIH IRB). Additional post-mortem human brain tissue samples were provided by the National Institute of Child Health and Human Development Brain and Tissue Bank for Developmental Disorders (http://www.BTBank.org) under contracts NO1-HD-4-3368 and NO1-HD-4-3383. The IRB of the University of Maryland at Baltimore and the State of Maryland approved the protocol, and the tissue was donated to the Lieber Institute for Brain Development under the terms of a Material Transfer Agreement. Clinical characterization, diagnoses, and macro- and microscopic neuropathological examinations were performed on all samples using a standardized paradigm, and subjects with evidence of macro- or microscopic neuropathology were excluded. Details of tissue acquisition, handling, processing, dissection, clinical characterization, diagnoses, neuropathological examinations, RNA extraction, and quality control measures were as described previously[79]. The Brain and Tissue Bank cases were handled in a similar fashion (http://medschool.umaryland.edu/BTBank/ProtocolMethods.html).

**Human FFPE specimens**. FFPE GB samples were obtained from Tel Aviv Sourasky Medical Center following an informed consent. A total of 60 samples were collected: 36 samples of patients who survived short-term- STS (69% men; $65 \pm 2$ years;

survival 3.7 ± 0.2 months), and 24 samples of patients who survived long-term- LTS (58% men; 56 ± 3 years; survival 48 ± 3.9 months). Normal human brain FFPE samples were obtained from the Lieber Institute (Baltimore, MD, USA).

**Frozen OCT tissue fixation**. Tumor bearing mice were anesthetized with an IP injection of ketamine (150 mg/kg) and xylazine (12 mg/kg) and perfused with PBS followed by 4% Paraformaldehyde (PFA). Brains were harvested, and incubated with 4% PFA for 4 h, followed by 0.5 M of sucrose (BioLab) for 1 h, and 1 M sucrose overnight (ON). The brains were then embedded in optimal cutting temperature (OCT) compound (Scigen) on dry ice and stored at −80 °C.

**Immunostaining**. OCT embedded tumor samples were cut into 5 μm thick sections. Staining was performed using BOND RX autostainer (Leica). Sections were stained by hematoxylin and eosin (H&E) and immunostained for: pro-liferating cells using rabbit anti-human/mouse KI67 antibody and Alexa Fluor 488-goat anti-rabbit secondary antibody; Blood vessels using rat anti-mouse CD31 antibody and Alexa Fluor 488-goat anti-rat secondary antibody; Iba1 activated microglia using rabbit anti-mouse Iba1 antibody and Alexa Fluor 488-goat anti-rabbit secondary antibody; CD8-positive T-cells using rat anti-mouse CD8 antibody and Alexa Fluor 488-goat anti-rat secondary antibody; CD4-positive T-cells using rat anti-mouse CD8 antibody and Alexa Fluor 488-goat anti-rat secondary antibody; and FOXP3-positive T-cells using rabbit anti-mouse/human FOXP3 and Alexa Fluor 488-goat anti-rabbit secondary antibody. Prior to antibody incubation, slides were incubated with 10% goat serum in PBS x1 + 0.02% Tween-20, for 30 min to block non-specific binding sites. Slides were incubated with primary antibodies for 1 h, and then washed and incubated with secondary antibodies for an additional 1 h. They were then washed and treated with ProLong® Gold mounting with DAPI before being covered with coverslips. Patient FFPE samples were stained for Iba1 and human SELP as described above. Stained samples were imaged using the EVOS FL Auto cell imaging system (ThermoFisher Scientific). At least three fields of each individual sample were imaged and quantified using ImageJ 1.52v software. Quantification of positive staining was performed by measuring the total area stained in each image fol-lowing background subtraction, using single color images representing the correlated marker.

**Statistical analysis**. Data are expressed as mean ± standard deviation (s.d.) for in vitro assays or ± standard error of the mean (s.e.m.) for in vivo assays. Statistical significance was determined using an unpaired, two-sided $t$ test when comparing between two groups, and multiple comparisons ANOVA test when comparing more than two groups. $P < 0.05$ was considered statistically significant. For Kaplan–Meier survival curves, $p$ values were determined using log rank test. For in vivo tumor growth curves, $p$ values were determined using one-way ANOVA, Dunn's test, or Holm-Sidak's test. Statistical analysis was performed using GraphPad Prism 8.

**Molecular characterization of human GB samples**. The molecular subtypes of our GB FFPE samples from short-term survivals (STS) included at least one pro-neural, one mesenchymal and one classical subtypes. Human U251 GB cell line is considered as proneural[80]. The patient-derived cell lines exploited in this study are all IDH WT while PD-GB1, PD-GB2 and PD-GB4 are p53-mutated, PD-GB3 is p53 WT and ATRX mutated, and PD-GB4 is ATRX WT.

**Molecular characterization of murine GB samples**. GL261 is considered to be a mesenchymal-like phenotype[81]. GB cell lines isolated from lenti-induced GB mouse models include: iAGR53—primary astrocytes from GFAP-Cre mice trans-duced with lenti-HRasV12-shp53—Mesenchymal subtype; EGFRviii-shp16- Clas-sical subtype; and PNp53—derived from a tumor induced in GFAP-Cre mouse (cell of origin neural stem cell, NSC, in the sub ventricular zone) injected with lenti-PDGFB-shp53—Proneural subtype[68,69]

**Reporting summary**. Further information on research design is available in the Nature Research Reporting Summary linked to this article.

## Data availability
The sequence data generated in this study have been deposited in the Gene Expression Omnibus (GEO) and are accessible through the GEO Series accession number GSE156663. The GBM patient sequencing data analyzed with the GEPIA2 web server are available in the TCGA database (Project ID: TCGA-GBM, https://portal.gdc.cancer.gov/projects/TCGA-GBM). The remaining data are available within the Article, Supplementary Information or available from the authors upon request. Source data are provided with this paper.

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

## Acknowledgements

The Satchi-Fainaro laboratory has received partial funding from the European Research Council (ERC) Consolidator Grant Agreement no. [617445]-PolyDorm and ERC Advanced Grant Agreement no. [835227]-3DBrainStrom; and ERC Proof of Concept (PoC) Grant (862580; 3DCanPredict), The Israel Science Foundation (Grant nos. 918/14 and 1969/18); The Israel Cancer Research Fund (PROF-18-682); The Nancy and Peter Brown friends of The Israel Cancer Association (ICA) USA, in memory of Kenny and Michael Adler (20150909); and from the Morris Kahn Foundation. E.Y. is a Gail White and Billy Cohen Fellow in Brain Cancer Research. P.O. thanks the Naomi Foundation for the Global Research and Training Fellowship in Medical and Life Sciences. We thank Nir Friedman for his help with the single cell RNA-seq experiments and Anat Reiner Benaim for her advice on statistical analysis. We thank Nadir Arber and Shiran Shapira (Tel Aviv Sourasky Medical Center) for kindly providing us anti-human (SWA11) and anti-murine (Ml.69)-CD24 neutralizing antibodies. The Lieber Institute for Brain Development extends our appreciation to the physicians and staff at the Washington DC, Northern Virginia, and State of Maryland Medical Examiners' Offices, as well as the donors and their next of kin for the provision of brain tissue for this study.

## Author contributions

Conceptualization: R.S.-F. and E.Y.; Experimental: E.Y., P.O., S.P., N.A., D.B.-S., G.T., S.G., R.K., S.I.D., S.R.-Z.; Formal Analysis and Investigation: E.Y., R.S.-F., A.M., R.S.; Resources: R.G., Z.V., H.B., T.M.H., D.M.-F., P.M.; Writing: E.Y., R.S.-F., A.M., P.O., D.B.-S., G.T., S.P.; All authors reviewed the manuscript and provided comments. Funding Acquisition: R.S.-F.; Supervision: R.S.-F.

## Competing interests

H.B. is a consultant for AsclepiX Therapeutics, Perosphere Inc., StemGen, InSightec, Accelerating Combination Therapies, Camden Partners, LikeMinds, Inc., Galen Robotics, Inc., Nurami Medical and Be Cured: Fighting Brain Cancer. R.S.F. is a Board Director at Teva Pharmaceutical Industries Ltd. All other authors have no competing interests to declare.
