## [Peer Review File · Nature Communications]

REVIEWER COMMENTS

Reviewer #1 (Remarks to the Author):

In the work by Yeini et al, the authors examine how the tumor microenvironment, specifically the immune microenvironment, contributes to the progression of glioblastoma multiforme, the most common and lethal primary brain tumor. Two major hallmarks of these tumors include their highly invasive/infiltrative and angiogenic nature and an immunosuppressive tumor microenvironment characterized by skewed polarization of microglia, the brain resident macrophages. Here, the authors tackle the immunosuppressive microenvironment and suggest that an adhesion molecule called P-Selectin (SELP) expressed on tumor cells and microglia as well its ligand, P-Selectin Glycoprotein Ligand 1(PSGL1), expressed on microglia facilitates cross talk between tumor cells and immune cells through modulation of this receptor-ligand levels. Specifically, they suggest that the interaction results in an immunosuppressive polarization of microglia, likely through activation of NF-kappa B signaling and secretion of immunosuppressive molecules such as IL10 and TGF-B. They test their hypothesis through genetic and pharmacological ablation of SELP-PSGL1 signaling.

Overall, this work follows up on a previous publication by this group, which showed that conjugation of the chemotherapeutic agent, paclitaxel, to a dendritic polyglycerol sulfate (dPGS) nano carrier promoted its binding to SELP found on tumor cells and the tumor endothelium, which resulted in a significant decrease in tumor burden. Even though this work used SELP for delivery of therapeutics, it provided the lead-in for work described in the current manuscript. This work has clinical significance for a disease that has been extremely refractive to treatment, despite years of research. Targeting selectins is the focus of a few ongoing clinical trials. Thus, the rationale for this work by Yeini et al., is strong.

Having said that, it is established by other studies that abrogating signaling through SELP reduces cell proliferation. Previous work from the senior author's (Satchi-Fainaro) group showed that angiogenesis is decreased when SELP on the tumor endothelium is bound by paclitaxel conjugated nanoparticles. Therefore, the novelty of the current work stems from the connection between SELP and control of microglial polarization, and their phagocytic activity. Although stated tangentially in the discussion section, previous findings from a number of other groups has demonstrated the involvement of SELs and microglia in the pathogenesis of autoimmune and neurodegenerative diseases. Therefore, the major observation from the current work that activation of SELP-PSGL1 signaling in microglia is not entirely new, but can be considered so in the context of GBM. This is not certainly not trivial, considering the difficulties in treating GBM and the importance of the immunosuppressive microenvironment in the lack of success with many immunotherapeutic approaches. From a mechanistic point of view, the authors suggest that targeting the SELP-PSGL1 axis could potentially involve NF-K B signaling, and imposition of an immunosuppressive phenotype on microglia. This is quite important, but has been addressed only somewhat cursorily (in supplemental data) in the manuscript. A last global comment will be that the authors have to address (even if in the discussion section), the possibility of redundancy and some level of compensation by the other selectin family members when SELP is targeted.

Comments for each figure is provided below.

Fig. 1: Please show Ki-67 staining of tissues from Fig. 1A to support data shown in Fig. 1B. The authors have utilized the different PD-GB cells interchangeably. Can the authors comment on the sub-classification of the PD-GBs? Are they pro-neural, mesenchymal or classic? Please provide the IDs of the other cytokines on the array in Fig. 1E. Along the same lines, what are the array locations of the cytokines evaluated in Fig. 1D? What is the status of PSGL1 in PD-GB3 and conversely, what is the SELP status in PD-GB4?

Fig. 2: The authors state that multiple shRNAs were tested and only one resulted in significant SELP knockdown. Can the authors please use pooled pre-validated siRNA to show an acute response so that there is more confidence regarding the specificity of the SELP shRNA utilized in

the study.

Fig. 3: Please indicate in the legend which experiments involved the use of PD-GB4 and U251 cells.

Fig. 4: Images for CD31 staining in Fig. 4C do not reflect the quantitation shown. Specifically, it appears that the control shRNA causes a reduction in CD31 staining relative to control panel. Also, since the shRNA expression is not conditional, and SELP knockdown blocks tumor cell proliferation, how can the authors rule out the observed effects on tumor size being a result of few viable cells?

Fig. 5: Approximately 14 clusters are shown in Fig 5A. Most are identified by the cell type. However, some are indicated by numbers. Can the authors please clarify which of these cell types are shown in Fig. 5B, which has only 8 bars. Likewise, please label the clusters in Fig. 5C with the cell type ID. Please include a table to show the top 10 most significantly upregulated or downregulated genes for each cluster so that there is a clearer picture of all of the cell types that are affected by SELP knockdown. It is acknowledged that the authors' focus is on angiogenesis, tumor cell proliferation and microglial biology.

Fig. 6: On page 32, the authors make the comment that: "We note that caspase staining increased in shSELP tumors only in immunocompetent mice. This phenomenon could be correlated with enhanced infiltration of CD8+ T cells suggesting that these cells play a cytotoxic role in shSELP GB". Can the authors please add a figure number for this comment?

Fig. 7: Can the authors clarify why the intraventricular and systemic administration of SELP inhibitor was not done with the same GBM model? On page 34 and elsewhere, references have been made regarding published work describing SELP inhibition and T cell trafficking. Please cite the reference(s). CD31 staining in SELPi treated samples does not seem to very different from controls in Fig. 7F. Can the authors include a section that is more reflective of the graphical data? Please include the following data for PD-GB4 tumors in Fig. 7G: H&E, Iba1, CD31, Ki67 staining.

Discussion:

1. Please expand on whether soluble versus membrane bound SELP may have unique contributions to GBM pathogenesis. Can one work as a decoy, for example?
2. Please provide more details on the relationship between selectins and microglia in neurodegenerative diseases and autoimmune diseases and describe the novelty of your findings in the context of what has been shown in the above systems.
3. Please expand on your discussions regarding the therapeutic potential of SELPi for GBMs. Can they be used as single agents or in combination with immunotherapy or standard of care? Is anything known about radiation or chemotherapy altering the levels of SELP and contributing to the immunosuppressive microenvironment?

Reviewer #2, expert in neuroimmunology/trafficking (Remarks to the Author):

Glioblastoma is a highly lethal brain tumor and more effective therapeutic strategies are needed. In this paper the authors investigated the molecular mechanisms controlling the interplay between the local immune system and glioblastoma (GB) cells. They show a role for P-selectin, a molecule classically involved in leukocyte trafficking and platelet activation, in microglia-tumor cell interactions and GB progression. A role for P-selectin has been previously suggested in the context of other oncological diseases. However, the role of this molecule and its main ligand PSGL-1 is unclear in GB pathogenesis, and the authors comprehensively show that P-selectin expression on human and mouse GB cells modulates microglia activation and T cell recruitment. The usage of a

P-selectin inhibitor delays the growth of murine and human GB tumors in mice. These results suggest that P-selectin blockade interferes with microglia phenotype and may represent a new therapeutic approach for GB. The manuscript is well-written and the results are interesting and overall convincing. However, the study raises some concerns that need to be addressed.

1. It is unclear from Fig. 1 if GB also induces an increase of microglia density, which represents a common sign of microglia activation. Also, how many patients (FFPE samples) were studied for microglia activation in Fig. 1a?

2. In line with point 1, do the authors have evidence that GB cells may recruit blood myeloid cells and increment brain macrophage/microglia cells?

3. Figure 1K shows the quantification of immunostaining of SELP suggesting more positive staining in short-term survivors (STS) GB patient FFPE samples compared to long-term survivors (LTS) or normal human brain tissue. However, this reviewer could not find a description of the method used for data quantification.

4. The role of P-selectin-PSGL-1 axis is not well demonstrated in this paper. rP-selectin induces an M2-like phenotype in microglia cells in vitro (Figure 3). However, in this experimental setting, the authors assume that the effect of rP-selectin is mediated by PSGL-1 expressed by microglial cells. It would be important to more clearly demonstrate that the induction of M2 phenotype is mediated by P-selectin-PSGL-1 interactions and/or if other P-selectin ligands may be involved. The assays performed in the presence of a blocking anti-PSGL-1 antibody or using PSGL-1 deficient microglial cells would be helpful to clarify this aspect. In addition, is PSGL-1 crosslinking in microglial cells inducing similar effects as soluble rP-selectin?

5. In some experiments GB-CM have been used either alone or in combination with SELP neutralizing antibody or SELP inhibitors. However, P-selectin shedding may be variable and no direct evidence showing the content of P-selectin in GB-CM is shown (Figure 3 C, D, F, G, N).

Minor points

1. "SELP" is more commonly used to indicate the P-selectin encoding gene. "P-selectin" or "CD62P" would be more indicated to refer to the protein.

2. The size of scale bars in all immunofluorescence images is unclear and should be mentioned in figure legend.

3. A more detailed description of the stereotactic GB cell injections should be provided (e.g. brain coordinates, injected volume).

4. SELPi dosage and the vehicle need to be specified on page 15 ("To examine the effect of SELPi treatment in the murine GL261 GB model, GL261 cells (5×10^4 cells) were stereotactically implanted... and treated with SELPi three times a week").

5. The number of the supplementary figure showing flow cytometry data is lacking on page 28 "(Supplementary Figure C, D)".

6. Page 29, Figure 4, panel C: the representative MRI images are not mentioned.

7. Page 34: there are some typing errors in "(Figure 7A C, D)- C)".

8. Supplementary figure legend 3, page 50: the identification of the panels from "C" to "F" have some errors.

9. Supplementary figure 5, page 52: "C" is missing in the figure panel.

Reviewer #3, expert in microglia/macrophages in GBM and single cell sequencing (Remarks to the Author):

This is an exciting and excessive work that identifies P-selectin as an important regulator of GBM associated microglia phenotype and microglia-tumor interaction. This is an exciting and excessive work that identifies P-selectin as an important regulator of GBM associated microglia phenotype and microglia-tumor interaction. I acknowledge the work is extreme and has a limitless number of supplementary figures; I can only imagine how much work went into it. But unfortunately, the way it is presented, it is tough to read and follow it. I do have some serious concerns about the experimental design and quality of some of the important data.

In vitro work is based only on microglia and tumor cell cultures; in reality, in vivo context, there is an immense contribution of bone-marrow-derived macrophages that are different from microglia. Authors should use BMDM cultures and see whether the effects are microglia specific or, in general, myeloid-specific.

Many tumor lines are used both human and mouse, but unfortunately, they are used back and forth for various parts of one story. There is a big statement of using different murine lines PN, MES, and CL, but they appear best in some subfigures, so do some of the human lines and primary cultures. Most of the story really on GL-261 and U251. These extra lines should be removed, or authors should do all the key experiments using all these lines. All these lines have different genotypes, and they will affect myeloid cells differently, which is apparent even from low-quality representative images for IBA1.

All the representative images are of low quality. Double and triple staining are hard to see. P-selectin staining should be double-stained with staining specific for tumor cells, GAMs, and endothelial cells.

MRI images show some extracranial tumor growth, which should be clarified, and better-quality images should be provided.

The introduction would benefit from more focus and a clear message. The microglia and microglia/macrophages are used interchangeably, which is confusing and not appropriate. Claiming all the effects are microglia driven is not reasonable without the use of BMDM.

Treatment with inhibitor needs more clarification and information; in addition, more evidence is required to show target inhibition and specificity.

There is a serious attempt to claim M2 versus M1 changes in the macrophage profile. Authors should focus on signaling and leave M1 and M2 states out, since they fail to emerge as isolated pure phenomena in vivo, especially in GBM.

Reviewer #4, expert in single cell sequencing and cancer (Remarks to the Author):

Yeini et al aim to investigate the crosstalk of microglia with glioma cells and how this impacts glioma cell proliferation and invasion.

1. They first use a commercially available and patient-derived microglia and glioma cells and show

that glioma proliferation and migration is enhanced in the presence of microglia. They use a protein array to check for differences in murine co-cultures vs. glioma vs. microglia supernatants and find that a few are differentially regulated in the co-culture, and find among others SELP (P-selectin) to be higher in co-culture. They then measure mRNA and protein in a separate human model incubated with CM of microglia. Staining for Iba1, SELP and PSGL-1 on mouse FFPE healthy brain and malignant tissue.

2. Co-culture of cancer cells with SELP KD results in impaired proliferation and prolonged wound healing time compared to WT cells or scramble KD controls. The authors conclude that the microglia interactions are necessary for increased proliferation/migration.

3. Vice versa, glioma CM incubation of microglia results in increased SELP expression/secretion and upregulation of PSGL-1 in scRNA-seq data. TCGA analysis finds an association of PSGL-1 and CD4. Analysis of other single-cell data shows PSGL-1 expression in TILs.

4. The authors then investigate the effect of SELP on microglia phenotypes and perform flow-cytometry for "M2" markers CD163 and CD206, and find that microglia incubated with glioma CM have increased CD163 and this is reduced in SELP KD. They look at mRNA levels of "M1" vs. "M2" markers after treatment with recombinant SELP. They check for IL10 and TGFb. They make a link to phagocytosis and show that rSELP induces phagocytosis in microglia, and reduction of NO release.

5. They then generate a SELP KD in a murine model (with reduced proliferation) and show that intra-cranial injection results in smaller tumors and improved survival compared to control and scramble control. IF of tumors shows reduced Ki67 activity, increased caspase-3 and reduced CD31 (endothelial cell density)

6. ScRNA-seq of shNC vs. shSELP tumors: they define various cell types and focus on the main ones that are sufficiently "represented". They define tumor cell "clusters" in shNC vs. shSELP and show differential expression among these tumor cell clusters. They select "differentially expressed" genes of major cellular functions, including cell proliferation and invasion. They then generate more scRNA-seq data after cd11b sorting because the microglia/macrophage population because there is "uneven internal distribution" among shNC vs. shSELP tumors. They claim that among clusters, there were differences in shNC vs. shSELP. They suggest that cells from shSELP score more highly for a "neurodegenerative" signature while other clusters scored more highly for an "antigen presentation and chemokine signature".

7. They repeat the in vivo experiment with a different model in immunocompromised mice and find again that shSELP results in smaller tumors.

8. They repeat the in vivo experiment in syngeneic model and a pdx model using a SELPi and find results concordant with shSELP models.

The manuscript is confusingly written and really needs to be streamlined. The authors move between murine and human models and cell types without obvious rationale which makes it difficult to really understand the message of the paper. Apart from this, there are several technical and experimental which limit the interpretability of the data and in some instances simply miss the correct controls. Also, the entire manuscript is void of any necessary experiments, below are a few examples. I will address each of the major results using the same numbering:

Major

Point 1: In their description of results (and throughout the MS) the authors conflate macrophages with microglia; for example, they consider CD11b, which is a non-specific surface marker, for some of their experiments. These populations need to be distinguished with more accurate

granularity. The steroid flow-cytometry does not show any obvious differences. The authors should include raw data and gating. In figures 1A, J, K, and really throughout the entire manuscript, the intensity of DAPI varies between conditions, indicating that these images were taken with different exposure times or were stained differently. This makes a comparison (leave alone quantification) of any IF in this manuscript impossible. There is also a lot of background staining, for example several non-cellular punctae in the PSGL1 staining, which indicate that these antibodies are not adequately titrated. The authors also conflate expression and secretion which makes interpretation of their claims extremely difficult to follow. The use of gray scale quantification is somewhat outdated and extremely prone for errors - the authors should show in immunoblots the differences of SELP.

Point 2: This experiment misses essential controls: first, there is no SELP KD alone group included. It is impossible to separate effects of SELP KD alone vs. SELP KD with co-cx. As outlined above, the entire manuscript is void of rescue experiment. The authors need to re-express an SELP ORF in the KD (or KOs) and repeat experiments. On a more basic level, the fact of the matter is that the authors did not achieve a convincing KD as shown in Fig. 2B - there is still a large population of cells with preserved SELP expression. They should include gating data. The fluorescence intensity shown in C is inconsistent with the data shown in B.

Point 3 and 4: Co-authors on this manuscript with training in immunology should be aware of the outdated use of M1 vs. M2 nomenclature in general, but in particular their classification based on 1-2 marker proteins/genes. Since the authors have single-cell data, they can infer M1-like or M2-like cells, but this was not performed here - why not? This entire section is extremely confusing moving between models to show different aspects of "M1 vs. M2" features. Can the authors simply show all (at least once) in one model, but show it consistently.

Point 5: The fatal flaw of these in vivo experiment (and subsequent ones) is the fact that SELP KD have a reduced proliferation rate in vitro as the authors show in the supplement. Of course this affects the (reduced) growth in vivo and will result in a prolonged survival of animals. The correct control is a matched cell line with the same proliferation rate. The authors only measured tumor volume 3 times in the entire experiment over the course of 15 days? Ki67 staining is non-specific in some sections shown, staining cell membranes etc. All images suffer from a too high contrast and show different intensities/exposures, they need to be repeated at the same exposure time. Also, caspase staining shows a lot of non-specific background which is much higher in the shSELP group.

Point 6: First, The authors do not provide convincing evidence that what they call tumor cells are indeed tumor cells. They should provide inferred CNV analysis which will show pathognomonic CNV alterations which are well reflected in single-cell data, as previously shown by many others. The interpretation of cell abundances in A and B is impossible, because cells were derived from tumors of different sizes. It is a well known bias that smaller tumors have a vastly different immune composition (typically more T cells, less myeloid cells) compared to larger tumors of the same histology/experiment. The fact of the matter is also that there is no difference in the result shown in Fig. 5F, and the neurogenerative vs. antigen presentation signatures are evenly distributed between control and SELP KD.

Points 7 and 8: same limitation of in vivo experiments as above. shSELP have a lower proliferation rate, which will result in smaller tumors in vivo (and therefore better survival) and increased CD8 T cells (because the tumors are smaller). The authors need to at least correct each quantification for mg of tumor. Also, Ki67 in Fig. 6 C seems to stain everything in a highly non-specific manner in the WT tumors while in the SELP KD there are several punctae that do not seem to correspond to a nucleus, suggesting further that the authors measure a lot of background artifact. In Fig 7C, there is a discrepancy between the standard deviations when in C and E (in that, the SD in 7C is quite large, consistent with biological noise, while in E there appears to be nearly no error rate for the SELP KD tumors despite). Why do the authors show different regions in 7F and not sequential

tissue sections?

minor:

page 29 missing figure number in suppl. figure

page 35 needs to be corrected (Figure 7A C, D)- C)

Reviewer #1, expert in brain tumors and P-selectin targeting (Remarks to the Author):

In the work by Yeini et al, the authors examine how the tumor microenvironment, specifically the immune microenvironment, contributes to the progression of glioblastoma multiforme, the most common and lethal primary brain tumor. Two major hallmarks of these tumors include their highly invasive/infiltrative and angiogenic nature and an immunosuppressive tumor microenvironment characterized by skewed polarization of microglia, the brain resident macrophages. Here, the authors tackle the immunosuppressive microenvironment and suggest that an adhesion molecule called P-Selectin (SELP) expressed on tumor cells and microglia as well its ligand, P-Selectin Glycoprotein Ligand 1 (PSGL1), expressed on microglia facilitates cross talk between tumor cells and immune cells through modulation of this receptor-ligand levels. Specifically, they suggest that the interaction results in an immunosuppressive polarization of microglia, likely through activation of NF-kappa B signaling and secretion of immunosuppressive molecules such as IL10 and TGF-B. They test their hypothesis through genetic and pharmacological ablation of SELP-PSGL1 signaling.

Overall, this work follows up on a previous publication by this group, which showed that conjugation of the chemotherapeutic agent, paclitaxel, to a dendritic polyglycerol sulfate (dPGS) nano carrier promoted its binding to SELP found on tumor cells and the tumor endothelium, which resulted in a significant decrease in tumor burden. Even though this work used SELP for delivery of therapeutics, it provided the lead-in for work described in the current manuscript. This work has clinical significance for a disease that has been extremely refractive to treatment, despite years of research. Targeting selectins is the focus of a few ongoing clinical trials. Thus, the rationale for this work by Yeini et al., is strong. Having said that, it is established by other studies that abrogating signaling through SELP reduces cell proliferation.

We would like to thank the Reviewer #1 for appreciating the strength of our work and its clinical relevance for GB patients.

To the best of our knowledge, except for one clinical trial testing anti-E-Selectin (not anti-P-selectin) treatment to prevent multiple myeloma metastasis (NCT02306291), clinical trials targeting selectins focus on sickle-cell anemia and cardiovascular diseases (NCT03814746, NCT03474965, NCT01245634, NCT00876902), or cancer-related conditions such as thrombosis and rheumatoid arthritis (NCT00048581, NCT02285738, NCT00967148). No clinical trials have been conducted to test the direct effect of anti-selectins (nor specifically anti-P-Selectin) on tumor cell proliferation or the immune-response in solid tumors.

Previous work from the senior author's (Satchi-Fainaro) group showed that angiogenesis is decreased when SELP on the tumor endothelium is bound by paclitaxel conjugated nanoparticles.

Therefore, the novelty of the current work stems from the connection between SELP and control of microglial polarization, and their phagocytic activity. Although stated tangentially in the discussion section, previous findings from a number of other groups has demonstrated the involvement of SELs and microglia in the pathogenesis of autoimmune and neurodegenerative diseases.

In our previous work, we showed that angiogenesis was decreased when the therapy (paclitaxel-bound to sulfated nanoparticle) was selectively targeted to SELP on the angiogenic endothelial cell membrane in order to enhance the internalization of paclitaxel, hence killing the proliferating endothelial cells by the targeted-paclitaxel (and not by the inhibition of SELP). The targeting moiety in our previous work was SO₃ which does not have any inhibitory effect on SELP, but rather only binds to it.

In our discussion, we stated that other groups have shown the involvement of microglia (without connection to SELs in general nor SELP in particular) in autoimmune and neurodegenerative diseases (Krasemann, *et al.*, The TREM2-APOE pathway drives the transcriptional phenotype of dysfunctional microglia in neurodegenerative diseases. *Immunity*, 2017. 47(3): p. 566-581.e9). We did not state in the Discussion section the involvement of SELs in relation to microglia in the pathogenesis of these diseases. In order to address the reviewer's comment, we have now found a new report that mentions PSGL-1 as part of a list of microglia core genes that are differentially expressed (down-regulated) in Multiple Sclerosis (Jordão, M.J.C., *et al.*, Single-cell profiling identifies myeloid cell subsets with distinct fates during neuroinflammation. *Science*, 2019. 363(6425): p. eaat7554). We added this report to the Discussion.

Therefore, the major observation from the current work that activation of SELP-PSGL1 signaling in microglia is not entirely new, but can be considered so in the context of GBM. This is certainly not trivial, considering the difficulties in treating GBM and the importance of the immunosuppressive microenvironment in the lack of success with many immunotherapeutic approaches. From a mechanistic point of view, the authors suggest that targeting the SELP-PSGL1 axis could potentially involve NF- κ B signaling, and imposition of an immunosuppressive phenotype on microglia. This is quite important, but has been addressed only somewhat cursorily (in supplemental data) in the manuscript. A last global comment will be that the authors have to address (even if in the discussion section), the possibility of redundancy and some level of compensation by the other selectin family members when SELP is targeted. In this work, we found that SELP is overexpressed following GB-microglia interactions, independently from our previous work that showed the use of SELP only for drug targeting as we found it to be overexpressed on GB cells and on proliferating endothelial cells of GB-angiogenic blood vessels (Ferber, Tiram, *et al.*, eLife 2017). Although we and others have shown the expression of SELP in different cancer tissues, its role in tumor invasion and immunology, and specifically in GB, as well as in the biology of glioma-associated microglia/macrophages (GAMs), was not reported before. We agree that the observation of NF- κ B induction is important. However, deeper investigation of this pathway is out of the scope of this manuscript and is under investigation in different projects in our lab. We have now added the following paragraph to the Discussion section regarding other members of the Selectin family:

“Other members of the selectin family also bind PSGL-1. Although SELP binds to PSGL-1 at higher affinity than E-Selectin and L-Selectin (137), their expression may serve as a compensation mechanism for PSGL-1 activation. In addition, it is plausible that there is some redundancy between the different Selectins. However, the shRNA used in this research was designed specifically to silence SELP, and SELPi was previously shown by the manufacturer to be selective for SELP with no effect observed neither on E-Selectin nor on L-Selectin (https://www.tocris.com/products/kf-38789_2748). Thus, we show that specific inhibition of SELP using both molecular and pharmacological agents delayed tumor progression. Nonetheless, further research is required in order to discriminate the possibility of compensation by other Selectins while looking into the added value of inhibiting all three, P, E and L selectins, in order to halt GB’s aggressive progression.”

137. Mehta, P., R.D. Cummings, and R.P. McEver, Affinity and kinetic analysis of P-selectin binding to P-selectin glycoprotein ligand-1. *J Biol Chem*, 1998. 273(49): p. 32506-13.

Comments for each figure is provided below.

Fig. 1: Please show Ki-67 staining of tissues from Fig. 1A to support data shown in Fig. 1B. The authors have utilized the different PD-GB cells interchangeably. Can the authors comment on the sub-classification of the PD-GBs? Are they pro-neural, mesenchymal or classic?. Please provide the IDs of the other cytokines on the array in Fig. 1E. Along the same lines, what are the array locations of the cytokines evaluated in Fig. 1D? What is the status of PSGL1 in PD-GB3 and conversely, what is the SELP status in PD-GB4?

Unfortunately, there is no clinical data which directly indicates the sub-classification of the tissue samples from which we isolated the different PD-GB cells. However, we did mention in the Methods section all the available clinical data regarding the molecular phenotype of PD-GB cells:

Molecular characterization of human GB samples. The molecular subtypes of our GB FFPE samples from short-term survivals (STS) included at least one proneural, one mesenchymal and one classical subtypes. Human U251 GB cell line is considered as proneural [55]. The patient-derived cell lines exploited in this study are all IDH WT while PD-GB1, PD-GB2 and PD-GB4 are p53-mutated, PD-GB3 is p53 WT and ATRX mutated, and PD-GB4 is ATRX WT.

In addition, we have now stained for SELP and PSGL-1 staining on PD-GB2, PD-GB3 and PD-GB2 (Supplementary Figure 3). We have also co-stained for Iba1 and Ki-67 using U251, PD-GB4 and GL261

tumors. The results show positive staining of SELP and PSGL-1 in all the models used and positive staining of Ki-67 in tumor areas enriched with Iba1 positive cells.

Supplementary Figure 3. Immunostaining of GB mouse models. A-B. Immuno-fluorescence staining presented at separate fluorescence channels, showing the co-staining for Iba1 and SELP/PSGL-1 in different GB mouse models shown in Figure 1 and in two additional PDX GB mouse models. **C, D.** Immuno-fluorescence staining presented at separate fluorescence channels, showing co-staining for Iba1 + Ki-67 (C) and SELP + CD31 (D) in U251, PD-GB4 and GL261 GB mouse models.

We have now performed additional cytokine array using PD-GB4 cells as well, showing high fold change in SELP secretion in the co-culture of PD-GB4 and human microglia. As per the reviewer's request, we added the IDs of other cytokines detected and their location on the membranes in the Supplementary Information (Supplementary Figure 4). A full list of the cytokines evaluated using both cytokine arrays (human and murine) can be found in the following links: https://www.rndsystems.com/products/proteome-profiler-mouse-xl-cytokine-array_ary028, <https://www.raybiotech.com/l-series-507-label-based-human-array-1-membrane-2/>.

Supplementary Figure 4. Identified cytokines by human and murine cytokine arrays. A, C. Human (A) and murine (B) cytokine arrays membranes showing in red the over secreted factor in a co-culture of GB cells and microglia compared to monocultures. **B, D.** Gene IDs of over-secreted factors detected by human (B) and murine (D) cytokine arrays.

Fig. 2: The authors state that multiple shRNAs were tested and only one resulted in significant SELP knockdown. Can the authors please use pooled pre-validated siRNA to show an acute response so that there is more confidence regarding the specificity of the SELP shRNA utilized in the study.

For the murine SELP shRNA which was used on GL261, we indeed tested for five different sequences and chose the one that showed the highest inhibition as shown now in Supplementary Figure 5. This shRNA was used for *in vivo* experiments performed on GL261 mouse model (Figure 5). For the human shRNA used on PD-GB4 and U251 (shown in Figure 2 and Figure 4 on PD-GB4 and in Supplementary Figure 5 and Figure 4 for U251) we did use a pool of three pre-validated shRNA sequences. In order to further validate the specificity of SELP as our target, we included the use of SELPi or neutralizing antibody in all of the above mentioned cell lines in a variety of *in vitro* and *in vivo* assays.

Fig. 3: Please indicate in the legend which experiments involved the use of PD-GB4 and U251 cells.

All the conditioned media used for the experiments shown in Figure 3 and in new Supplementary Figure 8 were generated using PD-GB4 treating human microglia. Co-cultured spheroids consisted of PD-GB4 cells and microglia are shown now in Supplementary Figure 8. This is mentioned now in the figure legend.

Fig. 4: Images for CD31 staining in Fig. 4C do not reflect the quantitation shown. Specifically, it appears that the control shRNA causes a reduction in CD31 staining relative to control panel.

As can be seen in the representative images and in the quantification, the average of positive CD31 staining detected was slightly lower in the control shRNA than in the control. However, the differences in CD31 staining between the control and the negative shRNA are not statistically significant ($p=0.3$). The images shown are the closest to the average signal detected in all the fields and different samples from mice that were used. We believe that these non-significant changes rise from the relatively high variability

usually observed in *in vivo* studies. Nonetheless, the differences between SELP shRNA and both controls (WT and Negative Control shRNA) are statistically significant and observed when using different methods for SELP inhibition and in different GB models. Please note that now we show this experiment in new Figure 5.

Also, since the shRNA expression is not conditional, and SELP knockdown blocks tumor cell proliferation, how can the authors rule out the observed effects on tumor size being a result of few viable cells?

First, the same amount of viable GB cells was injected. SELP inhibition delayed GB cell proliferation but did not induce apoptosis as shown now in Supplementary Figure 6 using SELP inhibitor (SELPi). Second, we showed that SELP knockdown results in **slightly lower** proliferation rate of GB cells. However, as can be seen in Figure 2 and now in Supplementary Figure 5, a much stronger inhibition of cell proliferation was observed when GB cells were co-cultured in the presence of microglia than when they were grown as GB mono-culture. In addition, we mentioned in the Discussion section that SELP seems to have a role in GB tumorigenesis affecting both intrinsic GB cell properties and the interaction with the microenvironment similar to other oncogenic factors which are involved in several oncogenic pathways. We extensively showed the effect of SELP on microglia phenotype and the effect of its inhibition on the brain microenvironment. This is the first time any of these roles of SELP in GB are reported.

Fig. 5: Approximately 14 clusters are shown in Fig 5A. Most are identified by the cell type. However, some are indicated by numbers. Can the authors please clarify which of these cell types are shown in Fig. 5B, which has only 8 bars. Likewise, please label the clusters in Fig. 5C with the cell type ID. Please include a table to show the top 10 most significantly upregulated or downregulated genes for each cluster so that there is a clearer picture of all of the cell types that are affected by SELP knockdown. It is acknowledged that the authors' focus is on angiogenesis, tumor cell proliferation and microglial biology.

In Figure 6A (previously called Figure 5) we show all the different cell types that were identified. Few small clusters exhibited differential gene expression but we could not determine their specific cell type identity. As per the reviewer's request, we have now provided additional heat map with the top 10 most significantly upregulated genes for each cluster (Supplementary Figure 13C). To analyze the tumor cells cluster, we further classified it to sub-clusters. Figure 6B shows the different sub-clusters identified within the tumor cell cluster (shown in Supplementary Figure 14). Figure 6C shows the most upregulated genes in each tumor sub-cluster (corresponding to Figure 6B).

Supplementary Figure 13. Cell type specific gene expression and population distribution of different cell types between shNC and shSELP tumors, identified in whole-tumor single-cell RNA-seq. C. Heat map showing the ten most up-regulated differentially expressed genes in each cluster

Fig. 6: On page 32, the authors make the comment that: “We note that caspase staining increased in shSELP tumors only in immunocompetent mice. This phenomenon could be correlated with enhanced infiltration of CD8+ T cells suggesting that these cells play a cytotoxic role in shSELP GB”. Can the authors please add a figure number for this comment?

This statement refers to Figure 5 and Supplementary Figure 12. We have now added figure number in the comment. The comment was now changed to:

“We note that caspase staining increased in shSELP tumors only in immunocompetent mice (Figure 5C). This phenomenon could be correlated with enhanced infiltration of CD8+ T cells (Supplementary Figure 12C, D) suggesting that these cells play a cytotoxic role in shSELP GB”.

Fig. 7: Can the authors clarify why the intraventricular and systemic administration of SELP inhibitor was not done with the same GBM model?

The use of intraventricular injection was tested on PD-GB4 to show the effect of SELPi in a patient-derived model *in vivo*. Now we have added an *in vivo* study using SELP knockdown (shSELP) PD-GB4 cells compared negative control (shNC) PD-GB4 cells and wild type PD-GB4 cells. Most of the *in vivo* experiments were initially performed using U251 and GL261 as they are commonly used GB models treated systemically with different compounds, in which this effect can be compared and replicated in other laboratories.

To summarize, we are now showing *in vivo* experiments using (i) shSELP models of all three GB cell types: PD-GB4 (patient-derived), GL261 (murine cell line), and U251 (human cell line), (ii) Systemic administration of SELPi in three GB cell types: GL261 (murine cell line), U251 (human cell line), iAGR53 (murine lenti-induced cells), and (iii) locally delivered (intraventricularly) SELPi using cannulas in PD-GB4. This experiment is incredibly expensive and time consuming as each mouse is housed in individual metabolic cage as otherwise, if caged together, mice compromise the integrity of the cannulas and tear them off.

On page 34 and elsewhere, references have been made regarding published work describing SELP inhibition and T cell trafficking. Please cite the reference(s).

We thank Reviewer #1 for pointing this out. We now added the following references:

82. Angiari, S. and G. Constantin, *Regulation of T cell trafficking by the T cell immunoglobulin and mucin domain 1 glycoprotein*. Trends Mol Med, 2014. **20**(12): p. 675-84.
83. Ali, A.J., A.F. Abuelela, and J.S. Merzaban, *An Analysis of Trafficking Receptors Shows that CD44 and P-Selectin Glycoprotein Ligand-1 Collectively Control the Migration of Activated Human T-Cells*. Frontiers in Immunology, 2017. **8**(492).
84. Sperandio, M., C.A. Gleissner, and K. Ley, *Glycosylation in immune cell trafficking*. Immunological reviews, 2009. **230**(1): p. 97-113.

CD31 staining in SELPi treated samples does not seem to very different from controls in Fig. 7F. Can the authors include a section that is more reflective of the graphical data? Please include the following data for PD-GB4 tumors in Fig. 7G: H&E, Iba1, CD31, Ki67 staining.

We have now selected more representative images for CD31 staining in Figure 7F, which reflect the average staining for each group (Figure 7F). We have stained PD-GB4 tumors presented in this figure for H&E, Ki67, Iba1 and CD31 (Figure 7H) and added this to Figure 7.

Figure 7. SELPi delays the growth of murine and human GB tumors in mice. F. Quantification and representative images of immunostaining for blood vessels (CD31) in the tumors. Results demonstrate reduced blood vessel density in SELPi treated tumors. Data represent mean \pm s.d. N = 3 images per mouse, 3 mice per group.

Figure 7. SELPi delays the growth of murine and human GB tumors in mice. H. H&E staining of PD-GB4 tumors showing reduced invasion in SELPi treated PD-GB4 tumors. Quantification and representative images of immunostaining for proliferating cells (Ki-67), activated microglia (Iba1), and blood vessels (CD31) in the tumors. Results demonstrate reduced proliferation and blood vessel density, and activated microglia in SELPi treated tumors. Data represent mean \pm s.d. N = 3 images per mouse, 2 mice per group.

Discussion:

We thank the reviewer for the suggestions. They greatly improved the Discussion section of our manuscript.

1. Please expand on whether soluble versus membrane bound SELP may have unique contributions to GBM pathogenesis. Can one work as a decoy, for example?

We think that SELP binding to PSGL-1 on microglia/macrophages affect their phenotype regardless to its form. We concluded this as we show the effect of rSELP on microglia/macrophages in the absence of GB cells as well as in their presence. Since we observed that SELP mediates both GB cell invasion and immunosuppression, we hypothesized that the membrane bound SELP has a role in GB invasion while microglia/macrophages phenotype is mainly affected by sSELP. We did discuss this point in the Discussion section and now we further elaborated:

“Importantly, we observed that GB cells not only expressed SELP on the membrane but also secreted the soluble protein (sSELP). Moreover, SELP had an immunosuppressive effect on microglia/macrophages in the presence and in the absence of GB cells when grown in culture. This indicates that SELP-PSGL-1 axis may be activated in microglia/macrophages by membrane-bound SELP and sSELP, and that both forms may have a role in microglia/macrophages phenotype not only in the context of GB. Indeed, it was shown that sSELP can be found in both the monomeric and the dimeric forms in human plasma (112) and that both forms can bind PSGL-1 expressed on neutrophils and monocytes (113-114). Although the exact role of sSELP has yet to be elucidated, increased plasma levels of sSELP are known to be associated with certain types of cancer (115-120). However, neither its role nor its effect was shown in the brain, in general, nor in GB, in particular, until now. This raises the question as to whether the soluble and membrane forms of SELP function in the same way in GB progression. One speculation could be that one form, SELP, contributes to GB invasion due to its adhesion properties, while the other, sSELP, contributes to microglia/macrophages suppression due to its higher exposure to the neighboring stromal cells.”

112. Woollard, K.J., et al., Raised plasma soluble P-selectin in peripheral arterial occlusive disease enhances leukocyte adhesion. *Circ Res*, 2006. 98(1): p. 149-56.
113. Mehta, P., et al., Soluble monomeric P-selectin containing only the lectin and epidermal growth factor domains binds to P-selectin glycoprotein ligand-1 on leukocytes. *Blood*, 1997. 90(6): p. 2381-9.
114. Conde, I.d., et al., Effect of P-Selectin on Phosphatidylserine Exposure and Surface-Dependent Thrombin Generation on Monocytes. *Arteriosclerosis, Thrombosis, and Vascular Biology*, 2005. 25(5): p. 1065-1070.
115. Majchrzak-Baczmanska, D.B., et al., Serum concentrations of soluble (s)L- and (s)P-selectins in women with ovarian cancer. *Prz Menopauzalny*, 2018. 17(1): p. 11-17.
116. Lemancewicz, D., et al., Bone marrow megakaryocytes, soluble P-selectin and thrombopoietic cytokines in multiple myeloma patients. *Platelets*, 2014. 25(3): p. 181-7.
117. Grafetstätter, M., et al., Plasma Fibrinogen and sP-Selectin are Associated with the Risk of Lung Cancer in a Prospective Study. *Cancer Epidemiol Biomarkers Prev*, 2019. 28(7): p. 1221-1227.
118. Graf, M.E., et al., Pre-diagnostic plasma concentrations of Fibrinogen, sGPIIb/IIIa, sP-selectin, sThrombomodulin, Thrombopoietin in relation to cancer risk: Findings from a large prospective study. *Int J Cancer*, 2018. 143(11): p. 2659-2667.
119. Komala, A.S. and A. Rachman, Association of Peripheral Monocyte Count with Soluble P-Selectin and Advanced Stages in Nasopharyngeal Carcinoma. *Adv Hematol*, 2018. 2018: p. 3864398.
120. Grilz, E., et al., Association of complete blood count parameters, d-dimer, and soluble P-selectin with risk of arterial thromboembolism in patients with cancer. *J Thromb Haemost*, 2019. 17(8): p. 1335-1344.

2. Please provide more details on the relationship between selectins and microglia in neurodegenerative diseases and autoimmune diseases and describe the novelty of your findings in the context of what has been shown in the above systems.

It was reported that PSGL-1 levels in microglia were reduced in EAE model of multiple sclerosis, and reduced plasma levels of SELP and L-Selectin were detected in AD patients (135-136). However, the role and the mechanisms in which SELP is involved in neurodegenerative and autoimmune disorders remain unknown. Our findings on the importance of SELP-PSGL-1 crosstalk in cancer may contribute to elucidate the role of microglia in neurodegenerative diseases as well.

We have now added this paragraph to the Discussion section in the manuscript.

135. Corsi, M.M., et al., *Reduced plasma levels of P-selectin and L-selectin in a pilot study from Alzheimer disease: relationship with neuro-degeneration*. *Biogerontology*, 2011. 12(5): p. 451-4.
136. Jordão, M.J.C., et al., *Single-cell profiling identifies myeloid cell subsets with distinct fates during neuroinflammation*. *Science*, 2019. 363(6425): p. eaat7554.

3. Please expand on your discussions regarding the therapeutic potential of SELPi for GBMs. Can they be used as single agents or in combination with immunotherapy or standard of care? Is anything known about radiation or chemotherapy altering the levels of SELP and contributing to the immunosuppressive microenvironment?

We have added this part to our Discussion regarding the therapeutic potential of SELPi:

“Since inhibition of SELP did not cause direct tumor-killing, combining anti-SELP treatment with the standard chemotherapy (TMZ) may be beneficial. Radiation therapy was shown to increase SELP expression on endothelial cells on the lumen of angiogenic vessels (38, 138), which may affect the phenotype of recruited BMDM. Thus, SELP inhibition may also improve the efficacy of radiotherapy. Moreover, as we show that SELP mediates the secretion of T cells suppressing cytokines, and that its inhibition facilitates the infiltration of CD8 positive T cells into the tumor, anti-SELP treatment may improve the susceptibility of GB tumors to existing immunotherapies, hence sensitizing this non-responsive tumor to become immune checkpoint therapy (ICT)-responsive.”

38. Shamay, Y., et al., P-selectin is a nanotherapeutic delivery target in the tumor microenvironment. *Sci Transl Med*, 2016. 8(345): p. 345ra87.
138. Hallahan, D.E., et al., *X-ray-induced P-selectin localization to the lumen of tumor blood vessels*. *Cancer Res*, 1998. 58(22): p. 5216-20.

Reviewer #2, expert in neuroimmunology/trafficking (Remarks to the Author):

Glioblastoma is a highly lethal brain tumor and more effective therapeutic strategies are needed. In this paper the authors investigated the molecular mechanisms controlling the interplay between the local immune system and glioblastoma (GB) cells. They show a role for P-selectin, a molecule classically involved in leukocyte trafficking and platelet activation, in microglia-tumor cell interactions and GB progression. A role for P-selectin has been previously suggested in the context of other oncological diseases. However, the role of this molecule and its main ligand PSGL-1 is unclear in GB pathogenesis, and the authors comprehensively show that P-selectin expression on human and mouse GB cells modulates microglia activation and T cell recruitment. The usage of a P-selectin inhibitor delays the growth of murine and human GB tumors in mice. These results suggest that P-selectin blockade interferes with microglia phenotype and may represent a new therapeutic approach for GB. The manuscript is well-written and the results are interesting and overall convincing. However, the study raises some concerns that need to be addressed.

We thank Reviewer #2 for the insightful and supportive comments.

1. It is unclear from Fig. 1 if GB also induces an increase of microglia density, which represents a common sign of microglia activation. Also, how many patients (FFPE samples) were studied for microglia activation in Fig. 1a?

We have now performed co-culture proliferation assay using GFP-labeled microglia and PD-GB4 cells, and a TransWell migration assay evaluating the migration of human microglia toward PD-GB4 cells. The results show enhanced proliferation and migration of human microglia in the presence of GB cells, indicating the activation of microglia following the interaction with GB cells in our models (Supplementary Figure 1A-B), suggesting that GB cells may induce microgliosis in the brain as a result of the neuroinflammation induced by their crosstalk. We have now added these results to the Supplementary Information (Supplementary Figure 1A-B).

For microglia activation in human samples we used three different patient FFPE samples for this analysis. However, we have 70 FFPE GB samples that show microglia activation by gene expression and proteomics signatures.

Supplementary Figure 1. GB cells enhance the proliferation and migration rate of human microglia cells and the Isolated murine CD11b-positive cells and purchased human microglia are positive for microglia specific markers. A. Proliferation of GFP-labeled human microglia, showing enhanced proliferation of microglia cells when co-cultured with PD-GB4 GB cells (ratio of 1:1) compared to microglia monoculture. Images and analysis were obtained using IncuCyte instrument. Data represent mean \pm s.d. of triplicate wells. The graph is a representative of three independent repeats. **B.** Results and representative images of TransWell migration assay showing enhanced migration of human microglia cells towards PD-GB4 GB cells compared to naïve DMEM. Data represent mean \pm s.d. of triplicate wells. The graph is a representative of three independent repeats.

2. In line with point 1, do the authors have evidence that GB cells may recruit blood myeloid cells and increment brain macrophage/microglia cells?

The presence of activated microglia/macrophages in GB was shown by us and by others. It was also shown that GB cells recruit microglia and BMDM by the secretion of various chemokines such as CCL2 (see references below). In addition, the results shown above (Supplementary Figure 1A-B) indicate the ability of GB cells to attract and activate microglia cells resulting in higher proliferation and migration rates of microglia.

Charles, N.A., et al., The brain tumor microenvironment. *Glia*, 2011. 59(8): p. 1169-80.

Maas, S.L.N., et al., Glioblastoma hijacks microglial gene expression to support tumor growth. *Journal of Neuroinflammation*, 2020. 17(1): p. 120.

Gutmann, D.H. and H. Kettenmann, Microglia/Brain Macrophages as Central Drivers of Brain Tumor Pathobiology. *Neuron*, 2019. 104(3): p. 442-449.

3. Figure 1K shows the quantification of immunostaining of SELP suggesting more positive staining in short-term survivors (STS) GB patient FFPE samples compared to long-term survivors (LTS) or normal human brain tissue. However, this reviewer could not find a description of the method used for data quantification.

As for all immunostaining, stained samples were imaged using the EVOS FL Auto cell imaging system. At least three fields of each individual sample were imaged and quantified using ImageJ software. Quantification of positive staining was performed by measuring the total area stained in each image following background subtraction, using single color images representing the correlated marker. We have now written this description in detail in the immunostaining part of the Methods section.

4. The role of P-selectin-PSGL-1 axis is not well demonstrated in this paper. rP-selectin induces an M2-like phenotype in microglia cells in vitro (Figure 3). However, in this experimental setting, the authors assume that the effect of rP-selectin is mediated by PSGL-1 expressed by microglial cells. It would be important to more clearly demonstrate that the induction of M2 phenotype is mediated by P-selectin-PSGL-1 interactions and/or if other P-selectin ligands may be involved. The assays performed in the presence of a blocking anti-PSGL-1 antibody or using PSGL-1 deficient microglial cells would be helpful to clarify this aspect.

We are indebted to Reviewer #2 for the suggestions regarding the addition of PSGL-1 neutralizing antibody. It added clarity to the mechanism by which SELP affects microglia/macrophages immunophenotype.

In order to answer the reviewer's question, we have now performed the following experiments using rSELP, SELPi and neutralizing antibodies for PSGL-1 and for the additional SELP ligands, CD44 and CD24 (Figure 3F-K, Supplementary Figure 9A-B, Supplementary Figure 10E-F):

- Evaluation of AGR-1 and iNOS expression by human microglia and IL-10 and TGF- β expression by human and murine microglia (Figure 3F-K, Supplementary Figure 9A-B).
- Evaluation of nitric oxide release by microglia and BMDM (Supplementary Figure 10E).
- Evaluation of the phagocytic abilities of microglia and BMDM (Supplementary Figure 10F).

Our findings demonstrate that SELPi or anti-PSGL-1 neutralizing antibody had the ability to revert the effects of rSELP treatment, while anti-CD44/CD24 neutralizing antibodies did not or only slightly affected rSELP treatment.

Figure 3. SELP inhibits the microglia pro-inflammatory phenotype and promotes their immunosuppressive activity. **F.** Real-Time PCR showing higher expression of ARG1 by human microglia when treated with rSELP compared to untreated. ARG1 expression was reduced when SELPi or anti-PSGL-1 neutralizing antibody were added but not when anti-CD44 or anti-CD24 neutralizing antibodies were added. Data represent mean + s.d. The graph shows the average of three independent studies. **G.** Real-Time PCR showing lower expression of iNOS by human microglia when treated with rSELP compared to untreated. iNOS expression was elevated when SELPi or anti-PSGL-1 neutralizing antibody were added but not when anti-CD44 or anti-CD24 neutralizing antibodies were added. Data represent mean + s.d. The graph shows the average of three independent studies. **H.** Real-Time PCR showing higher expression of IL-10 by human microglia when treated with rSELP compared to untreated. IL-10 expression was reduced when SELPi or anti-PSGL-1 neutralizing antibody were added but not when anti-CD44 or anti-CD24 neutralizing antibodies were added. Data represent mean + s.d. The graph shows the average of three independent studies. **I.** Real-Time PCR showing higher expression of TGF-β by human microglia when treated with rSELP compared to untreated. TGF-β expression was reduced when SELPi or anti-PSGL-1 neutralizing antibody were added but not when anti-CD44 or anti-CD24 neutralizing antibodies were added. Data represent mean + s.d. The graph shows the average of three independent studies. **J.** Latex beads engulfment by human microglia, demonstrating reduced phagocytic activity of human microglia treated with rSELP compared to untreated. Phagocytosis activity was restored when SELPi or anti-PSGL-1 neutralizing antibody were added but not when anti-CD44 or anti-CD24 neutralizing antibodies were added. Data represent mean ± s.d. The graph is a representative of three independent repeats. **K.** NO release by LPS stimulated human microglia was reduced when treated with rSELP, and was restored when SELPi or anti-PSGL-1 neutralizing antibody were added but not when anti-CD44 or anti-CD24 neutralizing antibodies were added. NO levels in the medium were analyzed by a total NO detection kit. Data represent mean ± s.d. of triplicate wells. The graph is a representative of three independent experiments.

Supplementary FIG 9. SELP mediates the expression of IL-10 and TGF-β and the phagocytosis abilities of murine microglia. A-B. Real-time PCR showing induction of IL-10 and TGF-β expression by murine microglia when treated with rSELP. IL-10 and TGF-β expression was reduced when SELPi or anti-PSGL-1 neutralizing antibody were added but not when anti-CD44 or anti-CD24 neutralizing antibodies were added. Data represent mean + s.d. Each graph is representative of three independent repeats.

Supplementary Figure 10. SELP mediates BMDM immunophenotype. E. Phagocytic activity of BMDM was reduced following treatment with rSELP and GB CM. This activity was restored when SELPi or anti-PSGL-1 neutralizing antibody were added but not when anti-CD44 or anti-CD24 neutralizing antibodies were added. Data represent mean ± s.d. The graph is a representative of three independent repeats. **F.** NO release by LPS stimulated BMDM was reduced when treated with rSELP, and was restored when SELPi or anti-PSGL-1 neutralizing antibody were added but not when anti-CD44 or anti-CD24 neutralizing antibodies were added. NO levels in the medium were analyzed by a total NO detection kit. Data represent mean ± s.d. of triplicate wells. The graph is a representative of three independent experiments.

In addition, is PSGL-1 crosslinking in microglial cells inducing similar effects as soluble rP-selectin? It was previously reported that both the monomeric and the dimeric (which crosslinks PSGL-1) forms of SELP can efficiently bind to PSGL-1 expressed by neutrophils and monocytes. This may suggest that crosslinking of PSGL-1 might have similar effects as rSELP also on microglia/macrophages.

Mehta, P., *et al.*, Soluble monomeric P-selectin containing only the lectin and epidermal growth factor domains binds to P-selectin glycoprotein ligand-1 on leukocytes. *Blood*, 1997. 90(6): p. 2381-9.

Conde, I.d., *et al.*, Effect of P-Selectin on Phosphatidylserine Exposure and Surface-Dependent Thrombin Generation on Monocytes. *Arteriosclerosis, Thrombosis, and Vascular Biology*, 2005. 25(5): p. 1065-1070.

5. In some experiments GB-CM have been used either alone or in combination with SELP neutralizing antibody or SELP inhibitors. However, P-selectin shedding may be variable and no direct evidence showing the content of P-selectin in GB-CM is shown (Figure 3 C, D, F, G, N).

In order to generate GB CM for these experiments, GB cells were grown in serum-free medium for 24 – 48 h. We evaluated SELP secretion in this GB CM (named PD-GB4 in Figure 1E). Although higher secretion was observed when treated with microglia CM (named PD-GB4 + MG CM in Figure 1E), sSELP was detected in the GB CM control as well, suggesting a basal secretion of SELP by PD-GB4 when grown in serum-free conditions. We show here that we have detected a secretion level of 1.7 $\mu\text{g/ml}$ of SELP by 1×10^6 PD-GB4 cells cultured for 24 h in serum-free medium (Figure 1E), while in the manuscript we showed the fold change. In addition, we have shown that treatment with GB CM facilitates the secretion of SELP by human microglia (MG) as well (Figure 3). Thus, adding SELPi to GB CM blocks both the direct and the indirect effects of GB CM mediated by SELP.

Figure 1. Microglia facilitates the proliferation and migration rate of GB cells and enhances the expression of SELP by GB cells. E. ELISA results showing over-secretion of SELP by PD-GB4 cells treated with microglia CM (MG CM) compared to naïve microglia medium. Data represent mean \pm s.d. The graph shows the average of three independent studies. **Results are presented as $\mu\text{g/ml}$ SELP secretion.**

Minor points

1. “SELP” is more commonly used to indicate the P-selectin encoding gene. “P-selectin” or “CD62P” would be more indicated to refer to the protein.

We can of course change to P-selectin if requested, however, this nomenclature is also commonly used and takes less space in the figure legends, especially as we are showing interchangeably both gene expression by real time PCR or protein levels by ELISA or FACS. We are certain that this is clear to all readers.

2. The size of scale bars in all immunofluorescence images is unclear and should be mentioned in figure legend.

We have now added the size of the scale bars to all relevant Figure legends.

3. A more detailed description of the stereotactic GB cell injections should be provided (e.g. brain coordinates, injected volume).

We have stereotactically injected 50×10^3 GB cells in $5 \mu\text{l}$ at $1 \mu\text{l/min}$ flow rate. Injection coordination relative to the mouse Bregma point were: 2 mm lateral (left), 0.5 mm anterior, 3.5 mm ventral. We have now added this information in the “Animal models” part of the Methods section.

4. SELPi dosage and the vehicle need to be specified on page 15 (“To examine the effect of SELPi treatment in the murine GL261 GB model, GL261 cells (5×10^4 cells) were stereotactically implanted... and treated with SELPi three times a week”).

We specified this in the beginning of page 15 when describing the first experiment using SELPi: “SELPi (0.8 mg/ml) was dissolved in DMSO (0.01%), polyethyleneglycol (93.2 mg/ml), Tween-80 (14.8 mg/ml), and DDW. Mice were treated intravenously (IV) with PBS, 16 mg/kg SELPi or vehicle (0.01% DMSO, 93.2 mg/ml polyethyleneglycol and 14.8 mg/ml Tween-80 in DDW).”

We have now added a section describing the SELPi compound in details and its use in this study at the end of the “Animal models” part of the Methods section.

5. The number of the supplementary figure showing flow cytometry data is lacking on page 28 “(Supplementary Figure C, D)”.

We have now corrected this mistake. This statement is now referred to Supplementary Figure 12C-D.

6. Page 29, Figure 4, panel C: the representative MRI images are not mentioned.

We have now updated this Figure legend.

7. Page 34: there are some typing errors in “(Figure 7A C, D)- C”.

We have now corrected this mistake.

8. Supplementary figure legend 3, page 50: the identification of the panels from “C” to “F” have some errors.

We have now corrected this Figure legend.

9. Supplementary figure 5, page 52: “C” is missing in the figure panel.

We have now corrected this Figure panel.

We apologize for these oversights. We have now systematically corrected typos and incorrect labeling issues.

Reviewer #3, expert in microglia/macrophages in GBM and single cell sequencing (Remarks to the Author):

This is an exciting and excessive work that identifies P-selectin as an important regulator of GBM associated microglia phenotype and microglia-tumor interaction. I acknowledge the work is extreme and has a limitless number of supplementary figures; I can only imagine how much work went into it. But unfortunately, the way it is presented, it is tough to read and follow it. I do have some serious concerns about the experimental design and quality of some of the important data.

We thank Reviewer #3 for appreciating the quantity and quality of our findings. We have ameliorated our manuscript based on the constructive critique as detailed herein.

In vitro work is based only on microglia and tumor cell cultures; in reality, in vivo context, there is an immense contribution of bone-marrow-derived macrophages that are different from microglia. Authors should use BMDM cultures and see whether the effects are microglia specific or, in general, myeloid-specific.

In order to broaden our characterization of the BMDM phenotype in response to the important question of the reviewer, we have now added the evaluation of nitric oxide release by BMDM, and showed that the inhibition of the phagocytic abilities and nitric oxide release caused by rSELP treatment, is mediated by SELP-PSGL-1 axis compared to other SELP ligands (new Supplementary Figure 10F). This is in addition to the use of BMDM cultures showing the expression of PSGL-1, CD206 and CD38 in response to SELP and/or SELP inhibition (new Supplementary Figure 10). In addition, our single-cell analysis which showed the changes in shSELP tumors compared to shNC, includes both microglia and macrophages clusters. We now have comprehensive data showing the vital role of SELP on both microglia and BMDM phenotype in GB.

Supplementary Figure 10. SELP mediates BMDM immunophenotype. **A.** Flow cytometry analysis showing that freshly isolated BMDM are positive for the macrophages markers CD11b and F4/80. **B.** PSGL-1 expression by BMDM was increased following treatment with rSELP and GB CM, and was reduced when SELPi was added to GB CM. **C-D.** BMDM expressed higher levels of the anti-inflammatory marker CD206 (C) and lower levels of the pro-inflammatory marker CD38 (D) when treated with rSELP compared to untreated cells. CD206 expression was reduced while CD38 expression was elevated when SELPi was added to rSELP treatment. **E.** Phagocytic activity of BMDM was reduced following treatment with rSELP and GB CM. This activity was restored when SELPi or anti-PSGL-1 neutralizing antibody were added but not when anti-CD44 or anti-CD24 neutralizing antibodies were added. Data represent mean \pm s.d. The graph is a representative of three independent repeats. **F.** NO release by LPS stimulated BMDM was reduced when treated with rSELP, and was restored when SELPi or anti-PSGL-1 neutralizing antibody were added but not when anti-CD44 or anti-CD24 neutralizing antibodies were added. NO levels in the medium were analyzed by a total NO detection kit. Data represent mean \pm s.d. of triplicate wells. The graph is a representative of three independent experiments.

Many tumor lines are used both human and mouse, but unfortunately, they are used back and forth for various parts of one story. There is a big statement of using different murine lines PN, MES, and CL, but they appear best in some subfigures, so do some of the human lines and primary cultures. Most of the story really on GL-261 and U251. These extra lines should be removed, or authors should do all the key experiments using all these lines. All these lines have different genotypes, and they will affect myeloid cells differently, which is apparent even from low-quality representative images for IBA1.

We agree with the reviewer that different GB cells may affect differently glioma-associated microglia/macrophages (GAMs) phenotype and *vice versa*. Thus, we show the key role of SELP in multiple GB models exhibiting different phenotypes. Although showed some differences in the levels of GAMs activation, SELP expression and response to treatments, all the models used showed the key involvement of SELP in GB progression.

Most of the story relies on PD-GB4, U251 and GL261 cell lines. In order to be consistent, all the *in vitro* experiments in the main figures using PD-GB4 cells, while key *in vitro* experiments were reproduced using U251 and GL261 which are now shown in the Supplementary Information. When relevant, some of the experiments were performed using additional PD-GB cells and murine GB cell lines.

We have included the proneural- PN (PNp53), mesenchymal- MES (iAGR53) and classical- CL (EGFRviii-shP16) murine GB cell lines to show the relevance of SELP in GB cells-GAMs interactions in the different GB molecular subtypes in three additional immunocompetent murine GB models, besides the GL261 model. As GB different subtypes react differently to anti-cancer treatments, it is important to show that SELP has a key role in GB cells-GAMs interactions mediating GB cell invasion (spheroid assays; Supplementary Figure 7) and tumor growth in these three subtypes (Supplementary Figure 7, Figure 7).

We now demonstrate the effect of inhibiting SELP in multiple *in vivo* experiments using PD-GB4, U251 and GL261 for SELP knockdown (by shRNA) and PD-GB4, U251, GL261 and iAGR53 for SELPi treatments. Thus, we show the role of SELP in GB progression *in vitro* and *in vivo*, in multiple human and murine GB models representing GB genetic and phenotypic diversity.

Of note- we originally added murine, human and patient-derived GB cells interchangeably as we were asked to demonstrate SELP's role in multiple GB models to verify that this SELP-PSGL-1 crosstalk is not a peculiarity of one single GB cell line/cell type.

All the representative images are of low quality. Double and triple staining are hard to see. P-selectin staining should be double-stained with staining specific for tumor cells, GAMs, and endothelial cells.

We have now added all the figures separately in order to improve the quality of the images and prevent the decrease in resolution due to multiple document transfers. Figure 1I shows double staining for SELP and activated microglia/macrophages (Iba1) in GB tumors in which the cancer cells are labeled in red. We included all the separate channels of the staining shown in Figure 1 in the Supplementary Information (Supplementary Figure 3A-D). When examining the staining of SELP or PSGL-1 together with Iba1 and labeled GB cells, it can be seen that SELP staining correlates mainly with the tumor cells and the blood vessels, while PSGL-1 staining correlates mainly with Iba1. We have now added co-staining for SELP and CD31 (Supplementary Figure 3D). As expected, results show positive staining of SELP co-localized with endothelial cells (CD31). We note that in order to observe SELP expression by the mouse endothelial cells, we used anti-mouse SELP antibody. Thus, when we stained human-derived GB tumors, we detected positive staining only on the blood vessels (murine stromal endothelial cells), while staining of murine GB tumors detected positive staining for SELP which correlated with both murine cell types, the stromal CD31 and the labeled GL261 GB cells.

Supplementary Figure 3. Immunostaining of GB mouse models. A-B. Immuno-fluorescence staining presented at separate fluorescence channels, showing the co-staining for Iba1 and SELP/PSGL-1 in different GB mouse models shown in Figure 1 and in two additional PDX GB mouse models. **C, D.** Immuno-fluorescence staining presented at separate fluorescence channels, showing co-staining for Iba1 + Ki-67 (C) and SELP + CD31 (D) in U251, PD-GB4 and GL261 GB mouse models.

MRI images show some extracranial tumor growth, which should be clarified, and better-quality images should be provided.

As mentioned above, we now added all the figures separately to improve the quality of the images. Some of the fast-growing tumors, such as those of the control groups, showed extracranial growth at advanced stages of tumor development. This was observed only in a few cases and the extracranial part was not included in tumor size quantifications.

The introduction would benefit from more focus and a clear message. The microglia and microglia/macrophages are used interchangeably, which is confusing and not appropriate. Claiming all the effects are microglia driven is not reasonable without the use of BMDM.

We have now referred only to microglia/macrophages or GAMs, except when describing results which were observed only in one cell type. In addition, although focusing on microglia, we did not claim that the effects are only microglia-driven, but the contrary, we show key experiments evaluating GAMs phenotype using BMDM (Supplementary Figure 10). As mentioned above, we have now added the evaluation of nitric oxide release by BMDM in addition to phagocytosis evaluation and the expression of immunophenotype-related markers (Supplementary Figure 10F shown above). Our results indicate that SELP-PSGL-1 axis has immunosuppressive effects on both microglia and macrophages immunophenotypes.

Treatment with inhibitor needs more clarification and information; in addition, more evidence is required to show target inhibition and specificity.

SELPi was chemically and biologically characterized by the manufacturers, showing specificity and potency (https://www.tocris.com/products/kf-38789_2748). In addition, it was used in various studies including cancer metastasis research (Li, S.-S., et al., Sialyl Lewisx-P-selectin cascade mediates tumor–mesothelial adhesion in ascitic fluid shear flow. *Nature Communications*, 2019. 10(1): p. 2406.). Nonetheless, we have used complementary methods of SELP inhibition such as anti-SELP neutralizing antibody and SELP shRNA to validate that the effects observed are indeed mediated by SELP. To provide more information on SELPi target inhibition and specificity, and to clarify the description of our use of SELPi treatment, we have now added the following information to the Methods section:

“SELPi treatments. KF 38789 (SELPi) is a commercially-available P-Selectin inhibitor manufactured by Tocris BioScience. SELPi was shown to be a selective P-Selectin inhibitor (IC₅₀ = 1.97 μM) with no inhibiting effects on L-Selectin or E-Selectin. It was shown to block SELP-binding in vitro and in vivo [43-45]. This compound is commonly used for research of immune-related processes such as tumor metastases formation. For the experiment described above, SELPi (0.8 mg/ml) was dissolved in DMSO (0.01%), polyethyleneglycol (93.2 mg/ml), Tween-80 (14.8 mg/ml) and DDW. Mice were injected with 16 mg/kg IV, twice a week, three times a week or QOD as described above. As control (vehicle), mice were injected with DMSO (0.01%), polyethyleneglycol (93.2 mg/ml), Tween-80 (14.8 mg/ml), in DDW at equivalent volume.”

43. Haggarty, S.J., et al., Dissecting cellular processes using small molecules: identification of colchicine-like, taxol-like and other small molecules that perturb mitosis. *Chem Biol*, 2000. 7(4): p. 275-86.

44. Ohta, S., et al., Inhibition of P-selectin specific cell adhesion by a low molecular weight, non-carbohydrate compound, KF38789. *Inflamm Res*, 2001. 50(11): p. 544-51.

45. Kaila, N., et al., Synthesis and biological evaluation of quinoline salicylic acids as P-selectin antagonists. *J Med Chem*, 2007. 50(1): p. 21-39.

46. Zhang, Y., et al., Prostaglandin E2 receptor 4 mediates renal cell carcinoma intravasation and metastasis. *Cancer letters*, 2017. 391: p. 50-58.

47. Li, S.-S., et al., Sialyl Lewisx-P-selectin cascade mediates tumor–mesothelial adhesion in ascitic fluid shear flow. *Nature Communications*, 2019. 10(1): p. 2406.

48. Zhao, Z., et al., Deletion of junctional adhesion molecule A from platelets increases early-stage neointima formation after wire injury in hyperlipidemic mice. *Journal of cellular and molecular medicine*, 2017. 21(8): p. 1523-1531.

49. Krishnamurthy, V.R., et al., Glycopeptide analogues of PSGL-1 inhibit P-selectin in vitro and in vivo. *Nature Communications*, 2015. 6(1): p. 6387.

There is a serious attempt to claim M2 versus M1 changes in the macrophage profile. Authors should focus on signaling and leave M1 and M2 states out, since they fail to emerge as isolated pure phenomena in vivo, especially in GBM.

We agree with the reviewer that the *in vivo* scenario is much more complex, thus, we have now referred to GAMs phenotypes as a more pro-inflammatory/anti-tumorigenic or anti-inflammatory/pro-tumorigenic phenotypes instead of M1/M2-like. In addition, the main figure evaluating GAMs phenotype is now focusing on signaling, cytokines expression and functional assays (Figure 3). Evaluation of the additional immune-related markers are now presented in the Supplementary Information (Supplementary Figure 8, Supplementary Figure 10).

Reviewer #4, expert in single cell sequencing and cancer (Remarks to the Author):

Yeini et al aim to investigate the crosstalk of microglia with glioma cells and how this impacts glioma cell proliferation and invasion.

We thank the reviewer for the constructive comments. We first responded to some of the points raised and following below we have added the new experiments and findings and detailed the reorganization of the manuscript to make the story clearer and coherent.

1. They first use a commercially available and patient-derived microglia and glioma cells and show that glioma proliferation and migration is enhanced in the presence of microglia. They use a protein array to check for differences in **murine** co-cultures vs. glioma vs. microglia supernatants and find that a few are differentially regulated in the co-culture, and find among others SELP (P-selectin) to be higher in co-culture. They then measure mRNA and protein in a separate human model incubated with CM of microglia. Staining for Iba1, SELP and PSGL-1 on **mouse** FFPE healthy brain and malignant tissue.

Of note- We now added protein array of **human** co-cultures vs. glioma vs. microglia supernatants.

We also show stained FFPEs of healthy and GB **human** samples.

2. Co-culture of cancer cells with SELP KD results in impaired proliferation and prolonged wound healing time compared to WT cells or scramble KD controls. The authors conclude that the microglia interactions are necessary for increased proliferation/migration.

3. Vice versa, glioma CM incubation of microglia results in increased SELP expression/secretion and upregulation of **PSGL-1 in scRNA-seq data**. TCGA analysis finds an association of PSGL-1 and CD4. Analysis of other single-cell data shows PSGL-1 expression in TILs.

Of note- we have shown the upregulation of PSGL-1 expressed on microglia following glioma CM incubation also by FACS (showing the protein level performed in our hands).

4. The authors then investigate the effect of SELP on microglia phenotypes and perform flow-cytometry for "M2" markers CD163 and CD206, and find that microglia incubated with glioma CM have increased CD163 and this is reduced in SELP KD. They look at mRNA levels of "M1" vs. "M2" markers after treatment with recombinant SELP. They check for IL10 and TGFb. They make a link to phagocytosis and show that rSELP **induces phagocytosis** in microglia, and reduction of NO release.

Of note- We show that rSELP **inhibits (does not induce)** phagocytosis in microglia and macrophages.

5. They then generate a SELP KD in a **murine model** (with reduced proliferation) and show that intracranial injection results in smaller tumors and improved survival compared to control and scramble control. IF of tumors shows reduced Ki67 activity, increased caspase-3 and reduced CD31 (endothelial cell density).

We now show two human GB models of SELP KD in vivo (in addition to the murine model shown before).

The slight reduction in GB cell proliferation following SELP KD in all models is significantly and greatly enhanced following co-culture with microglia.

6. ScRNA-seq of shNC vs. shSELP tumors: they define various cell types and focus on the main ones that are sufficiently "represented". They define tumor cell "clusters" in shNC vs. shSELP and show differential expression among these tumor cell clusters. They select "differentially expressed" genes of major cellular functions, including cell proliferation and invasion. They then generate more scRNA-seq data after cd11b sorting because the microglia/macrophage population because there is "uneven internal distribution"

among shNC vs. shSELP tumors. They claim that among clusters, there were differences in shNC vs. shSELP. They suggest that cells from shSELP score more highly for a "neurodegenerative" signature while other clusters scored more highly for an "antigen presentation and chemokine signature".

7. They repeat the *in vivo* experiment with hey another model in immunocompromised mice and find again that shSELP results in smaller tumors.

In order to be consistent with the same GB cells used throughout the manuscript, we now added the same patient-derived cells (PD-GB4) SELP KD study *in vivo*, which resulted in similar findings.

Of note- The slight reduction in GB cell proliferation following SELP KD in all models, including the new GB PD KD *in vivo* model presented now, is significantly and greatly enhanced following co-culture with microglia.

8. They repeat the *in vivo* experiment in syngeneic model and a pdx model using a SELPi and find results concordant with shSELP models.

Indeed, we find that the results were concordant with the shSELP models (including the additional patient-derived model). Interestingly, as seen in the graph below (Figure R1), SELPi did not inhibit the proliferation of GB cells grown as mono-culture. However, it did inhibit their growth when grown in spheroid models co-cultured with microglia (Figure 2, Supplementary Figure 5, Supplementary Figure 7).

Figure R1: SELPi did not affect GB cell proliferation in 2D mono-culture.

The manuscript is confusingly written and really needs to be streamlined. The authors move between murine and human models and cell types without obvious rationale which makes it difficult to really understand the message of the paper.

We have now reorganized the manuscript and streamlined it to include all experiments in the main manuscript using patient-derived cells (PD-GB4). We have now reproduced these findings in a systematic manner in additional murine and human models and presented them in the Supplementary Information. We believe that this presents now a clearer picture.

Of note- we originally added murine, human and patient-derived GB cells interchangeably as we were asked to demonstrate SELP's role in multiple GB models to verify that this SELP-PSGL-1 crosstalk is not a peculiarity of one single GB cell line/cell type.

Apart from this, there are several technical and experimental which limit the interpretability of the data and in some instances simply miss the correct controls. Also, the entire manuscript is void of any necessity experiments, below are a few examples. I will address each of the major results using the same numbering:

Major

Point 1: In their description of results (and throughout the MS) the authors conflate macrophages with microglia; for example, they consider CD11b, which is a non-specific surface marker, for some of their experiments. These populations need to be distinguished with more accurate granularity.

For the *in vitro* assays, we isolated murine microglia from naïve mice in which most of the CD11b positive cells are microglia (Florian Klemm *et al.*, Cell, 2020) compared to pathological conditions in which other

immune cells are recruited (i.e. BMDM amongst them). CD11b positive selection is the most common method for isolation of microglia from **naïve** mice (see references below). Nonetheless, as shown in the Supplementary Information, we characterized CD11b-positive cells for microglia specific markers (P2Y12 and TMEM119) showing that both the murine (freshly-isolated) and the human microglia (in addition to the characterization performed by Celprogen, from which we purchased the human microglia cells) we used are positive for these two markers. We have now added the gating strategy (Supplementary Figure 1C-D). For the *in vitro* assays using macrophages, we isolated BMDM from the bone-marrow of naïve mice (Supplementary Figure 10). The Results section clearly indicates which experiments were performed using human microglia, murine microglia or BMDM and shows their characterization.

C

D

Supplementary Figure 1. Molecular and functional characterization of human and murine microglia. C-D. Freshly-isolated murine CD11b-positive cells and purchased human microglia are positive for microglia specific markers. Flow cytometry analysis showing positive staining for TMEM119 and P2Y12 in isolated murine CD11b positive cells (C) and primary human microglia (D).

Rothhammer V, et al., Microglial control of astrocytes in response to microbial metabolites, *Nature*, 557, 724–728 (2018).

Lodygin D, et al., β -Synuclein-reactive T cells induce autoimmune CNS grey matter degeneration, *Nature*, 566, 503–508 (2019).

Marschallinger J, et al., Lipid-droplet-accumulating microglia represent a dysfunctional and proinflammatory state in the aging brain, *Nature Neuroscience*, 23, 194–208 (2020).

Ayata P, et al., Epigenetic regulation of brain region-specific microglia clearance activity, *Nature Neuroscience*, 21, 1049–1060 (2018).

van der Poel M, et al., Transcriptional profiling of human microglia reveals grey–white matter heterogeneity and multiple sclerosis-associated changes, *Nature Communications*, 10, 1139 (2019).

Yun SP, et al., Block of A1 astrocyte conversion by microglia is neuroprotective in models of Parkinson’s disease, *Nature Medicine*, 24, 931–938 (2018).

The steroid flow-cytometry does not show any obvious differences. The authors should include raw data and gating.

We do not understand what the “steroid flow-cytometry” means... Assuming the reviewer meant to say spheroid (?), we included the gating strategy (added to Supplementary Figure 3F) and attached here the

raw data for this experiment (Figure R2). In addition, we have now added gating strategy for all the relevant experiments in the manuscript or its Supplementary Information.

Supplementary Figure 3. Immunostaining of GB mouse models, H&E of GB patient samples and GB spheroids FACS gating strategy. F. FACS gating strategy used for the evaluation of SELP expression in GB spheroids.

IC	Control	MG CM			IC	Control	MG CM
0.05	0.09	0.17			1	1.8	3.4
0.05	0.09	0.36			1	1.8	7.2
0.0634	0.12957	0.3371			1	2.043691	5.317035
				AVE	1	1.88123	5.305678
				SD	0	0.140695	1.900025

Figure R2. Raw data of SELP expression by GB spheroids presented in Figure 1.

In figures 1A, J, K, and really throughout the entire manuscript, the intensity of DAPI varies between conditions, indicating that these images were taken with different exposure times or were stained differently. This makes a comparison (leave alone quantification) of any IF in this manuscript impossible. There is also a lot of background staining, for example several non-cellular punctae in the PSGL1 staining, which indicate that these antibodies are not adequately titrated.

DAPI staining and autofluorescence of the tissue may differ between samples, especially between malignant and healthy tissues (see indeed new Figure 1A, I, J). All images were taken using the exact same parameters for all the channels. In addition, background was subtracted from each individual channel and the quantification was performed using only the marker-corresponding channel. Thus, DAPI intensity did not influence image quantification of different markers.

The authors also conflate expression and secretion which makes interpretation of their claims extremely difficult to follow.

As we observed SELP to be expressed on GB cell membrane and to be secreted, we reported our findings and described them. We did not conflate between the two forms of SELP, we explained these differences throughout the whole manuscript and discussed this issue in the Discussion section. We have now elaborated our Discussion on the differences between sSELP and membrane-bound SELP and the possible impact of each form of SELP on GB tumorigenesis.

The use of gray scale quantification is somewhat outdated and extremely prone for errors - the authors should show in immunoblots the differences of SELP.

We show SELP expression by Cytokine array (presented as immune-dot blot), ELISA, FACS and immunostainings for the protein level and RT-PCR for the RNA level. Each of these experiments uses a different method of analysis relevant to the question and SELP form discussed. We also show SELP expression in different *in vitro* GB models, several GB mouse models and several GB patient FFPE samples.

Point 2: This experiment misses essential controls: first, there is no SELP KD alone group included. It is impossible to separate effects of SELP KD alone vs. SELP KD with co-cx.

As we show that microglia cells enhanced GB cell proliferation (Figure 1, Supplementary Figure 2), adding only SELP KD group to these experiments (shown in Figure 2) will not be reflective of the actual influence of SELP KD. These are different experimental settings which influence the cell proliferation, migration, gene expression, etc. Thus, we compared the proliferation of SELP KD GB cells in mono-culture to control-WT and shNC mono-cultures (Supplementary Figure 5), and the SELP KD in co-culture to control-WT and shNC co-cultures (Figure 2). As we mentioned and discussed, SELP KD had only minor effect on GB cell proliferation in mono-culture (relative to its controls) compared to SELP KD in co-culture with microglia which inhibited cell proliferation to a higher extent.

As outlined above, the entire manuscript is void of rescue experiment. The authors need to re-express an SELP ORF in the KD (or KOs) and repeat experiments.

On a more basic level, the fact of the matter is that the authors did not achieve a convincing KD as shown in Fig. 2B - there is still a large population of cells with preserved SELP expression.

Since we used shRNA knockdown, and not genetic knockout, introducing ORF plasmid in these cells will not necessarily rescue SELP expression or influence cell proliferation or invasion. It will depend on the competition between the silencing plasmid and the ORF plasmid and the outcomes will be difficult to interpret. Thus, we show the effects of SELP inhibition on GB cells using both molecular and pharmacological agents in a variety of *in vitro* and *in vivo* models. Nonetheless, we show the effects of SELP KD *in vitro* and *in vivo* even when the cells still express some levels of SELP.

In addition, we are showing rescue experiments for glioma-associated microglia/macrophages (GAMs) phenotype (Figure 3, Supplementary Figures 8-10) using recombinant SELP (rSELP) in the presence or absence of SELPi or anti-PSGL-1 neutralizing antibody.

They should include gating data. The fluorescence intensity shown in C is inconsistent with the data shown in B.

The same gating strategy was used as shown above (FACS gating strategy used for the evaluation of SELP expression in GB spheroids), and now shown in the manuscript Supplementary Information. Figure 2B shows FACS histogram as the instrument raw data output. Figure 2C shows the median fluorescence intensity relative to the isotype control as demonstrated above (Figure R1).

Point 3 and 4: Co-authors on this manuscript with training in immunology should be aware of the outdated use of M1 vs. M2 nomenclature in general, but in particular their classification based on 1-2 marker proteins/genes.

We have now excluded M1/M2-like nomenclature and define GAMs phenotypes as pro-inflammatory/anti-tumorigenic and anti-inflammatory/pro-tumorigenic programming activation states evaluating signaling and functional characteristics of these states including cytokines expression, phagocytosis and nitric oxide release in addition to a variety of immune-related markers. In addition, we have now focused on signaling and functional assays rather than immune-related markers which are shown in the Supplementary Information.

since the authors have single-cell data, they can infer M1-like or M2-like cells, but this was not performed here - why not?

As mentioned above, we have now changed our terminology from M1 or M2 to pro-inflammatory/anti-tumorigenic and anti-inflammatory/pro-tumorigenic programs. Our single cell analysis provides us with a broader picture and further demonstrates that the cells are not necessarily polarized but are on a spectrum between these two states. For example, in the shSELP condition we see that microglia/macrophage cells induce gene programs such as neurodegenerative signature or antigen presentation and chemokines signature related to the pro-inflammatory/anti-tumorigenic phenotype.

This entire section is extremely confusing moving between models to show different aspects of "M1 vs. M2" features. Can the authors simply show all (at least once) in one model, but show it consistently.

To have a clearer statement, the main figure now shows all the different aspects of GAMs phenotype in one model (hMG) using the same experimental settings (Figure 3). Additional models and experimental settings which support and strengthen these observations are now included in the Supplementary Information (Supplementary Figures 8-10).

Of note- in order to have more clarity on a single GB model, we have now kept and added all data shown in the main figures on the PD-GB4 model. Additional human and murine GB models were organized in the Supplementary Information.

Point 5: The fatal flaw of these in vivo experiment (and subsequent ones) is the fact that SELP KD have a reduced proliferation rate in vitro as the authors show in the supplement. Of course this affects the (reduced) growth in vivo and will result in a prolonged survival of animals. The correct control is a matched cell line with the same proliferation rate.

As discussed above, and in the manuscript, we observed that SELP KD delayed GB cell proliferation in the presence of microglia to a higher extent than in their absence (Figure 2, Supplementary Figure 5). In addition, our results clearly show the influence of SELP on GAM phenotypes *in vitro* and *in vivo* using different models and analyses. Using a different cell line as a control, even if exhibits similar proliferation rate, may influence the interactions of GB cells with their microenvironment, among other parameters which affect tumor progression.

The authors only measured tumor volume 3 times in the entire experiment over the course of 15 days?

MRI imaging was performed 2-3 times a week assessing tumor volume in all the mice included in the experiment. However, brain tumors in mice are detectable only 7-10 days (varies in the different models) post tumor inoculation, hence we show tumor volume based on the MRI images from day 0 (no tumors detected) and then only from day 9 and 14.

This comment stating that tumors were detected by MRI only from day 9 was added to the manuscript.

Ki67 staining is non-specific in some sections shown, staining cell membranes etc. All images suffer from a too high contrast and show different intensities/exposures, they need to be repeated at the same exposure time. Also, caspase staining shows a lot of non-specific background which is much higher in the shSELP group.

As mentioned above, images were taken using the same parameters including intensity and exposure time. As more positive staining is detected in the tissue, the intensity and the scattering of the signal is higher. Since we use the same parameters, tissues with higher positive staining exhibited stronger intensity and increased light scattering. Contrast, brightness, gain and exposure time were adjusted according to secondary antibody-only staining or negative control tissue and were kept constant for all images. In addition, background was subtracted from each image and specificity was validated by the negative control staining.

Point 6: First, The authors do not provide convincing evidence that what they call tumor cells are indeed tumor cells. They should provide inferred CNV analysis which will show pathognomonic CNV alterations which are well reflected in single-cell data, as previously shown by many others.

To further confirm the annotation of the tumor cell clusters, in addition to signature projection and markers, we used inferred CNV analysis as previously described (Tirosh, I., Izar, B., Prakadan, S. M., Wadsworth, M. H., Treacy, D., Trombetta, J. J., ... Garraway, L. A. (2016). Dissecting the multicellular ecosystem of metastatic melanoma by single-cell RNA-seq. *Science*) (Supplementary Figure 14C). The analysis identified large multiploidy genomic region in the clusters annotated as tumor cells in both shSELP and shNC populations, while the rest of the clusters exhibited no such features.

Supplementary Figure 14. Identified Tumor cell clusters found in whole-tumor single-cell RNA-seq and their distribution between shNC and shSELP GL261 tumors. C. Inferred CNV analysis. The analysis identified large multiploidy genomic region in the clusters annotated as tumor cells in both shSELP and shNC populations, while the rest of the clusters exhibited no such features.

The interpretation of cell abundances in A and B is impossible, because cells were derived from tumors of different sizes. It is a well known bias that smaller tumors have a vastly different immune composition (typically more T cells, less myeloid cells) compared to larger tumors of the same histology/experiment.

As described in the manuscript, for each sample, we used the same amount of CD11b-positive cells, TILs and the remaining cell-suspension which contained the cancer cells as well as other brain microenvironment components. Thus, the amount of the different **cell-types** cannot be concluded from these experiments and we did not claim to do so. Our analysis is focused on the distribution of the **sub-clusters** within each cell type and we did not include any statements regarding the ratios of T cells, myeloid or other cell types between the different groups based on the Single-Cell analysis. Since we used equal amount of each cell type, it allows us to determine the abundances and distribution of the different **sub-clusters** within each cell type.

The fact of the matter is also that there is no difference in the result shown in Fig. 5F, and the neurogenerative vs. antigen presentation signatures are evenly distributed between control and SELP KD. The differences of the expression of these signatures between the groups are statistically significant as indicated in the graphs ($2E-16$ for neurodegenerative signature, and 0.002 for antigen presentation and chemokines signature). As can be seen in the new Figure 6E, 6F, 6H and 6I (Figure 5 in the submitted version), these signatures include clusters 0 and 1 which were significantly more abundant in the shSELP group compared to the shNC group.

Points 7 and 8: same limitation of in vivo experiments as above. shSELP have a lower proliferation rate, which will result in smaller tumors in vivo (and therefore better survival) and increased CD8 T cells (because the tumors are smaller). The authors need to at least correct each quantification for mg of tumor. Also, Ki67 in Fig. 6 C seems to stain everything in a highly non-specific manner in the WT tumors while in the SELP KD there are several punctae that do not seem to correspond to a nucleus, suggesting further that the authors measure a lot of background artifact.

Staining shows only tumor areas using the same size of the field of view (FOV). It does not show total number of positive cells, but rather the total area of positive staining relative to the control group. In addition, FACS analysis shows the percent of CD8 and Tregs out of the total cells isolated. We agree with

the reviewer that the immune composition may differ between different stages of tumor development. However, we show the effects of SELP inhibition on the different tumor characteristics at the same time post tumor inoculation, indicating on the actual state of the tumors and the therapeutic effects of our treatments. Single-Cell RNA-seq analysis revealed an increase in T cell recruitment signature of microglia which support the increase in CD8 T cells. We did elaborate on the different possible explanations for the observed effects in the Discussion section.

As mentioned above regarding the Ki-67 staining, images were taken using the same parameters including intensity and exposure time. As more positive staining is detected in the tissue, the intensity and the scattering of the signal is higher. Since we use the same parameters, tissues with higher positive staining exhibited stronger intensity and increased light scattering. Contrast, brightness, gain and exposure time were adjusted according to secondary antibody-only staining or negative control tissue and were kept constant for all images. In addition, background was subtracted from each image and specificity was validated by the negative control staining.

As shown above, SELPi did not inhibit GB cell proliferation when grown in mono-culture on 2D (Figure R1).

In Fig 7C, there is a discrepancy between the standard deviations when in C and E (in that, the SD in 7C is quite large, consistent with biological noise, while in E there appears to be nearly no error rate for the SELP KD tumors despite).

Figure 7 shows **SELPi** treatments (and not KD) in different GB models. Figure 7C shows tumor volume of **murine iAGR53** (mesenchymal subtype) tumors detected by MRI imaging, while Figure 7E shows tumor growth of human **U251 tumors** detected by **CRI Maestro™ imaging** based on fluorescence intensity. These are different models, different group sizes, different treatment regimen and different imaging detection systems.

Why do the authors show different regions in 7F and not sequential tissue sections?

As explained here and in the manuscript, staining quantification was performed using several mice per group and several images per mouse. **Representative** images were chosen by quantified values which best represent group average. In most cases, since different parameters were analyzed, distinct tumor areas correlate with the averages of the different markers evaluated.

minor:

page 29 missing figure number in suppl. Figure page 35 needs to be corrected (Figure 7A C, D)- C)

We apologize for these oversights. We have now systematically these errors.

REVIEWER COMMENTS

Reviewer #1 (Remarks to the Author):

The authors have responded to my concerns quite satisfactorily. There are no additional questions.

Reviewer #2 (Remarks to the Author):

The authors have satisfactorily responded to all my comments and made the necessary changes to the manuscript.

Reviewer #3 (Remarks to the Author):

Overall authors did a good job addressed most of my concerns. Manuscript is much better and easier to read.

I only have a couple of comments-

I am still concerned about extracranial tumor growth, especially as the authors state it happens only in the control group. It is apparent in multiple MRI images in the manuscript. Not including it in tumor volume is a selection that applies to only one group. These mice are used for survival experiments. I wonder how they exclude its effect from survival experiments. At the minimum, the table with the exact number and frequency of these tumors should be included. The extracranial tumors have different microenvironments, and it is a concern to have both in the same mouse and compare that to only cranial tumors in other groups.

Citing manuscripts by others that used SELPi in other systems and not showing target inhibition in their system is not enough; at the used doses in their system, authors should provide the degree and specificity of target inhibition. shRNA approach is only complimentary.

Reviewer #4 (Remarks to the Author):

The revised manuscript is significantly improved and the authors have adequately addressed several comments. However, some concerns remain:

Previously raised Point 2:

- The authors did not perform a simple, and in my opinion important control (SELP KD alone). They refer to prior data generated in other experiments, but they simply need to demonstrate the SELP KD phenotype only within the same experiment.

- since they initially chose a genetic approach (SELP knockdown by shRNA), it is critical to demonstrate rescue experiments using similar approaches. The authors claim that this was not possible, because they used shRNA, and rescue with an ORF were to compete with residual endogenous SELP. My recommendation is that they perform a CRISPR-Cas9 knockout, and rescue with the SELP allele to overcome this issue. As they author must well know, neutralizing antibodies and "rescuing" with recombinant protein, is subject to other biases (degradation, non-specific or unknown effect, or insufficient binding etc). Thus a genetic rescue is indispensable.

Previously raised Point 5:

The authors justification is insufficient. They refer to other results in their manuscript (which have similar flaws). The in vivo results are uninterpretable without correcting for growth rate of different genotypes. The authors seem to be unwilling to perform the proposed or alternative experiments to address this concern. They state that tumor volumes were measured 2-3 times a week, yet

their results shown in Fig. 5 A show exactly 2 data points (day 9 and day ?14).

Regarding Ki67 staining: even if these images were acquired at the same settings, and the difference in intensity is due to inter-sample variability, this still doesn't explain why Ki67 staining is found extranuclear. This clearly indicates that the antibody condition for this experiment simply did not work. I find this finding concerning, and the staining has to be repeated in an independent experiment.

Previously raised Point 6:

The authors state that the finding in 5F is "statistically significant". The fact of the matter is 1) there is no difference and 2) "statistical significance" was reached, because the authors used a t test, which is an inappropriate test for any single-cell RNA-seq (or high-dimensional single-cell method for that matter). To test statistical significance, more appropriate tests, e.g. hypergeometric test. The current p value is conflated.

Reviewer #3

Overall authors did a good job addressed most of my concerns. Manuscript is much better and easier to read.

I only have a couple of comments-

I am still concerned about extracranial tumor growth, especially as the authors state it happens only in the control group. It is apparent in multiple MRI images in the manuscript. Not including it in tumor volume is a selection that applies to only one group. These mice are used for survival experiments. I wonder how they exclude its effect from survival experiments. At the minimum, the table with the exact number and frequency of these tumors should be included. The extracranial tumors have different microenvironments, and it is a concern to have both in the same mouse and compare that to only cranial tumors in other groups.

We would like to emphasize that the extracranial tumors were observed at late stages of the disease as shown below, meaning that for most of the experiment, all the tumors have interacted with the same intracranial microenvironment and only later at an advanced stage, they broke through the skull, probably via the hole drilled for the inoculation of the cells.

Extracranial growth was observed when the mice already had large tumors, thus it had minimal effect, if any, on the survival of the mice. However, in order to exclude the extracranial growth effect from the survival data, we can correct for it by defining it as an event for withdrawal from the experiment, *i.e.* “time to progression event” (similar to other events such as body weight loss beyond 20% or neurological symptoms beside death). We show here an example for this correction for the study on mice bearing orthotopic U251. As can be seen, this approach will further demonstrate the important role of P selectin in GB progression as it did not significantly affect the analyzed survival data.

In addition, as we mentioned that this was observed mainly in the control groups (since those had fast-developing tumors), we did observe extracranial tumors in rare cases in the treated groups. Thus, it represents the reduced tumor growth inside the brain prior to the extracranial growth. As per the reviewer’s request, we include a table with the numbers and frequencies of the extracranial growth.

U251 SELP knockdown			
	WT	shNC	shSELP
No Frequency	3	3	0
	1/3	1/2.6	0

PD-GB4 SELP knockdown			
	WT	shNC	shSELP
No Frequency	2	3	0
	1/4	1/3	0

GL261 SELP knockdown			
	WT	shNC	shSELP
No Frequency	5	7	2
	1/5.8	1/4.1	1/14.4

GL261 SELPi treatment			
	WT	shNC	shSELP
No Frequency	2	1	1
	1/2.5	1/5	1/5

PD-GB4 SELPi treatment			
	WT	shNC	shSELP
No Frequency	1	0	0
	1/5	0	0

iAGR SELPi treatment			
	WT	shNC	shSELP
No Frequency	4	3	1
	1/3.5	1/3	1/15

Citing manuscripts by others that used SELPi in other systems and not showing target inhibition in their system is not enough; at the used doses in their system, authors should provide the degree and specificity of target inhibition. shRNA approach is only complimentary.

It is common to use previously characterized and published materials especially that its SELP specificity was shown independently of the system. We did show previously that SELPi inhibited the internalization of SELP-targeted nanoparticles in a concentration-dependent manner, using similar systems and now we are adding the citation in the relevant place (Ferber, S., *et al.*, Co-targeting the tumor endothelium and SELP-expressing glioblastoma cells leads to a remarkable therapeutic outcome. *eLife*, 2017. 6.). Adding further experiments in our systems beyond the ones already presented, in order to show target inhibition and specificity in tumor spheroids and *in vivo*, is challenging and will require significant time and efforts. Exploiting complementary methods, we have already shown that our findings are SELP-specific using shSELP, neutralizing antibody to SELP and selectivity of inhibition of several ligands in our revision. We also showed that SELPi rescues the effects of rSELP (Figure 3, Supplementary Figure 9, 10) which indicates that it is specific.

Reviewer #4

The revised manuscript is significantly improved and the authors have adequately addressed several comments. However, some concerns remain:

Previously raised Point 2:

- The authors did not perform a simple, and in my opinion important control (SELP KD alone). They refer to prior data generated in other experiments, but they simply need to demonstrate the SELP KD phenotype only within the same experiment.

We did perform a proliferation assay of PD-GB4 cells using all the six groups in the same experiment. The results show that shSELP proliferation in a co-culture is similar (no significant difference) to that of the WT cell proliferation in mono-culture, and that shSELP cell proliferation in the co-culture is similar to shSELP proliferation in mono-culture. These results suggest that SELP knockdown eliminates the differences observed between mono-culture and co-cultures, hence inhibiting the microglia-enhanced GB cell proliferation. We now replaced the graph in Figure 2 to the one below that includes all groups as requested by the reviewer.

- since they initially chose a genetic approach (SELP knockdown by shRNA), it is critical to demonstrate rescue experiments using similar approaches. The authors claim that this was not possible, because they used shRNA, and rescue with an ORF were to compete with residual endogenous SELP. My recommendation is that they perform a CRISPR-Cas9 knockout, and rescue with the SELP allele to overcome this issue. As they author must well know, neutralizing antibodies and "rescuing" with recombinant protein, is subject to other biases (degradation, non-specific or unknown effect, or insufficient binding etc). Thus a genetic rescue is indispensable.

Each system used for knockdown has its advantages and disadvantages. This is the reason we used 3 complementary methods to inhibit SELP: (1) genetic inhibition by SELP shRNA, (2) pharmacological inhibition by small molecule SELPi, and (3) molecular inhibition by neutralizing antibody to SELP. CRISPR-Cas9 KO systems have their own limitations, and usually not resulting in 100% gene knockout, especially when using cancer cells. Previous experiment with several tumor types resulted in 30-60% KD by CRISPR-Cas9 and took us 2 years to conclude all the required experiments. Therefore, we do not think this is necessary for this manuscript as we performed rescue experiments using different methods from those suggested by the reviewer, including negative control shRNA showing specific SELP silencing using multiple GB models.

Previously raised Point 5:

The authors justification is insufficient. They refer to other results in their manuscript (which have similar flaws). The *in vivo* results are uninterpretable without correcting for growth rate of different genotypes. The authors seem to be unwilling to perform the proposed or alternative experiments to address this concern.

We still do not think that using a different cell line is a correct nor an appropriate control. Our results, including the new proliferation assay shown above, indicate that the reduced proliferation rate of shSELP GB cells is mainly due to perturbation of GB-microglia interactions. We have now clearly showed that SELP mediates GB-microglia interactions which result in enhanced GB cell proliferation. In addition, as shown in the previous rebuttal letter, SELPi did not affect GB cell proliferation in mono-cultures *in vitro* in 2D, however, it did show similar tumor growth inhibition *in vivo* as the SELP KD. As we used all the essential controls, including the use of negative controls and vehicles, we do not think that an additional control is required. Using a different GB cell line that has similar proliferation rate as the shSELP GB cell lines that we used, as the reviewer is suggesting has limitations as well. These are two different cell lines that are distinct by many parameters beyond growth kinetics *in vitro*. Of note, we have a different study on 3D models of GB that show that similar growth kinetics *in vitro* of pairs of cell lines are not predictive of their growth in 3D nor *in vivo* (manuscript currently in review in *Nature Biomedical Engineering*).

They state that tumor volumes were measured 2-3 times a week, yet their results shown in Fig. 5 A show exactly 2 data points (day 9 and day ?14).

Tumor growth curve showed only time points in which all the mice in all the groups were still alive. However, we did, of course, continue to measure tumor volume by MRI, as shown below for Figure 5A. We now added this to the Supplementary Information.

Regarding Ki67 staining: even if these images were acquired at the same settings, and the difference in intensity is due to inter-sample variability, this still doesn't explain why Ki67 staining is found extranuclear. This clearly indicates that the antibody condition for this experiment simply did not work. I find this finding concerning, and the staining has to be repeated in an independent experiment.

We have now repeated the Ki-67 staining. The results show specific nuclear staining and demonstrated reduced proliferation in shSELP and SELPi treated tumors, as was observed before. We replaced all the relevant images in the Figures and re-analyzed the quantification of the Ki-67 staining.

U251 SELP knockdown

U251 SELPi treatment

GL261 SELP knockdown

GL261 SELPi treatment

Previously raised Point 6:

The authors state that the finding in 5F is "statistically significant". The fact of the matter is 1) there is no difference and 2) "statistical significance" was reached, because the authors used a t test, which is an inappropriate test for any single-cell RNA-seq (or high-dimensional single-cell method for that matter). To test statistical significance, more appropriate tests, e.g. hypergeometric test. The current p value is conflated.

As described in the methods, given a gene signature (list of genes), a cell-specific signature score was computed by averaging the score of two methods - ranking based and expression based. Thus, a hypergeometric test would not be the straightforward comparison between cell scores in group A and cell scores in group B. In order to address the reviewer's comment, we now also used Wilcoxon rank sum test and show that the differences reported are significant (p value $< 2.2 \times 10^{-16}$ for the neurodegenerative signature and p value = 8.969×10^{-16} for the T cell communication signature).

To further corroborate these significant differences, we used two additional independent statistical approaches for signature enrichment in single cell data. The first method was based on the recently published CERNO algorithm (Mattei, D., *et al.*, Enzymatic Dissociation Induces Transcriptional and Proteotype Bias in Brain Cell Populations. *International Journal of Molecular Sciences*, 2020. 21(21): p. 7944.; Zyla, J., *et al.*, Gene set enrichment for reproducible science: comparison of CERNO and eight other algorithms. *Bioinformatics*, 2019. 35(24): p. 5146-5154.). Briefly, we used the CERNO method to recalculate the enrichment for a specific gene signature per cell. First, genes were ranked by their normalized expression level, followed by calculation of the statistic $F = -2 \sum_{i=1}^N \left[\ln \left(\frac{r_i}{N_{tot}} \right) \right]$, where N_{tot} the total number of the gene in the experiment, and the size of the gene signature is N , and for every $g_i \in G$: $r_i = \text{rank}(g_i)$ as shown before, $F \sim \chi^2_{2N}$, therefore, we used the chi-square CDF function to calculate a p – value per cell, then, we used FDR correction on the result of each condition. Finally, we used Fisher exact test to see if one condition is more enriched with cells that are significantly enriched in the CERNO test, than the other condition. The CERNO enrichment test shows that the differences reported are significant (p value = 4.4×10^{-9} for the neurodegenerative signature and p value = 6.48×10^{-8} for the T cell communication signature).

The second enrichment method was the AUCell (Aibar, S., et al., SCENIC: single-cell regulatory network inference and clustering. *Nat Methods*, 2017. 14(11): p. 1083-1086). AUCell uses the "Area Under the Curve" (AUC) to calculate whether a gene signature is enriched within the expressed genes per cell. Briefly, the distribution of AUC scores across all the cells allows to explore the relative expression of the signature. AUCell is independent of the gene expression units and the normalization procedure as the scoring method is ranking-based. Following the AUCell enrichment scoring (default parameters), we used Fisher exact test to see if one condition is more enriched with cells that are significantly enriched in the AUCell test than the other condition. The AUCell enrichment test shows that the differences reported are significant (p value = 0.02 for the neurodegenerative signature and p value = 9×10^{-5} for the T cell communication signature).

REVIEWERS' COMMENTS

Reviewer #3 (Remarks to the Author):

No more comments!

Reviewer #4 (Remarks to the Author):

The authors reasonably addressed or discussed my concerns